**Investigation**

# Estimating the additive genetic variance for relative fitness from changes in allele frequency

Manas Geeta Arun (ID) ,[1,]* Aidan Angus-Henry (ID) ,[1,2] Darren J. Obbard,[1] Jarrod D. Hadfield (ID) [1]

[1]Institute of Ecology and Evolution, The University of Edinburgh, Ashworth Laboratories, Charlotte Auerbach Road, Edinburgh EH9 3FL, Scotland, United Kingdom
[2]Charité - Universitätsmedizin Berlin, Charitéplatz 1, Berlin 10117, Germany

*Corresponding author: Institute of Ecology and Evolution, The University of Edinburgh, Ashworth Laboratories, Charlotte Auerbach Road, Edinburgh EH9 3FL, Scotland, United Kingdom. Email: manas.geetaarun@ed.ac.uk

The rate of adaptation is equal to the additive genetic variance for relative fitness ($V_A$) in the population. Estimating $V_A$ typically involves obtaining suitable measures of fitness on a large number of individuals with known pairwise relatedness. Such data are hard to collect and the results are often sensitive to the definition of fitness used. Here, we present a new method for estimating $V_A$ that does not involve making measurements of fitness on individuals, but instead tracks changes in the genetic composition of the population. First, we show that $V_A$ can readily be expressed as a function of the genome-wide diversity/linkage disequilibrium matrix and genome-wide expected change in allele frequency due to selection. We then show how independent experimental replicates can be used to infer the expected change in allele frequency due to selection and then estimate $V_A$ via a linear mixed model. Finally, using individual-based simulations, we demonstrate that our approach yields precise and accurate estimates over a range of biologically plausible scenarios.

## Graphical Abstract

**Additive genetic variance for relative fitness from changes in allele frequency**

Rate of adaptation ≈ Additive genetic variance for relative fitness ($V_A$)

Average effects for rel. fitness ($\boldsymbol{\alpha}$)

Genetic diversity/LD matrix ($\mathbf{L}$)

Expected allele frequency change due to selection ($E(\Delta\mathbf{p})$)

$$V_A = \boldsymbol{\alpha}^\top \mathbf{L} \boldsymbol{\alpha}$$
$$= E(\Delta\mathbf{p})^\top \mathbf{L}^{-1} E(\Delta\mathbf{p})$$
$$= E(\Delta\mathbf{p})^\top \boldsymbol{\alpha}$$

Gen = 0 Gen = t ← - - Δp - - → Gen = τ

Base population Compute **L**

rep1, rep2, rep3, rep4

$$\Delta\mathbf{p}_{rep1} = \mathbf{L}\boldsymbol{\alpha} + E(\Delta\mathbf{p}_{rep1})_{unpred} + \Delta\mathbf{p}_{drift,rep1}$$
$$\Delta\mathbf{p}_{rep2} = \mathbf{L}\boldsymbol{\alpha} + E(\Delta\mathbf{p}_{rep2})_{unpred} + \Delta\mathbf{p}_{drift,rep2}$$
$$\Delta\mathbf{p}_{rep3} = \mathbf{L}\boldsymbol{\alpha} + E(\Delta\mathbf{p}_{rep3})_{unpred} + \Delta\mathbf{p}_{drift,rep3}$$
...

Linear mixed model (random locus effects)

$$\Delta p_{jk} \sim l_j + \epsilon_{jk}$$
(pred. selection) (unpred. selection + drift)

Keywords: Fisher's Fundamental theorem of natural selection; rate of adaptation; quantitative genetics; evolve and resequence

## Introduction

Despite its simplicity, the fundamental theorem of natural selection (FTNS) (Fisher 1930, 1958) is arguably one of the most central results in evolutionary biology, providing a concise mathematical statement of how quickly a population is expected to adapt. It describes the per generation gain in the mean fitness of a population as a result of natural selection, assuming the "environment" (including variables intrinsic to the population, such as density or allele frequencies) are held constant (Ewens 1989; Frank and Slatkin 1992). The crucial insight the FTNS provides is that the rate of this increase in mean fitness is exactly equal to the additive genetic variance for relative fitness ($V_A$) in the population (Burt 1995; Grafen 2015). Consequently, estimating $V_A$ is one of the "*holy grails*" of evolutionary genetics (Walsh 2022), despite the FTNS being criticised for its inability to predict the actual gain in mean fitness (Price 1972; Ewens 1989, 2024) (also see Edwards 1994 for an historical review of the debate around FTNS).

A number of attempts have been made to measure $V_A$ in wild populations. Typically, these have involved long term studies on natural populations in which the lifetime reproductive success of a large number of individuals has been measured. Combining these fitness data with information on the relatedness among individuals in a (generalized) linear mixed model approach yields estimates of $V_A$ (Kruuk 2004). This is far from straightforward in natural populations, as it can be notoriously difficult to tease apart additive genetic effects from common environmental effects, such as parental effects (Kruuk and Hadfield 2007; Shaw and Shaw 2014). In addition, wild study systems are rarely closed, meaning emigration can be misinterpreted as mortality and offspring sired outside the study area can be overlooked. Furthermore, many studies on wild populations lack genetic pedigrees, and as a consequence may miss substantial fitness variation acquired through undetected polygamy (Charmantier and Sheldon 2006; Vedder et al. 2011). In addition to these biases, estimates from wild populations also tend to come with considerable uncertainty. Burt (1995) reviewed studies estimating $V_A$ in three species of plants and three species of animals, and found that most estimates of $V_A$ were not significantly different from zero. They argued that the upper bound for estimates of $V_A$ could be as high as 0.3, but most likely less than 0.1. Consistent with this, and using a larger data set, Hendry et al. (2018) reported that estimates of $V_A$ varied between zero and 0.85, with the vast majority of estimates (73%) being less than 0.2. Overall, the mean $\widehat{V_A}$ across studies was 0.08. In a recent meta-analysis, Bonnet et al. (2022) applied Bayesian quantitative genetic methods to data obtained from 19 long term studies on wild vertebrate populations. They reported a meta-analytic mean $V_A$ of 0.185 across studies, considerably higher than those obtained by Burt (1995) and Hendry et al. (2018). Surprisingly, estimates of $V_A$ in populations of Spotted Hyenas (*Crocuta crocuta*), as well as two of the three populations of Blue Tits (*Cyanistes caeruleus*) were higher than 0.4. This is a remarkable finding, since it suggests that growth rates in these populations should increase nearly 1.5 fold every generation due to selection, provided the environment remains constant. All three meta-analyses investigating $V_A$ (Burt 1995; Hendry et al. 2018; Bonnet et al. 2022) have detected substantial variability between study systems in their estimates of $V_A$. Although most of this variability will be sampling error, the best estimate suggests the variability in actual $V_A$ across populations is also large with a standard deviation of 0.11, although there is substantial uncertainty about its exact value (95% credible intervals: 0.01–0.26) (Bonnet et al. 2022).

Measuring $V_A$ in the laboratory is considerably more straightforward, and involves either quantitative genetic breeding designs such as diallel crosses and full-sib half-sib experiments (Falconer and Mackay 1996; Lynch and Walsh 1998), or experimental techniques such as hemiclonal analysis (Abbott and Morrow 2011). While the controlled environment of the laboratory can, to a large extent, help overcome some of the challenges faced by field studies, laboratory environments often lack important features that are likely to generate fitness variation, such as parasites, predators, and competitors. Therefore, it is not entirely clear if laboratory estimates of $V_A$ are particularly relevant. Furthermore, many laboratory studies standardize their fitness measurements in a way that makes it hard to infer estimates of $V_A$ (e.g. Ruzicka et al. 2019) or work with absolute fitness but then fail to report $V_A$ (e.g. Singh A et al. 2023). However, these studies do suggest that genetic variance for fitness is likely highly dependent on the specific environment in which fitness is assayed (e.g. Punzalan et al. 2014). In one of the few laboratory studies that estimated the genetic variance for *relative* fitness, Martinossi-Allibert et al. (2018) reported estimates from isofemale lines of the bean beetle, *Acanthoscelides obtectus*, that were highly sensitive to evolutionary history, assay environment, and sex, being higher in males (0.13–0.42) than in females (0.013–0.056). A compromise between the precision of the laboratory environment and the biotic and abiotic complexity encountered by wild populations can be achieved by working with experimental populations established in the field, an approach especially tractable in annual plants. Working with field populations of the annual legume *Chamaecrista fasciculata*, Kulbaba et al. (2019) found estimates of $V_A$ that varied considerably among populations and years. Interestingly, many of their estimates were appreciably larger, with the largest estimate (calculated from Table 1 in the correction) being 3.05.

A common difficulty for current field and laboratory approaches is that, while Darwinian fitness is a deceptively intuitive concept, there is little consensus on its precise definition. In fact, it has been argued that the appropriate definition of fitness can vary depending on the context (Hendry et al. 2018). In the absence of a universal definition, empiricists can only use a measure of fitness deemed most appropriate for their study system, and it is reasonable to assume that estimates of $V_A$ are likely to be sensitive to the definition of fitness used. A useful illustration of this point is provided by two studies that estimated $V_A$ in a wild population of Red Deer (*Cervus elaphus*), using largely overlapping datasets, but markedly different definitions of fitness (Kruuk et al. 2000; Foerster et al. 2007). Kruuk et al. (2000) defined fitness as the total number of progeny produced by an individual in its lifetime and estimated $V_A$ to be 0.1, whereas Foerster et al. (2007) employed an alternative definition of fitness that measured an individual's contribution to population growth (Coulson et al. 2006) and obtained the appreciably higher estimate of 0.64.

Some of the definitional difficulties of measuring $V_A$ can be overcome by measuring $V_A$ as the rate of adaptation, rather than comparing measures of fitness among relatives. One of the earliest attempts to explore this idea was made by Fowler et al. (1997) (also see Gardner et al. 2005) in laboratory populations of *Drosophila melanogaster*. Using balancer chromosomes that allow recombination to be suppressed on the third chromosome, Fowler et al. (1997) could track the frequency trajectories of newly introduced wild-type third chromosomes over the course of 43 weeks. By modeling fitness using genotype frequency data for a number of different wild-type third chromosomes (Barton and Partridge 2000), they demonstrated the presence of substantial

**Table 1.** The ranges for the rates for non-neutral mutations used in the history phase of the full simulations implemented in msprime ($\mu_{msp}$) (the coalescent simulation) and SLiM ($\mu_{SLiM}$) (the forward simulation of the history), along with the ranges for the resulting number of non-neutral sites segregating in the population at the end of the history phase ($n_{L_s}$) in simulations with different values of the absolute degree of dominance ($|\kappa|$), the mean of the gamma distribution from which effect sizes for log absolute fitness were sampled for non-neutral mutations ($E[|\eta|]$), and the ratio of the rate of beneficial mutations to the rate of deleterious mutations in the history phase ($\mu_{ben} : \mu_{del}$).

| $|\kappa|$ | $E[|\eta|]$ | $\mu_{ben} : \mu_{del}$ | $\mu_{msp}$ | $\mu_{SLiM}$ | $n_{L_s}$ |
|---|---|---|---|---|---|
| 0 | 0.02 | 0 | $5.56 \times 10^{-8}$–$5.56 \times 10^{-7}$ | $5.56 \times 10^{-7}$–$5.56 \times 10^{-6}$ | 10,058–63,755 |
| 0 | 0.02 | 0.0002 | $5.56 \times 10^{-8}$–$5.56 \times 10^{-7}$ | $5.56 \times 10^{-7}$–$5.56 \times 10^{-6}$ | 9,742–63,797 |
| 0 | 0.02 | 0.02 | $2.0 \times 10^{-8}$–$2.0 \times 10^{-7}$ | $2.0 \times 10^{-7}$–$2.0 \times 10^{-6}$ | 3,127–29,999 |
| 0 | 0.03 | 0 | $3.6 \times 10^{-8}$–$3.6 \times 10^{-7}$ | $3.6 \times 10^{-7}$–$3.6 \times 10^{-6}$ | 6,565–42,097 |
| 0 | 0.06 | 0 | $1.8 \times 10^{-8}$–$1.6 \times 10^{-7}$ | $1.8 \times 10^{-7}$–$1.6 \times 10^{-6}$ | 2,998–18,011 |
| 0.5 | 0.03 | 0 | $7.2 \times 10^{-8}$–$3.24 \times 10^{-7}$ | $7.2 \times 10^{-7}$–$3.24 \times 10^{-6}$ | 13,738–49,299 |
| 0.75 | 0.03 | 0 | $7.5 \times 10^{-8}$–$2.07 \times 10^{-7}$ | $7.5 \times 10^{-7}$–$2.07 \times 10^{-6}$ | 16,870–38,695 |

Note that the ranges for the number of selected loci, $n_{L_s}$, are only shown for the simulations where the map length in the history phase was 0.5 morgan.

$V_A$ on the third chromosome. More recently, in a landmark study, Buffalo and Coop (2019) developed a method to estimate the amount of genome-wide allele frequency change that can be attributed to selection. The linchpin of their theory is the idea that linked selection induces across-generation covariances in allele frequency change at neutral loci, since associations between these neutral loci and their respective non-neutral backgrounds persist across generations. This new theoretical framework has the potential to pave the way for a powerful empirical tool to detect genomic signatures of linked selection (Buffalo and Coop 2020; Simon and Coop 2024). Of particular relevance here, Buffalo and Coop (2019) also show that their method can be used to obtain estimates of $V_A$, albeit under some potentially restrictive assumptions.

In this study, we present an alternative theoretical framework that relates $V_A$ to genome-wide changes in allele frequency. Using mathematical identities only, we show how $V_A$ can be obtained from an initial linkage disequilibrium (LD) matrix and expected changes in allele frequency due to selection, without making any assumptions about patterns of gene action or the relationships between genotype fitnesses and genotype frequencies. Our approach, like that of Buffalo and Coop (2019), relies on temporal genomic data and does not necessitate measuring fitness in individuals. However, in contrast to the "bottom-up" population genetic approach of Buffalo and Coop (2019), we use a "top-down" quantitative genetic approach which simplifies and generalizes some aspects of the problem.

Our aim here is three-fold. First, we derive our central theoretical result from first principles. Second, we develop the statistical machinery required to apply our result to real biological data, and validate it with individual based simulations that permit dominance effects, but assume random mating and an absence of epistasis. Third, we make a detailed comparison of our method with that of Buffalo and Coop (2019).

## Materials and methods

Table 2 summarizes the notation used in the following text and the subsection "Assumptions retained in our approach" (see below) summarizes the assumptions involved in our approach. Supplementary information S9 provides a detailed workflow for applying our method to real data.

### Outline of the theory

We consider a population consisting of $N$ diploid individuals. We assume that there are $n_L$ segregating biallelic loci in the population. Let $c_{k,i}$ and $\alpha_i$ represent the proportion of copies of an

arbitrarily chosen reference allele at locus $i$ in individual $k$ and Fisher's average effect (Fisher 1941) for relative fitness at locus $i$, respectively. The widely accepted mathematical definition of the $\alpha$'s are the regression coefficients obtained from a multiple regression of the $c$'s on relative fitness, $w$ (Fisher 1941; Lee and Chow 2013). The vector $\alpha$ can be expressed as

$$\alpha = \mathbf{L}^{-1} COV(\mathbf{c}, w) \tag{1}$$

where $\mathbf{c}$ is the vector of predictors representing the $c$'s at all loci for an individual and $\mathbf{L}$ is a symmetric $n_L \times n_L$ matrix whose $i$th (diagonal) element is the variance in the $c$'s at locus $i$ computed over individuals, while the $ij$th (off-diagonal) element describes the covariance between the $c$'s at locus $i$ and locus $j$ computed over individuals.

The breeding value for the relative fitness of individual $k$ is then

$$u_k = \mathbf{c}_k^{\mathsf{T}} \alpha$$

and the additive genetic variance for relative fitness is the variance of this quantity across individuals:

$$\begin{aligned} V_A &= VAR(\mathbf{c}^{\mathsf{T}} \alpha) \\ &= \alpha^{\mathsf{T}} VAR(\mathbf{c}, \mathbf{c}^{\mathsf{T}}) \alpha \\ &= \alpha^{\mathsf{T}} \mathbf{L} \alpha \end{aligned} \tag{2}$$

This follows from the fact that at any given point in time, $\alpha$'s are constant across individuals. In the absence of mutation and meiotic drive, the allele frequency in parents is transmitted to offspring without bias, such that the vector of expected change in allele frequencies due to selection can be expressed as Robertson's covariance (Robertson 1966; Price 1970; Queller 2017):

$$\begin{aligned} E[\Delta \mathbf{p}] &= E[\Delta \bar{\mathbf{c}}] \\ &= COV(\mathbf{c}, w) \end{aligned} \tag{3}$$

where the expectation is taken over the evolutionary process.

Substituting Equation 1 into Equation 3 gives $E[\Delta \mathbf{p}] = \mathbf{L} \alpha$, which is the multivariate analog of Equation 10 in Kirkpatrick et al. (2002). Combining this with Equation 2 yields,

$$\begin{aligned} V_A &= (\mathbf{L}^{-1} E[\Delta \mathbf{p}])^{\mathsf{T}} \mathbf{L} (\mathbf{L}^{-1} E[\Delta \mathbf{p}]) \\ &= E[\Delta \mathbf{p}]^{\mathsf{T}} \mathbf{L}^{-1} E[\Delta \mathbf{p}] \end{aligned} \tag{4}$$

Equation 4 is a general result and involves no assumptions about the patterns of dominance or epistasis for fitness, or about

**Table 2.** Notation.

| Symbol | Description |
|---|---|
| $\mathbf{B}$ | Diagonal matrix of standard deviations for the $c$'s |
| $c_{k,i}$ | Number of reference alleles at locus $i$ for individual $k$ divided by 2. |
| $C_A(t \to \tau)$ | Additive genetic covariance for relative fitness between generation $t$ and $\tau$ for a population with genetic structure equal to that in generation $t$. |
| $C_a(t \to \tau)$ | Additive genic covariance for relative fitness between generation $t$ and $\tau$ for a population with genetic structure equal to that in generation $t$. |
| $\mathbf{D}_2$ | Diagonal matrix of square-rooted eigenvalues of $\mathcal{D}$ |
| $\mathcal{D}_m$ | Matrix that gives the covariance in allele frequency changes between generation $t_m$ and $\tau_m$ in replicate $m$ due to drift. |
| $\mathcal{E}_m$ | Matrix that gives the covariance in allele frequency change estimation errors in replicate $m$. |
| $\mathbf{F}_{t\tau}$ | $\mathbf{W}_{t\tau}$ but with the diagonals set to zero. |
| $g$ | Distance between two sites in units of number of base-pairs. |
| $\mathbf{H}_{t\tau}$ | $\mathbf{W}_{t\tau}$ but with the off-diagonals set to zero. |
| $\mathbf{L}_{t,m}$ | Covariance matrix of $c$'s at time $t$ in replicate $m$. |
| $\mathbf{L}'_{t,m}$ | Covariance matrix of $c$'s at time $t$ in replicate $m$ due to being on the same gametic contribution. |
| $\mathbf{L}''_{t,m}$ | Covariance matrix of $c$'s at time $t$ in replicate $m$ due to being on different gametic contributions. |
| $\Delta\mathbf{L}'_{t,m}$ | The stochastic change in $\mathbf{L}'$ between time zero and $t$ in replicate $m$. |
| $\tilde{\mathbf{L}}_0$ | Weighted sum of $\mathbf{L}'_0$ and $\mathbf{L}''_0$ with weights for element $ij$ being 1 and $z_{0,ij}/\zeta_{0,ij}$ respectively. |
| $\mathcal{L}_{t,m}$ | A matrix that gives, when postmultiplied by $\boldsymbol{\alpha}$, the predictable change in allele frequency due to selection between generation $t_m$ and $t_m + 1$ in replicate $m$. |
| $\mathcal{L}_m$ | A matrix that gives, when postmultiplied by $\boldsymbol{\alpha}$, the predictable change in allele frequency due to selection between generation $t_m$ and $\tau_m$ in replicate $m$. |
| $\mathbf{M}_{t,m}$ | Matrix of weights for $\tilde{\mathbf{L}}_0$ that gives the covariance in allele frequency changes due to drift between time $t > 0$ and $t + 1$ in replicate $m$. |
| $\mathbf{M}_m$ | Matrix of weights for $\tilde{\mathbf{L}}_0$ that gives the covariance in allele frequency changes due to drift between time $t_m > 0$ and $\tau_m$ in replicate $m$ |
| $n_L$ | Number of loci |
| $n_{L_\mathcal{S}}$ | Number of non-neutral segregating sites |
| $N_{t,m}$ | Census population size at time $t$ in replicate $m$ |
| $N_{e_{t,m}}$ | Variance effective population size at time $t$ in replicate $m$ |
| $N_{E_{t,m}}$ | Variance effective population size at time $t$ in replicate $m$ ignoring the impact of linked-selection |
| $\mathbf{N}_{t,m}$ | Matrix of weights for $\tilde{\mathbf{L}}_0$ that gives the expected $\mathbf{L}$ at time $t > 0$ in replicate $m$. |
| $\mathbf{N}_m$ | Matrix of weights for $\tilde{\mathbf{L}}_0$ that gives the sum of the expected $\mathbf{L}$ from time $t_m > 0$ to $\tau_m$ in replicate $m$. |
| $\mathcal{N}$ | Used as a subscript to indicate the set of neutral loci. |
| $O_{ij}$ | Number of pool-seq reads spanning sites $i$ and $j$. If $i = j$, $O_{ij}$ is the coverage. |
| $p_{\bar{a}}$ | Parameter that takes $\mathbf{L}_0$ to some power |
| $\mathbf{p}_{t,m}$ | Vector of reference allele frequencies at time $t$ in replicate $m$. |
| $\Delta\mathbf{p}_{t,m}$ | Vector of reference allele frequency changes between time $t$ and $t + 1$ in replicate $m$. |
| $\Delta\mathbf{p}_m$ | Vector of reference allele frequency changes between time $t_m$ and $\tau_m$ in replicate $m$. |
| $\mathbf{P}$ | Projection matrix for allele frequencies. |
| $\mathbf{q}_{t,m}$ | Vector of alternate allele frequencies at time $t$ in replicate $m$. |
| $\mathbf{Q}_m$ | Matrix that is proportional to the covariance in allele frequency change estimation errors in replicate $m$. |
| $r_{ij}$ | Recombination rate between loci $i$ and $j$. |
| $R_{t,ij}$ | Correlation in allele count between locus $i$ and $j$ at time $t$ (Note the use of the uppercase to distinguish from the recombination rate $r$). |
| $\mathbf{R}_+$ | Matrix of recombination probabilities. |
| $\mathbf{R}_-$ | Matrix of non-recombination probabilities. |
| $\mathbf{R}$ | Correlation matrix of the $c$'s. |
| $\mathbf{S}_{\bar{a}}$ | Sampling covariance matrix for the parameters of the regression on the mean average effects, $\boldsymbol{\beta}_{\bar{a}}$. |
| $\mathcal{S}$ | Used as a subscript to indicate the set of selected loci. |
| $t_m$ | Time at which allele frequencies are first measured in replicate $m$. |
| $\mathbf{U}$ | Eigenvectors of $\mathbf{L}_0$. |
| $\mathbf{U_L}$ | Eigenvectors of $\mathbf{L}_0$ with nonzero eigenvalues. |
| $\mathbf{U}_2$ | Eigenvectors of $\mathcal{D}$ |
| $\mathcal{U}_m$ | Matrix that gives the covariance in allele frequency changes between generation $t_m$ and $\tau_m$ in replicate $m$ due to the unpredictable response to selection. |
| $V_a(t)$ | Additive genic variance for relative fitness at time $t$ |
| $V_A(t)$ | Additive genetic variance for relative fitness at time $t$ |

**Table 2.** (continued)

| Symbol | Description |
|---|---|
| $V_{\bar{A}}(t)$ | Additive genetic covariance between replicate/time-points for relative fitness in a population with genotypic composition equal to that at time $t$. |
| $V_x$ | Variance of the normal distribution from which $\log(\lambda_x)$'s are sampled |
| $\mathbf{V}_{\bar{a}}$ | Covariance matrix for the mean average effects. |
| $w$ | Relative fitness |
| $W$ | Absolute fitness |
| $\mathbf{W}_{t\tau}$ | A matrix with the $ij$th element equal to $R_{t,ji}R_{\tau,ki}(b_{\tau,i}/b_{t,i})$ |
| $\mathbf{X}$ | Design matrix for the regression of the mean average effects on $\mathbf{p}_0 - \mathbf{q}_0$. |
| $y_{k,i}$ | Genotypic contribution made by locus $i$ to the log absolute fitness ($\log(W)$) of individual $k$ |
| $Y_k$ | Genotypic value of log fitness for individual $k$ |
| $z_{t,ij}$ | $(1 - r_{t,ij})(1 - \frac{1}{2N_{e_t}})$ |
| $\boldsymbol{\alpha}_{t,m}$ | Vector of average effects for relative fitness at time $t$ in replicate $m$. |
| $\bar{\boldsymbol{\alpha}}$ | Vector of mean average effects for relative fitness. |
| $\Delta\boldsymbol{\alpha}_{t,m}$ | Vector of deviations of the average effects for relative fitness at time $t$ in replicate $m$ from the global mean. |
| $\beta_{\bar{a}}^{(1)}$ | Slope of the regression of the mean average effects on $\mathbf{p}_0 - \mathbf{q}_0$. |
| $\gamma$ | Total branch length of the tree sequence recorded in the history phase of the simulations. |
| $\zeta_{t,ij}$ | $r_{t,ij}(1 - \frac{1}{2N_{e_t}})$ |
| $\eta_i$ | Difference between the genotypic contributions to $\log(W)$ by the reference and non-reference homozygotes at locus $i$ |
| $\eta_i^{(a)}$ | Average effect for $\log(W)$ at locus $i$ |
| $\eta_{scale}$ | Scale of the gamma distribution from which $\eta$'s were sampled in the history phase of the simulations |
| $\kappa_i$ | Degree of dominance at locus $i$ for log absolute fitness. |
| $\lambda_{x,k}$ | Mean of the Poisson distribution from which the number of reads mapping to individual $k$ are sampled while simulating pool-seq |
| $\mu_{msp}$ | Mutation rate for non-neutral mutations in the coalescent part of the history phase of the simulations |
| $\mu_{SLiM}$ | Mutation rate for non-neutral mutations in the forward part of the history phase of the simulations |
| $\mu_x$ | Mean of the normal distribution from which $\log(\lambda_x)$'s are sampled |
| $\boldsymbol{\mu}_{\bar{a}}$ | Vector of expected values for the mean average effects. |
| $\sigma_{\bar{a}}^2$ | Proportionality constant that relates $\mathbf{V}_{\bar{a}}$ to $\mathbf{L}_0^{p_a}$. |
| $\sigma_0^2$ | Parameter for scaling sampling (co)variances for allele frequencies: values greater than 1 indicate overdispersion. |
| $\tau_m$ | Time at which allele frequencies are finally measured in replicate $m$. |
| $\phi_{t,\tau}$ | The ratio of genetic diversity in generation $\tau$ to genetic diversity in generation $t$ assumed constant across all selected loci. |
| $\rightarrow$ | Used above a symbol to indicate it is on the projected space. |
| $\frown$ | Used above a symbol to indicate an estimate. |

patterns of mating. An intuitive explanation of why $V_A$ can be calculated this way is to note Fisher's Fundamental Theorem states that $V_A$ is equal to the (partial) increase in mean fitness caused by evolutionary change through natural selection. Equation 2 can be expressed as a sum of $\alpha_i E[\Delta p_i]$ over all loci, $i$:

$$V_A = \boldsymbol{\alpha}^\mathsf{T} E[\Delta\mathbf{p}] \qquad (5)$$

In this, $\alpha$ represents the proportional change in the population mean fitness that is caused by a unit change in allele frequency at a locus (Fisher 1941; Kojima 1959; Lee and Chow 2013). Therefore, multiplying $\alpha$ by the actual change caused by natural selection, $E[\Delta p]$, we get the proportional change in mean fitness caused by evolutionary change by natural selection at that locus (see Eq 51.4 Price 1972, also). If we then add these changes at every locus in the genome, we obtain the total proportional change in mean fitness due to evolutionary change by natural selection, and hence $V_A$.

## Extending our approach to practical situations

Our theoretical approach assumes $\alpha$ or $E[\Delta\mathbf{p}]$ are known. In reality, neither can be directly observed and must be inferred from data on observed allele frequency change, $\Delta\mathbf{p}$. Since $\Delta\mathbf{p}$ will vary

around $E[\Delta\mathbf{p}]$ due to genetic drift, $E[\Delta\mathbf{p}]$ must be inferred using replicate observations of $\Delta\mathbf{p}$. Since time cannot be replayed, we infer $E[\Delta\mathbf{p}]$ through experimental replicates (see Buffalo and Coop 2020, also). Our theoretical model also assumes $\mathbf{L}$ is known, but since it is hard to generate experimental replicates without at least one round of reproduction, we condition on $\mathbf{L}_0$ ($\mathbf{L}$ in a generation prior to the first generation over which allele frequency change is measured). In what follows we will refer to the population at time zero (generation 0) as the "base population". Our aim is then to approximate the additive genetic variance for fitness in the base population $V_A(0)$ as $V_{\bar{A}}(0) = \bar{\boldsymbol{\alpha}}^\mathsf{T}\mathbf{L}_0\bar{\boldsymbol{\alpha}}$, where $\bar{\boldsymbol{\alpha}}$ is the mean vector of average effects averaged over time and replicates. Note that if average effects are constant then $V_{\bar{A}}(0) = V_A(0)$, but if the average effects vary then $V_A(0)$ will in general exceed $V_{\bar{A}}(0)$ which can be interpreted as the additive genetic *covariance* in fitness between replicates/time points for a population with genotypic structure equal to that in the base population (see Supplementary information S2). It is important to note that conventional methods for estimating $V_A$—such as those relying on pedigreed individuals—also estimate additive genetic *covariances* among environments in which relatives are assayed (Vehviläinen et al. 2008) and these covariances equal $V_A$ only when the average effects remain constant.

We also allow allele frequency changes to be measured over multiple generations, rather than a single generation. Thus, $\Delta \mathbf{p}_m = \Delta \mathbf{p}_{t_m,m} + \Delta \mathbf{p}_{t_m+1,m} \ldots \Delta \mathbf{p}_{\tau_m-1,m}$ is the observed change in allele frequency from generation $t_m$ to $\tau_m$ in replicate $m$, with $\Delta \mathbf{p}_{t,m}$ being the change from time $t$ to $t+1$. Note that $t_m$ (i.e. the generation in which we start recording allele frequency changes in replicate $m$) can be any positive integer. If replicate populations are derived from the base population with exactly one round of reproduction, $t_m$ would be 1 for all $m$. $\tau_m$ can be any integer greater than $t_m$. We first note that the total change in allele frequency in replicate $m$ between times $t_m$ and $\tau_m$ is

$$\Delta \mathbf{p}_m = \sum_{t=t_m}^{\tau_m-1} \left( \mathbf{L}_{t,m} \boldsymbol{\alpha}_{t,m} + \underset{D}{\Delta} \mathbf{p}_{t,m} \right) \qquad (6)$$

where $\mathbf{L}_{t,m} \boldsymbol{\alpha}_{t,m} = E[\Delta \mathbf{p}_{t,m}]$ is the expected change in allele frequency due to selection in replicate $m$ between generation $t$ and $t+1$ and $\underset{D}{\Delta} \mathbf{p}_{t,m}$ is the change due to drift. It is important to note that $\mathbf{L}_{t,m} \boldsymbol{\alpha}_{t,m}$ captures responses to both direct and indirect selection. $\mathbf{L}$ will vary over time and we decompose $\mathbf{L}$ at a particular time/replicate into a part that can be predicted by $\mathbf{L}_0$ and the action of drift and recombination, and a part that cannot be predicted. To do this we decompose $\mathbf{L}$ into $\mathbf{L}'$ and $\mathbf{L}''$, following the notation of Buffalo and Coop (2019). $\mathbf{L}'$ represents the (co)variances in the $c$'s computed over the haploid genomes of all $2N$ gametic contributions that constitute the population. Thus, the diagonal elements of $\mathbf{L}'$ are proportional to gametic (gene) diversity and the off-diagonals are proportional to gametic-phase disequilibrium. On the other hand, $\mathbf{L}''$ represents the (co)variances that arise due to alleles within the different gametic contributions of a genotype, and thus the diagonal elements are proportional to the additive coefficients of Hardy-Weinberg disequilibrium (Bulmer 1980 and Chapter 12 in Weir and Cockerham 1989), and the off-diagonal elements are proportional to the nongametic-phase disequilibrium. Given this decomposition, the dynamics of $\mathbf{L}$ under drift and recombination are (See Supplementary information S1 for details and Hill and Robertson 1968; Santiago and Caballero 1998):

$$\mathbf{L}_{t,m} = \mathbf{N}_{t,m} \circ \tilde{\mathbf{L}}_0 + \Delta \mathbf{L}'_{t,m} + \mathbf{L}''_{t,m} \qquad (7)$$

where $\circ$ is the Hadamard (i.e. element-wise) product, $\tilde{\mathbf{L}}_0$ is the weighted sum of $\mathbf{L}'_0$ and $\mathbf{L}''_0$, with the $ij$th element of $\mathbf{L}''_0$ weighted by $r_{ij}/(1-r_{ij})$ and $\mathbf{N}_{t,m}$ is a matrix with the $ij$th element being $(1-r_{ij})^t \prod_{k=0}^{t-1} (1 - \frac{1}{2N_{e_{k,m}}})$. $r_{ij}$ is the recombination rate between the two loci and $N_{e_{k,m}}$ is the effective population size in generation $k$ in replicate $m$. $\Delta \mathbf{L}'_{t,m}$ is a stochastic term that represents the accumulated change in $\mathbf{L}'$ between generations 0 and $t$ in replicate $m$ that cannot be predicted. $\mathbf{L}''_{t,m}$ is the matrix of nongametic-phase disequilbria that arises in generation $t$ in replicate $m$. In the absence of selection, $\Delta \mathbf{L}'_{t,m}$ and $\mathbf{L}''_{t,m}$ have zero expectation. In the following, it will be useful to designate $\mathcal{L}_{t,m} = \mathbf{N}_{t,m} \circ \tilde{\mathbf{L}}_0$ as the predicted $\mathbf{L}_{t,m}$, for $t > 0$, conditional on $\mathbf{L}'_0$ and $\mathbf{L}''_0$. Also, we represent the deviation of $\mathbf{L}_{t,m}$ from this prediction as $\Delta \mathbf{L}_{t,m} = \Delta \mathbf{L}'_{t,m} + \mathbf{L}''_{t,m}$. It will also be useful to denote $\boldsymbol{\alpha}_{t,m} = \bar{\boldsymbol{\alpha}} + \Delta \boldsymbol{\alpha}_{t,m}$ where $\Delta \boldsymbol{\alpha}_{t,m}$ is the deviation of the average effects (in generation $t$ in replicate $m$) from the mean of the average effects across time

and replicates. We can then decompose the change due to selection into two terms:

$$\begin{aligned} \Delta \mathbf{p}_m &= \sum_{t=t_m}^{\tau_m-1} \left( \mathcal{L}_{t,m} \bar{\boldsymbol{\alpha}} + \Delta \mathbf{L}_{t,m} \bar{\boldsymbol{\alpha}} + \mathcal{L}_{t,m} \Delta \boldsymbol{\alpha}_{t,m} + \Delta \mathbf{L}_{t,m} \Delta \boldsymbol{\alpha}_{t,m} + \underset{D}{\Delta} \mathbf{p}_{t,m} \right) \\ &= \sum_{t=t_m}^{\tau_m-1} \left( \mathcal{L}_{t,m} \bar{\boldsymbol{\alpha}} + \underset{U}{\Delta} \mathbf{p}_{t,m} + \underset{D}{\Delta} \mathbf{p}_{t,m} \right) \\ &= \mathcal{L}_m \bar{\boldsymbol{\alpha}} + \underset{U}{\Delta} \mathbf{p}_m + \underset{D}{\Delta} \mathbf{p}_m \end{aligned} \qquad (8)$$

Here, quantities subscripted with just $m$ are the sums of the relevant quantity over $t$ (for example, $\mathcal{L}_m = \sum_{t=t_m}^{\tau_m-1} \mathcal{L}_{t,m}$). The predictable change due to selection is $\mathcal{L}_m \bar{\boldsymbol{\alpha}}$ and is similar to that derived for a single locus in Santiago and Caballero (1998). The unpredictable change due to selection caused by stochastic changes in $\mathbf{L}$ and/or $\boldsymbol{\alpha}$ is $\underset{U}{\Delta} \mathbf{p}_m = \sum_{t=t_m}^{\tau_m-1} (\Delta \mathbf{L}_{t,m} \bar{\boldsymbol{\alpha}} + \mathcal{L}_{t,m} \Delta \boldsymbol{\alpha}_{t,m} + \Delta \mathbf{L}_{t,m} \Delta \boldsymbol{\alpha}_{t,m})$. The total change due to drift is $\underset{D}{\Delta} \mathbf{p}_m$. The $\underset{D}{\Delta} \mathbf{p}_m$ have zero expectation and are independent across replicates (Buffalo and Coop 2019). In what follows we also assume the $\underset{U}{\Delta} \mathbf{p}_m$ have zero expectation, although we believe there are two primary mechanisms by which this assumption can fail. First, the model for the predicted changes in $\mathbf{L}$ (i.e. $\mathcal{L}_m$) may be inaccurate such that the $\Delta \mathbf{L}_{t,m}$ have nonzero expectation. Since $\mathcal{L}_m$ is derived under the assumptions of drift and recombination only, directional changes in $\mathbf{L}$ induced by selection is an obvious mechanism. However, if the genetic architecture of fitness is sufficiently polygenic, such that selection coefficients associated with individual loci are small, selection-induced changes in $\mathbf{L}$ may be safely ignored. In other words, selection-induced changes in $\mathbf{L}$ may cause the total allele frequency change due to selection to be negligibly different from $\mathcal{L}_m \bar{\boldsymbol{\alpha}}$, the expected allele frequency change due to selection. Second, the product $\Delta \mathbf{L}_{t,m} \Delta \boldsymbol{\alpha}_{t,m}$ may have nonzero expectation if changes in the average effects are correlated with changes in patterns of diversity/LD. Again, selection-induced changes in $\Delta \mathbf{L}_{t,m}$ is a possible mechanism. Additionally, unless all gene action is additive, changes in $\mathbf{L}$ also induce changes in $\boldsymbol{\alpha}$ since then average effects depend on allele frequencies (Supplementary information S8).

However, calculating change in allele frequency over a single generation (i.e. $\tau_m - t_m = 1$) would minimize any bias since any change in $\mathbf{L}$ due to selection should be minimized and the approximations that follow will be more accurate. Although the approximations will hold better when $\tau_m - t_m = 1$, increasing $\tau_m - t_m$ (i.e. calculating change in allele frequency over multiple generations) will increase power since the changes in allele frequency due to selection will be larger.

Assuming both $\underset{U}{\Delta} \mathbf{p}_m$ and $\underset{D}{\Delta} \mathbf{p}_m$ have zero expectation we can derive the mean and covariance structure (computed over the evolutionary process) of allele frequency change in a replicate as:

$$E[\Delta \mathbf{p}_m] = \mathcal{L}_m \bar{\boldsymbol{\alpha}} \qquad (9)$$

$$\mathrm{VAR}(\Delta \mathbf{p}_m) = \mathcal{D}_m + \mathcal{U}_m \qquad (10)$$

where $\mathcal{D}_m$ and $\mathcal{U}_m$ are the covariances due to the cumulative action of drift and the unpredictable response to selection, respectively. The drift (co)variances have expectation equal to $\tilde{\mathbf{L}}_0 \circ \mathbf{M}^{(m)}$ where $\mathbf{M}^{(m)} = \sum_{t=t_m}^{\tau_m-1} \mathbf{M}_{t,m} \circ \mathbf{N}_{t,m}$ and the $ij$th element of $\mathbf{M}_{t,m}$ is $(1-r_{ij})/N_{E_{t,m}}$ (Supplementary information S2). Note that since the

predictable response to selection includes both direct and indirect selection, the relevant effective population size for the drift covariance in allele frequency does not include the effects of linked selection and for this reason we denote it as $N_E$ rather than $N_e$. Using this notation, $N_E = 4N/(2 + V_o)$ where $N$ is the census population size and $V_o$ is the variance in offspring number in the absence of additive genetic fitness variation (Wright 1938). Although the dynamics of $\mathbf{L}$ (Equation 7, also see Supplementary information S1) were derived under drift and recombination in the absence of selection, and hence $N_e = N_E$, the dynamical equations for $\mathbf{L}$ will better approximate reality when an effective population size that incorporates all excess variation in fitness is used. Consequently, we use $N_e$ and $N_E$ to distinguish the effective population sizes that are relevant for stochastic changes in $\mathbf{L}$ and allele frequency, respectively.

There is no easy form for the covariance due to the cumulative unpredictable response to selection and we simply denote it as $\mathcal{U}_m$ (Supplementary information S2). While the drift terms will be independent across replicates, we also need to assume that $\underset{U}{\Delta}\mathbf{p}_m$ are independent across replicates for the covariance between replicates to be zero. An obvious source of nonindependence would be if replicates are not initiated from individuals independently generated from Generation 0 individuals (i.e. $t_m \neq 1$ for at least one $m$). If this were the case, the $\Delta\mathbf{L}$ will be dependent across replicates due to the shared changes in $\mathbf{L}$ from generation 0 to the generation from which the replicates are derived. This could be minimized by initializing replicates as early as possible (ideally at $t_m = 1$).

## Inference outline

In the previous section, the mean average effects, $\bar{\boldsymbol{\alpha}}$, are treated as fixed and expectations and variances are taken over the evolutionary process, including variation in the average effects over time and replicates. When making inferences, we treat the mean average effects as random variables rather than fixed quantities (See Supplementary information S3 and Gianola et al. 2009 for a discussion on what this implies) and derive expectations and (co)variances taken over both the evolutionary process and the distribution of $\bar{\boldsymbol{\alpha}}$. Using the laws of total expectation and (co)variance we obtain:

$$E\big[\Delta\mathbf{p}_m\big] = \mathcal{L}_m\boldsymbol{\mu}_{\bar{\alpha}} \tag{11}$$

and

$$\mathrm{VAR}\big(\Delta\mathbf{p}_m\big) = \mathcal{L}_m\mathbf{V}_{\bar{\alpha}}\mathcal{L}_m + \mathcal{D}_m + \mathcal{U}_m \tag{12}$$

where $\boldsymbol{\mu}_{\bar{\alpha}}$ and $\mathbf{V}_{\bar{\alpha}}$ and are the mean and covariance structure of the mean average effects respectively. Critically, when deconditioning on $\bar{\boldsymbol{\alpha}}$ the cross-replicate covariances become nonzero:

$$\mathrm{COV}(\Delta\mathbf{p}_m, \Delta\mathbf{p}_n^\top) = \mathcal{L}_m\mathbf{V}_{\bar{\alpha}}\mathcal{L}_n \tag{13}$$

where $m$ and $n$ are a pair of replicates.

In Supplementary information S3 we determine permissible models for $\boldsymbol{\mu}_{\bar{\alpha}}^\top$ and $\mathbf{V}_{\bar{\alpha}}$. Of those, we identify the sensible biological model for the mean average effect:

$$\boldsymbol{\mu}_{\bar{\alpha}} = \beta_{\bar{\alpha}}^{(1)}(\mathbf{p}_0 - \mathbf{q}_0) \tag{14}$$

where $\mathbf{p}_0$ is the frequency of the reference alleles at each locus in the base population and $\mathbf{q}_0 = \mathbf{1} - \mathbf{p}_0$. For the variance of the average effects, we identify the model

$$\mathbf{V}_{\bar{\alpha}} = \sigma_{\bar{\alpha}}^2 \mathbf{L}_0^{p_a} \tag{15}$$

although in our analysis of simulated data we set the off-diagonal elements of $\mathbf{L}_0$ to zero in the above equation, such that the variance in average effects is simply a power function of the genetic diversities (Zeng et al. 2018). When $p_{\bar{a}} = 0$ average effects are independent of genetic diversity, and when $p_{\bar{a}} = -1$ average effects scale inversely with genetic diversity. $p_{\bar{a}} = -1$ is a common assumption in many related approaches (e.g. Yang et al. 2011) and under this assumption $\sigma_{\bar{a}}^2$ is the average contribution of a locus to the additive genic variance.

By applying sum of squares theory (Searle 2006, page 355) to Equation 2 we can obtain the (posterior) expectation of $V_{\bar{A}}(0)$ after averaging over the distribution of average effects (Supplementary information S3 and Gianola et al. 2009):

$$\begin{aligned} E[V_{\bar{A}}(0)] &= E[\bar{\boldsymbol{\alpha}}^\top\mathbf{L}_0\bar{\boldsymbol{\alpha}}] \\ &= Tr(\mathbf{L}_0\mathbf{V}_{\bar{\alpha}}) + \boldsymbol{\mu}_{\bar{\alpha}}^\top\mathbf{L}_0\boldsymbol{\mu}_{\bar{\alpha}} \end{aligned} \tag{16}$$

where we aim to estimate $\boldsymbol{\mu}_{\bar{\alpha}}$ and $\mathbf{V}_{\bar{\alpha}}$ through Equations 11–13 using multiple evolutionary replicates starting from a common base population.

Rather than working with the allele frequency changes directly, we project them on to a new (reduced) basis and denote this new vector of changes as $\Delta\overrightarrow{\mathbf{p}} = \mathbf{P}\Delta\mathbf{p}$ where $\mathbf{P}$ is some projection matrix. We chose a projection that collapses allele frequency changes into the non-null subspace of $\mathbf{L}_0$, since $V_{\bar{A}}(0)$ only depends on this subspace (Supplementary information S4 and Supporting Information in de Los Campos et al. 2015). To do this, let $\mathbf{U_L}$ be the eigenvectors of $\mathbf{L}_0$ with nonzero eigenvalues and then the drift covariance in the reduced subspace is $\mathbf{U_L^\top}(\tilde{\mathbf{L}}_0 \circ \mathbf{M}^{(m)})\mathbf{U_L}$. If we have $\mathbf{U}_2$ and $\mathbf{D}_2$ as the eigenvectors and a diagonal matrix of square-rooted eigenvalues of this matrix, then the projection matrix $\mathbf{P} = \mathbf{D}_2^{-1}\mathbf{U}_2^\top\mathbf{U_L^\top}$ results in projected allele frequency changes that are identically and independently distributed under drift in the reduced subspace only (see Supplementary information S4).

The mean and covariance of the projected allele frequency changes due to predictable selection are:

$$E[\Delta\overrightarrow{\mathbf{p}}_m] = \mathbf{P}_m\mathcal{L}_m\boldsymbol{\mu}_{\bar{\alpha}} \tag{17}$$

and

$$\mathrm{COV}(\Delta\overrightarrow{\mathbf{p}}_m, \Delta\overrightarrow{\mathbf{p}}_n) = \mathbf{P}_m\mathcal{L}_m\mathbf{V}_{\bar{\alpha}}\mathcal{L}_n\mathbf{P}_n^\top \tag{18}$$

With $p_{\bar{a}}$ known, the model is a linear mixed model with covariance structure due to the predictable response to selection being proportional (by a factor $\sigma_{\bar{a}}^2$ that is to be estimated) to

$$\overrightarrow{\mathbf{V}}_{m,n} \propto \mathbf{P}\mathcal{L}_m\mathbf{L}_0^{p_{\bar{a}}}\mathcal{L}_n\mathbf{P}^\top \tag{19}$$

between replicates $m$ and $n$. When $\mathcal{L}^{(m)} = \mathcal{L}^{(n)}$ for all $n$ and $m$, then this can more easily be fitted by incorporating locus as a random effect with the above covariance structure. The vector of expected values (shown for replicate $m$) is also proportional to

$$\overrightarrow{\boldsymbol{\mu}}_m \propto \mathbf{P}\mathcal{L}^{(m)}(\mathbf{p}_0 - \mathbf{q}_0) \tag{20}$$

with the associated coefficient $\beta_{\bar{\alpha}}^{(1)}$ to be estimated.

We estimate the parameters of the model by treating it as a separable linear mixed model problem (Richards 1961). Conditional on a value of $p_{\bar{a}}$ the model is a linear mixed model and the conditional (restricted) maximum likelihood can be obtained from asreml (Butler et al. 2023). In order to maximize the (unconditional) likelihood we use the R function optim to find the value of $p_{\bar{a}}$ that results in the highest conditional likelihood. Note that the residual variance should equal one if there is no unpredictable response to selection.

Although linear (mixed) models generate unbiased estimates of $\boldsymbol{\mu}_{\bar{a}}$, the quadratic form $\widehat{\boldsymbol{\mu}}_{\bar{a}}^{\top}\mathbf{L}_0\widehat{\boldsymbol{\mu}}_{\bar{a}}$ will be upwardly biased by sampling error in $\beta_{\bar{a}}^{(1)}$. To correct for this bias, we use the inverse Hessian (conditional on the best estimate of $p_{\bar{a}}$) to get an approximate expression for the sampling variance ($S_{\bar{a}}$) of $\beta_{\bar{a}}^{(1)}$, and use this in order to get an improved estimate of the quadratic form as follows (see Supplementary information S5 for a general derivation):

$$\widehat{\boldsymbol{\mu}_{\bar{a}}^{\top}\mathbf{L}_0\boldsymbol{\mu}_{\bar{a}}} = \widehat{\boldsymbol{\mu}}_{\bar{a}}^{\top}\mathbf{L}_0\widehat{\boldsymbol{\mu}}_{\bar{a}} - \mathbf{X}^{\top}\mathbf{L}_0\mathbf{X}S_{\bar{a}} \qquad (21)$$

where $\mathbf{X}$ is the fixed effect design matrix.

In addition, allele frequency changes will rarely be known without error and will most likely be estimated using a pool-seq approach. In such cases, it is necessary to add an additional covariance structure into the model that accommodates the estimation error. In Supplementary information S6 we derive the covariance structure for projected allele-frequency change (in replicate $m$) as:

$$\vec{\mathcal{E}}_m = 2\sigma_o^2\mathbf{P}\left(\mathcal{L}_{t_m,m}\circ\mathbf{Q}_{t_m,m} + \mathcal{L}_{\tau_m,m}\circ\mathbf{Q}_{\tau_m,m}\right)\mathbf{P}^{\top} \qquad (22)$$

where element $ij$ of a $\mathbf{Q}$ matrix is the number of reads that span site $i$ and $j$ divided by the product of the coverages at the two sites. When $i = j$, $Q_{i,j}$ is simply the inverse of the coverage. When individuals have a constant probability of being sampled, $\sigma_o^2$ is expected to be one, but we treat it is a free parameter in order to capture any overdispersion (McCullagh and Nelder 1989).

## Comparison with Buffalo and Coop (2019)

To connect our work with Buffalo and Coop (2019) (henceforth "B&C") it will be useful to express the covariance matrix $\mathbf{L}$ in terms of a diagonal matrix of standard deviations, $\mathbf{B}$, (half the square-root of the genetic diversities under random mating) and the correlation matrix, $\mathbf{R}$, such that $\mathbf{L} = \mathbf{BRB}$. We can then split the response to selection in generation $t$ into two parts:

$$\begin{aligned}\mathbf{L}_t\boldsymbol{\alpha}_t &= \mathbf{B}_t\mathbf{B}_t\boldsymbol{\alpha}_t + \mathbf{B}_t(\mathbf{R}_t - \mathbf{I})\mathbf{B}_t\boldsymbol{\alpha}_t \\ &= \underset{S}{\Delta}\mathbf{p}_t + \underset{L}{\Delta}\mathbf{p}_t\end{aligned} \qquad (23)$$

where the first term is due to direct selection at the loci, and the second term is due to linkage-disequilibria with other selected loci. It will also be useful to distinguish the additive *genetic* variance ($V_A(t)$) from the additive *genic* variance ($V_a(t)$) in generation $t$:

$$V_A(t) = \boldsymbol{\alpha}_t\mathbf{L}_t\boldsymbol{\alpha}_t^{\top} \qquad (24)$$

versus

$$V_a(t) = \boldsymbol{\alpha}_t\mathbf{B}_t\mathbf{B}_t\boldsymbol{\alpha}_t^{\top} \qquad (25)$$

with the distinction being that the additive *genetic* variance ($V_A(t)$) captures the contributions of both genetic diversities at individual loci and the linkage disequilibria between pairs of loci, while the additive *genic* variance ($V_a(t)$) captures the contributions of genetic diversities only.

We can also think about the additive genetic/genic covariance in fitness between time points $t$ and $\tau$ for a population with genetic structure equal to that in generation $t$:

$$C_A(\tau \to t) = \boldsymbol{\alpha}_t\mathbf{L}_t\boldsymbol{\alpha}_\tau^{\top} \qquad (26)$$

and

$$C_a(\tau \to t) = \boldsymbol{\alpha}_t\mathbf{B}_t\mathbf{B}_t\boldsymbol{\alpha}_\tau^{\top} \qquad (27)$$

These will differ from $V_A(t)$ or $V_a(t)$ when the average effects at generation $\tau$ are different from those at generation $t$.

In the theory section above, we assumed $\mathbf{L}_t$ and $\boldsymbol{\alpha}_t$ were known and so the change due to both direct and linked selection are fixed quantities, as is $V_A(t)$. In contrast, B&C condition on $\mathbf{B}_t$ and $\boldsymbol{\alpha}_t$, and treat $\mathbf{R}_t$ as a random variable. Consequently, in B&C, the change due to direct selection and $V_a(t)$ are fixed, as in our approach, since genetic diversities and average effects are known. However, the change due to indirect selection and $V_A(t)$ is random since the linkage-disequilibrium is unknown. In our inference section we acknowledge that $\boldsymbol{\alpha}$ cannot be directly observed, but develop a model for the distribution of $\alpha$'s, where estimates of $V_A(t)$ can be made after marginalizing $\boldsymbol{\alpha}$. Similarly, in the approach of B&C the $\boldsymbol{\alpha}$'s and the elements of $\mathbf{B}$ for selected sites are also not directly observed, but they make a number of strong assumptions that effectively allow the joint distribution of the average effects and selected site diversities to be marginalized.

In Supplementary information S7 we work through the derivation of B&C using our own notation and retaining a full multilocus treatment. For easy comparison with the present work, here we summarize both the explicit and implicit assumptions underlying the general approach of B&C:

- A) Allele frequency changes are only measured over a single generation.
- B) There is no direct selection on the loci for which allele frequency change is measured.
- C) The reference allele at a neutral locus is chosen arbitrarily.
- D) The signed linkage-disequilibrium between selected sites is zero, which precludes processes such as Hill-Robertson interference. Under this assumption $V_A = V_a$.
- E) Changes in $\mathbf{R}$ are due to recombination alone—selection and drift are absent.
- F) Nongametic-phase linkage disequilibrium is absent.
- G) There is no relationship between the additive genic variation at a selected site and its LD with neutral sites, as measured through $R_{t,ij}R_{\tau,ij}$ where $i$ is a neutral locus and $j$ a selected locus. Note that since genetic diversity determines the additive genic variation and LD (even measured as a correlation) is constrained by the genetic diversities at the two loci, this is unlikely to be met (Sved and Hill 2018).
- H) The average effects are constant, i.e. $\boldsymbol{\alpha}_t = \boldsymbol{\alpha}_\tau$ if estimates are to be interpreted as the additive genic variance (or additive genetic variance, if Assumption D is met) in generation $\tau$. If they are not constant, it is the additive genic/genetic covariance between generations $t$ and $\tau$ that is measured.
- I) The initial expected LD-structure between selected and neutral loci, $E[R_{t,ij}^2]$, is approximated from an expression for $E[L_{t,ij}^2]$, which is itself derived under mutation-

recombination-drift balance (i.e. no selection). This assumption is partly relaxed in the Appendix by assuming the expected LD-structure between selected and neutral loci is equal to the observed LD-structure between all loci, not distinguishing between selected and neutral loci (Assumption I-b). However, in many cases the genetic diversity at selected loci will be less than that found at (or assumed for) neutral loci and so $E[R_{t,ij}^2]$ will likely be smaller than assumed (Sved and Hill 2018).

- J) The ratio of genetic diversity in generation $\tau$ to genetic diversity in generation $t$ ($\phi_{t,\tau}$), is constant across all selected loci. Under this assumption, dividing the estimate of $V_a(\tau)$ by $\phi_{t,\tau}$ gives an estimate of $V_a(t)$.
- K) The recombination rate between two sites $g$ base pairs apart is given by $(1 - e^{-2g\bar{r}})/2$ (i.e. Haldane's 1919 mapping function) where $\bar{r}$ is the average crossover rate per site per generation.
- L) Neutral and selected loci are distributed uniformly and independently across the genome such that the distance between them has a triangular distribution.
- M) Since the genetic diversity at selected sites is not measured, it is assumed that $\phi_{t,\tau}$ is equal to the average ratio of genetic diversity in generation $\tau$ to genetic diversity in generation $t$ across all neutral loci.

When applying the theory to data, the method of B&C requires two additional assumptions:

- Ib) The average LD between selected and neutral sites is equal to the average LD observed between all sites: see Assumption I) above.
- N) The (co)variance in allele frequency changes divided through by the average genetic diversity is a good approximation for the (co)variance of weighted allele frequency changes where the weights are the inverse of the square root of the genetic diversities.
- O) If allele frequencies are estimated from a sample, then the estimation error is binomially distributed.

### Assumptions retained in our approach

Our approach works with diploid populations throughout, and assumes nonoverlapping generations. While deriving our main theoretical result (Equation 4), we further assume an absence of meiotic drive and mutation. When extending our approach to practical situations and in our inference approach, we relax most of the assumptions made by B&C, but not all.

- Assumption E is only partly relaxed—the change in **R** does not assume a lack of drift but it does assume a lack of selection.
- We also make Assumption H if we choose to interpret $V_{\bar{A}}(0)$ as as an additive genetic variance (i.e. $V_A(0)$) rather than an additive genetic covariance over replicate/time-points.
- Assumptions E & H result in the unpredictable response to selection being zero. While we do not make this assumption, we do assume that the unpredictable responses to selection are independent across replicates. However, when applying the theory to data we assume that the within-replicate (residual) (co)variances are driven by drift only (rather than an unpredictable response to selection) and that the drift (co)variances are equal to their expected values (conditional on $\mathbf{L}_0$). In reality, the within-replicate covariances are likely larger, due to the unpredictable response to selection. This assumption might not be too severe if the unpredictable

response to selection results in within-replicate (co)variances that are proportional to those under pure drift, although the residual variance may be higher than expected.

- In the absence of a recombination map, Assumption K may also be applied in our method.
- If $\mathbf{L}_0$ cannot be partitioned into gametic-phase and nongametic-phase contributions, assumption F might also be made by substituting $\mathbf{L}_0$ for $\mathbf{L}_0'$ in $\tilde{\mathbf{L}}_0 \circ \mathbf{N}$ and $\tilde{\mathbf{L}}_0 \circ \mathbf{M}$.
- By relaxing some of the more extreme assumptions of B&C we are instead forced into making assumptions about the mean and covariance structure of the average effects of projected loci when making inferences.

## Multilocus simulations

To validate our method and inference approach, we performed multilocus simulations using msprime (version 1.2.0) (Kelleher et al. 2016) and SLiM (version 4.2.2) (Haller and Messer 2023). The code for the simulations and the downstream analyses performed in R (version 4.3.3) is available on GitHub (https://github.com/manas-ga/Va_simulations) and includes the R library, vw. We first describe the structure of the simulations in general terms and then discuss the specifics of our parameter choices in the context of the scaling used (see "Simulation parameters" below). While these simulations do incorporate dominance effects, it is important to acknowledge that they do not include epistatic interactions between loci. They also assume random mating and nonoverlapping generations. Furthermore, it is also important to note that genetic variation in fitness in our simulations was maintained, primarily, by drift-recombination-mutation-selection equilibrium, and our simulations do not capture other mechanisms of maintenance of genetic variation for fitness such as balancing selection or frequency-dependent selection.

We simulated a single, contiguous 1 million base-pair long genomic region. Our simulations had two distinct phases: the first phase simulated the history of an ancestral population and the second phase simulated a typical evolve and resequence experiment with independent replicate experimental populations derived from a base population drawn from the ancestral population.

### Model for fitness

We used an additive model across loci for log absolute fitness, $\log(W)$, as in Buffalo and Coop (2019). This choice was motivated by the fact that the variance in $\log(W)$ is approximately equal to the variance in relative fitness (see Appendix 1 in Lynch and Walsh 1998). Across-locus additivity implies that the genotypic value of log fitness for individual $k$ is equal to $Y_k = \sum_{i=1}^{n_L} y_{k,i}$, where $y_{k,i}$ represents the genotypic contribution made by locus $i$ to the log absolute fitness of individual $k$. Within loci, we employed the classical quantitative genetic fitness scheme (pp. 67 in Lynch and Walsh 1998) adapted for *proportions* ($c_{k,i}$'s) (as opposed to counts) of reference alleles within individuals:

$$y_{k,i} = \begin{cases} 0 & \text{if } c_{k,i} = 0 \\ (1 + \kappa_i)\eta_i/2 & \text{if } c_{k,i} = 0.5 \\ \eta_i & \text{if } c_{k,i} = 1 \end{cases} \tag{28}$$

where $\kappa$ determines the degree of dominance. Note that $\kappa = 0$ implies additivity and $\kappa = -1$ complete recessivity. The average effect for log absolute fitness at locus $i$ is then given by $\eta_i^{(a)} = \eta_i[1 + \kappa_i(q_i - p_i)]$ (see Equation 4.10b in Lynch and Walsh 1998). The average effects for (relative) fitness ($\alpha$'s) are well approximated by the $\eta^{(a)}$'s when they are small in magnitude, but higher-order

approximations are required when the $\eta^{(a)}$'s are large (see Supplementary information S8). To obtain the log fitness of each individual we added a noise term drawn from a standard normal distribution (mean = 0, variance = 1) to the genotypic value. The absolute fitness of each individual was then obtained by exponentiating the individual's log fitness.

We sampled the $\eta$'s from a distribution comprising a weighted mixture of three distributions: (1) a point mass at $\eta = 0$ representing neutral mutations, (2) a reflected gamma distribution with shape = 0.3 and scale = $\eta_{scale}$ (typically 0.066, unless specified otherwise) representing deleterious mutations, and (3) a gamma distribution with shape = 0.3 and scale = $\eta_{scale}$ (typically 0.066, unless specified otherwise) representing beneficial mutations. Thus, for both deleterious and beneficial mutations, the mean absolute $\eta$ (i.e. $E[|\eta|]$) (scale × shape) was, typically, 0.02. The ratio of the frequency of beneficial to deleterious mutations, and $\eta_{scale}$ varied among simulations. Below, we refer to the distribution of non-neutral $\eta$'s (i.e. omitting mixture component (1) described above) as the "distribution of fitness effects" (DFE), but note that this is in fact the distribution of nonzero effects on log fitness.

### Phase 1 ("history phase"): simulating the history of an ancestral population

We first used a neutral coalescent simulation implemented in msprime (Kelleher et al. 2016) to construct genealogies for 2,500 diploid genomes (i.e. $N_e = 2{,}500$). To initialize a (nonequilibrium) set of selected loci, we then simulated mutations at a rate $\mu_{msp}$, using the pyslim package to attach fitness effects ($\eta$) drawn randomly from the non-neutral part of the distribution described above. Since derived mutations are rare, and the DFE is predominantly deleterious, this will generate a positive relationship between the reference allele frequency and the fitness effects $\eta$'s (and therefore the $\alpha$'s, i.e. the average effects for relative fitness), such that $\beta_{\bar{a}}^{(1)} > 0$ but there should be no relationship between genetic diversities and the $\eta$'s (i.e. $p_{\bar{a}} = 0$). We implemented the fitness model described above by using SLiM's (Haller and Messer 2023) default recipe for polygenic selection (in the Wright-Fisher mode). SLiM code for implementing dominance effects in a quantitative genetic framework was adapted from Schaal et al. (2022). To reach mutation-selection-drift balance, we then let this population of 2,500 individuals evolve forward in time with selection for 25,000 generations. Non-neutral mutations drawn from the same DFE were allowed to occur in this period at a rate given by $\mu_{SLiM}$. At generation 25,000, as the alleles reach mutation-selection-drift balance, we expect $\beta_{\bar{a}}^{(1)}$ to have become more positive and $p_{\bar{a}}$ to have become more negative, better reflecting a real population undergoing selection.

In generation 25,000, we sampled $N_0$ diploid individuals (typically 1,000, unless specified otherwise) from this population, which then go on to become the base population in the next phase of the simulation. At this stage, we generated complete genomes for the $N_0$ individuals in the base population by using pyslim to add neutral mutations to the tree sequence recorded so far. To obtain our target number of loci, $n_L$ (typically 65,000, unless specified otherwise), we set the neutral mutation rate to be $(n_L - n_{L_s})/\gamma$ where $n_{L_s}$ is the number of non-neutral segregating sites already present and $\gamma$ is the total branch length of the recorded tree sequence. We recorded the phased genotype of each parent at each locus, allowing us to construct $\mathbf{L}_0$ and $\mathbf{L}_0'$.

### Phase 2 ("experiment phase"): simulating an evolve and resequence experiment

In the second phase of our simulations, again implemented in SLiM, we first allowed the base population to undergo one round

of reproduction without selection to establish replicate experimental populations (typically 1,000 individuals in each of 10 replicates, unless specified otherwise). Next, we allowed each of these populations to evolve forward in time (typically three generations, unless specified otherwise) with selection as in the history phase. Since our goal was to estimate $V_A$ in the base population, we restricted our analyses only to the set of loci segregating in the base population. Any new mutations in subsequent generations would, in all likelihood, occur at loci outside this set. Therefore, we did not simulate new mutations during the experiment phase. For each of the independent replicate populations, we recorded the genome-wide vector of allele frequencies in each generation of the experiment.

### Simulation parameters
#### Varying the true levels of $V_A$

We varied the true (i.e. simulated) levels of $V_A$ from ca. 0.01 to 0.1 by varying the number of non-neutral segregating sites ($n_{L_s}$) in the base population using a range of rates of non-neutral mutations in the history phase in both msprime ($\mu_{msp}$) and SLiM ($\mu_{SLiM}$). For example, in simulations with $E[|\eta|] = 0.02$ (i.e. $\eta_{scale} = 0.066$) and no beneficial mutations, we varied $\mu_{msp}$ between $5.56 \times 10^{-9}$ and $5.56 \times 10^{-8}$ and $\mu_{SLiM}$ between $5.56 \times 10^{-7}$ and $5.56 \times 10^{-6}$, which resulted in $n_{L_s}$ varying between ca. 10,000 and 64,000 in the base population (for other scenarios see Table 1). We set $\mu_{msp}$ to be an order of magnitude smaller than $\mu_{SLiM}$ in most simulations because otherwise all individuals would have had a fitness of zero at the end of the coalescent part of the history phase. This extreme mutation load arises because deleterious alleles, having previously evolved under neutrality, would segregate at fairly high frequencies. However, the choice of $\mu_{msp}$ is unlikely to significantly affect the composition of the base population in most simulations, since the non-neutral genetic diversity of the base population of the experiment phase was primarily determined by the drift-recombination-mutation-selection equilibrium reached over the course of the 25,000 generation long forward simulation, and therefore primarily dependent on $\mu_{SLiM}$.

#### Number of segregating sites ($n_L$)

The requirement for handling large $n_L \times n_L$ matrices (e.g. $\mathbf{L}_0$, $\mathbf{L}_0'$, $\mathbf{M}$, and $\mathbf{N}$, etc.) imposed an upper bound of around 70,000 on $n_L$ to permit the analysis of simulations in parallel. Under most scenarios the entire target range of simulated $V_A$ (i.e. approximately 0.01 to 0.1) could be achieved without $n_{L_s}$ exceeding 65,000. Therefore, in general, we set $n_L$ to 65,000 (but see "Simplified simulations" below). However, under a small minority of scenarios—for example, at higher map lengths in the history phase (Fig. 5b) or under strong dominance (Fig. 3)—generating a larger $V_A$ required significantly more than 65,000 non-neutral segregating sites. Consequently, in such cases, we had to discard simulations in which $n_{L_s}$ was greater than 65,000 which generally happened at the higher mutation rates and, therefore, the higher levels of $V_A$.

#### Map length of the simulated region

There are two recombination-related parameters likely to influence the performance of our method: (1) the $V_A$ per map length, and (2) the density of segregating sites per unit map length. Since our simulations were scaled down in an important way— namely, they assumed that $V_A$ was limited to only about 65,000 segregating sites as opposed to millions of sites in real populations—it was not possible to simultaneously set parameters (1) and (2) close to their realistic values. Therefore, we varied the

effective map length from 0.001 to 2 morgans which spans scenarios where either the total map length of the simulated region is comparable to a typical *D. melanogaster* autosome (∼0.5 morgan) or the density of segregating sites is comparable (∼0.065 morgan). Exact settings are detailed in the "Reference parameters" section below. The typical *D. melanogaster* map length was determined from an effective crossover rate of $10^{-8}$ (Comeron et al. 2012; Wang et al. 2023) and a 50 Mb chromosome. The typical site-density was determined under the assumption that there are one million segregating sites per morgan, which is reasonably consistent with a recent evolve-and-resequence study employing *D. melanogaster* (Bitter et al. 2024: 827,200 and 893,465 sites on Chromosome 2 and 3 respectively).

Since we only simulated a population of 2,500 individuals in the history phase, which is ∼500 times smaller than the estimate of $N_e$ in *D. melanogaster* (Campos et al. 2013; Campos and Charlesworth 2019), we scaled our map length upwards by a factor of 500 to achieve the target effective map length described above (Campos and Charlesworth 2019 but see Dabi and Schrider 2025; Ferrari et al. 2025 for potential issues). On the other hand, since the experiment phase simulates a realistic evolve-and-resequence experiment with $N_e \approx 1,000$, this scaling was not applied to map lengths in the experiment phase (see "Reference parameters" below). In what follows, the map lengths implemented in the simulations are reported rather than the target effective map-lengths.

### Degree of dominance

In the simulations with dominance effects (i.e. $\kappa \neq 0$) we always modeled deleterious alleles to be recessive. In practical terms, this meant that $\kappa_i$ was set to be positive when $\eta_i$ was positive, and $\kappa_i$ was set to be negative when $\eta_i$ was negative. For simplicity, in a given simulation, we assumed $|\kappa_i|$ to be the same across all loci, although the actual dominance deviation, $d_i = \kappa_i \eta_i$ would be locus-specific. We simulated two different levels of $|\kappa|$: (1) $|\kappa| = 0.5$, and (2) $|\kappa| = 0.9$. These correspond to $h = 0.25$ and $h = 0.05$, respectively, when the heterozygous fitness is $1 - hs$. While $h = 0.25$ is consistent with the theoretical expectation from a fitness landscape based model (Manna et al. 2011), $h = 0.05$, where every deleterious allele is almost completely recessive is probably extreme. Since full simulations with dominance were up to two orders of magnitude slower than their additive counterparts, we switched on dominance effects only in the last 5,000 generations of the history phase (i.e. from generation 20,001) as well as in the experiment phase. Relative to additive simulations, full simulations with dominance also required significantly more segregating sites to generate a given level of $V_A$. To generate sufficient $V_A$ (i.e. between *ca.* 0.01 and 0.1) while keeping $n_{L_s}$ below 67,500, we simulated deleterious mutations from a gamma distribution with a slightly larger mean effect on log fitness (i.e. $E[\eta] = 0.03$) in the history phase and set $|\kappa|$ to either 0.5 or 0.75. To test our method against extreme dominance (i.e. $|\kappa| = 0.9$), we performed additional full simulations in which we simulated the entire history phase under additivity, and switched on dominance effects only in the experiment phase. While the resulting genetic composition was far from the expected equilibrium, deleterious alleles were expected to be segregating at substantially lower frequencies than in the simplified simulations (see below).

### Reference parameters

Our aim was to investigate the sensitivity of our method to the following parameters: (1) map length in the history phase (0.5 morgan, 5 morgans, 50 morgans, 250 morgans), (2) map length in the experiment phase (0.01 morgan, 0.2 morgan, 2 morgans), (3)

number of replicate populations in the experiment phase (3, 5, 10), (4) the population size of each of the replicate populations (100, 500, 1,000), (5) number of generations over which allele frequency changes were recorded in the experiment phase (1, 3, 5), (6) the ratio of the rates of beneficial to deleterious mutations (0%, 0.02 %, 2%), and (7) the mean of the gamma distribution from which non-neutral $\eta$'s were sampled, $E[|\eta|]$ (0.02, 0.03, 0.06), and (8) the degree of dominance, $\kappa$ (0, 0.5, 0.9).

Rather than vary all parameters in a fully factorial design, we selected a reference parameter set and explored the sensitivity of the method by changing each parameter in turn. The reference parameter set was (1) a map length of 0.5 morgan in the history phase (2) a map length of 2 morgans in the experiment phase, (3) 10 replicate populations in the experiment phase, (4) a population size of 1,000 in each of the replicate populations in the experiment phase, (5) 3 generations over which allele frequency changes were recorded in the experiment phase, (6) no beneficial mutations in the history phase, (7) $E[|\eta|] = 0.02$, and (8) complete additivity (i.e. $\kappa = 0$). The reference parameter set was chosen to best reflect evolve-and-resequence experiments using *D. melanogaster*.

Note that we varied $E[|\eta|]$ by using three different values (0.066, 0.09, and 0.2) for the scale of the gamma distribution ($\eta_{scale}$) from which the $\eta$'s for non-neutral mutations were drawn while keeping the shape parameter fixed to 0.3. This meant that the coefficient of variation was fixed to $1/\sqrt{0.3}$.

### Simplified simulations

In addition to the full simulations described above, we performed a set of simplified, proof-of-principle simulations to test the logic of our method. These simulations were different from the full simulations in three ways:

(1) The history phase was highly abbreviated—the forward simulation implemented in SLiM lasted only a single generation without any selection. This meant that the ancestral population —from which replicate experimental population were founded— had evolved entirely under neutrality.

(2) Rather than attach $\eta$'s to the derived alleles generated with msprime, we attached the $\eta$'s randomly with respect to either of the two alleles segregating at the locus. Amendments (1) and (2) generate a scenario where there is no relationship between allele frequency and $\eta$: both $p_a$ and $\beta_a^{(1)}$ are expected to be zero.

(3) Given that the ancestral population had evolved without selection, these simplified simulations had considerably higher genetic diversities compared to the full simulations. This meant that fewer segregating non-neutral sites ($n_{L_s}$) were required to achieve true levels of $V_A$ between 0.01 and 0.1. Therefore, in these simulations, we set $n_L$ to be 3,000, much lower than the 65,000 used for the full simulations. We achieved this by varying $\mu_{msp}$ between $3 \times 10^{-9}$ and $2.35 \times 10^{-8}$. $\mu_{SLiM}$ was set to 0 in these simulations. This resulted in $n_{L_s}$ at the end of the history phase being between 160 and 1,902. We then added neutral mutations using a suitable rate as described above such that $n_L$ was expected to be 3,000.

### Simulating pool-seq

To investigate the consequences of sampling noise around true allele frequencies in real data, we reanalyzed the reference sets for full and simplified simulations with allele frequencies in the experiment phase obtained via simulated pool-seq. Specifically, we investigated the sensitivity of our approach to the pool-seq coverage (i.e. the expected number of reads spanning a given locus) and the degree of overdispersion in the number of individuals sampled. We assumed that the number of reads mapping to individual $k$ followed a Poisson distribution with mean $\lambda_{x,k}$, which in

**Table 3.** We applied B&C's method to our simulations using six different approaches.

| Approach | $\Delta p$ | Average LD between | $Cov(\Delta p_m, \Delta p_n)$ |
|---|---|---|---|
| $B^+Ib^+N^+$ | all sites | all sites | divided throughout by $(\bar{p}_m\bar{q}_m + \bar{p}_n\bar{q}_n)/2$ |
| $B^0Ib^+N^+$ | neutral sites | all sites | divided throughout by $(\bar{p}_m\bar{q}_m + \bar{p}_n\bar{q}_n)/2$ |
| $B^0Ib^0N^+$ | neutral sites | selected & neutral sites | divided throughout by $(\bar{p}_m\bar{q}_m + \bar{p}_n\bar{q}_n)/2$ |
| $B^+Ib^+N^0$ | all sites | all sites | replaced by $Cov(\Delta\overrightarrow{\mathbf{p}_m}, \Delta\overrightarrow{\mathbf{p}_n})$ |
| $B^0Ib^+N^0$ | neutral sites | all sites | replaced by $Cov(\Delta\overrightarrow{\mathbf{p}_m}, \Delta\overrightarrow{\mathbf{p}_n})$ |
| $B^0Ib^0N^0$ | neutral sites | selected & neutral sites | replaced by $Cov(\Delta\overrightarrow{\mathbf{p}_m}, \Delta\overrightarrow{\mathbf{p}_n})$ |

The requirement of Assumptions B, Ib, and N is indicated using superscripts ("+" when required and "0" when not required). We used allele frequency change ($\Delta\mathbf{p}$) at either all segregating sites (Assumption B required) or at neutral segregating sites only (Assumption B not required). We computed average LD using either the LD between all segregating sites (Assumption Ib required) or the LD between selected and neutral sites (Assumption Ib not required). We used either the (co)variance in $\Delta\mathbf{p}$ divided throughout by twice the average genetic diversity (Assumption N required) or the (co)variance in weighted $\Delta\mathbf{p}$ (i.e. $\Delta\overrightarrow{\mathbf{p}}$) where the weights are the square roots of twice the genetic diversity at each site (Assumption N not required).

turn is sampled from a log-normal distribution with parameters $\mu_x$ and $V_x$. Varying $V_x$ allowed us to control the degree of heterogeneity in the probabilities with which individuals are sampled. For a given level of coverage and read length, we determined the expected number of reads mapping to an individual ($E[\lambda_x]$). For a given value of $V_x$ and read-length, using the expression for the mean of a lognormal distribution ($E[\lambda_x] = exp(\mu_x + V_x/2)$), we chose $\mu_x$ to achieve the target coverage: $\mu_x = \log(E[\lambda_x]) - V_x/2$. We mapped reads to the genomes of various individuals in the population by sampling the starting positions for each read from a uniform distribution.

To select the length of the reads in these simulations, we first note that the expected number of segregating sites spanned by a 150 base-pair read in a recent evolve-and-resequence study (Bitter et al. 2024) (827,200 segregating sites on chromosome 2 which is ~50 Mb in length) is approximately 2.5. To achieve this, we fixed the read length to be 800 while analyzing simplified simulations (~3,000 segregating sites) and 37 when analyzing full simulations (~65,000 segregating sites). We used four different combinations of coverage and $V_x$: (1) coverage =100×, $V_x = 0$; (2) coverage =500×, $V_x = 0$; (3) coverage =1,000×, $V_x = 0$; (4) coverage =1,000×, $V_x = \log(2)$. Note that $V_x = 0$ corresponds to a situation where individuals are sampled without overdispersion.

### Comparison with B&C

To compare the precision and the accuracy of our approach to that of B&C, we performed additional simplified and full simulations. Although originally designed for the covariance in allele frequency change between multiple time points, B&C's method can be readily adapted to allele frequency changes recorded over a single generation in multiple independent evolutionary replicates (see Supplementary Equation S59 and Buffalo and Coop 2020). We set simulation parameters to their reference values (see above) with a few modifications. First, to make comparisons of biases clearer, we reduced noise by employing 50 replicate populations, and second, allele frequency change in the experiment phase was recorded over a single generation. We also ran these simulations at four different levels of map

length in the history phase: 0.5 morgan, 5 morgans, 50 morgans, and 100 morgans.

To investigate the consequences of Assumptions B, Ib, and N, we implemented B&C's method using six different approaches. For clarity, we label the six approaches using a code (see Table 3) that indicates with superscripts whether an assumption is required (+) or not required (0) for an approach to work. To test Assumption B, we recorded allele frequency changes at either all segregating sites ($B^+$) or neutral sites only ($B^0$). To test Assumption Ib, we used either the average LD between all sites ($Ib^+$) or the average LD between selected sites with neutral sites ($Ib^0$). And finally, we either divided the (co)variance in allele frequency change throughout by twice the average genetic diversity ($N^+$; analogous to Equation 16 in B&C) or used the (co)variance of weighted allele frequency change, where the weights are the inverse of the square root of twice the genetic diversity ($N^0$).

## Results
### Simplified simulations

We begin by discussing results from the simplified simulations in which we expect there to be no relationship between allele frequencies and $\alpha$'s. Under our reference parameter set (no dominance, map length in history = 0.5 morgan, map length in the experiment = 2 morgans), our method provided precise and unbiased estimates of $V_A$ throughout the simulated range (0.01–0.1) (Fig. 1a). Furthermore, the estimates of $p_{\bar{a}}$ and $\beta_{\bar{a}}^{(1)}$ were centered around 0 as expected (Fig. 1b–c), and the residual variance was marginally above 1 (Fig. 1d). Next, we investigated how our estimates of $V_A$ were affected by changing (1) map length of the genomic region being simulated, (2) population size of each replicate population in the evolve and resequence experiment, (3) number of replicate populations, and (4) number of generations over which allele frequency changes were recorded in the experiment. Estimates of $V_A$ remained unbiased as the map length in the experiment became smaller, although estimates became noisier, particularly at higher values of $V_A$ (Fig. 2a). As expected, estimates also became noisier as the number of individuals, replicate populations, or generations became smaller (Fig. 2b–d). However, estimates appeared upwardly biased when the number of replicate populations or generations was small (Fig. 2c–d). Estimates of $V_A$ from simplified simulations incorporating dominance effects in the fitness model were generally precise, but exhibited a small upward bias when dominance effects were large in magnitude (Fig. 3a). As the degree of dominance increased, the estimates of $p_{\bar{a}}$ and $\beta_{\bar{a}}^{(1)}$ became increasingly more negative and the residual variance became larger (Fig. 3b–d).

### Full simulations

In our full simulations we let the ancestral population evolve forward in time with selection for 25,000 generations before simulating the experiment. In these simulations, we expect the population to be at drift-recombination-mutation-selection equilibrium, such that $p_\alpha$ is negative and $\beta_{\bar{a}}^{(1)}$ is positive. Results from our reference set (no dominance, map length in history = 0.5 morgan, map length in the experiment = 2 morgans) suggest that not only does our method provide precise and unbiased estimates of $V_A$ (Fig. 4a), but it does so by correctly estimating the signs of $p_\alpha$ and $\beta_{\bar{a}}^{(1)}$ (Fig. 4b–c). As before, estimates of $V_A$ became slightly noisier at shorter map lengths during the experiment (Fig. 5a). The quality of our estimates of $V_A$ was not significantly affected by adding beneficial mutations during the history phase (Fig. 5c). On the other hand, our estimates were downwardly biased at larger map lengths in the history phase (Fig. 5b, also see Supplementary Fig. 1) and when non-neutral loci

## Simplified simulations

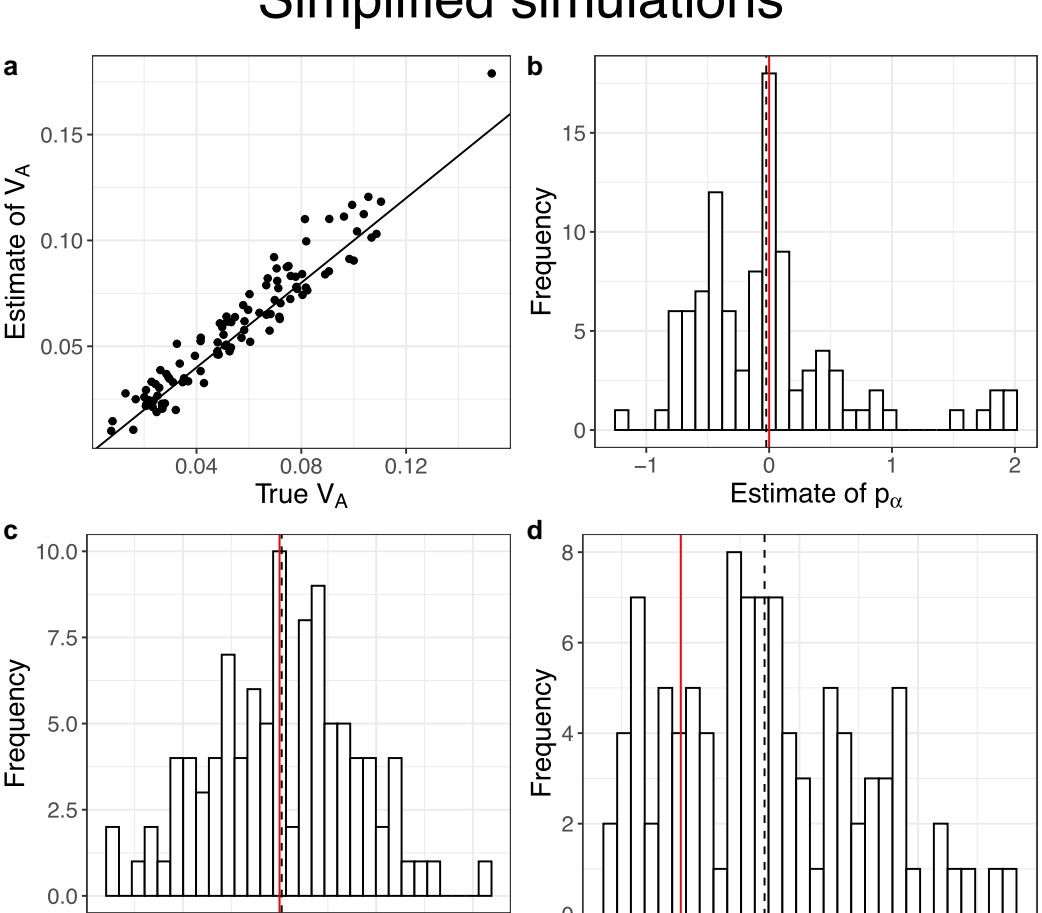

**Fig. 1.** Results of simplified simulations (map length in the history phase = 0.5 morgan, map length in the experiment phase = 2 morgans, number of replicate populations = 10, population size = 1,000, number of generations = 3, the mean of the gamma distribution from which effect sizes for log absolute fitness were sampled for non-neutral mutations ($E[|\eta|]$) = 0.02, and no dominance (i.e. $\kappa = 0$)). a) A scatter plot of estimates of $V_A$ vs true values of $V_A$. The solid black line indicates the 1:1 line. The inference of $V_A$ was obtained by modeling the mean and the (co)variance of the average effects for relative fitness as $\boldsymbol{\mu}_{\bar{a}} = \beta_{\bar{\alpha}}^{(1)}(\mathbf{p}_0 - \mathbf{q}_0)$ and $\mathbf{V}_{\bar{a}} = \sigma_{\bar{a}}^2 \mathbf{L}_0^{p_{\bar{a}}}$, respectively. b)–d) Histograms of the estimates of $p_{\bar{a}}$, $\beta_{\bar{\alpha}}^{(1)}$, and the residual variance, respectively. The vertical solid red lines indicate null expectations (0 for $p_{\bar{a}}$ and $\beta_{\bar{\alpha}}^{(1)}$, and 1 for the residual variance). The black dashed lines indicate the means of the respective distributions.

had on average larger fitness effects ($E[|\eta|]$) (i.e. when $\eta_{scale}$ was larger) (Fig. 5d). The bias at larger $E[|\eta|]$ was likely driven by the loss of additive genic variance at large-effect loci during the experiment (Supplementary Fig. 2a–d).

When we simulated the history phase under additivity and switched on dominance only in the experiment phase, we obtained fairly reliable estimates of $V_A$ (Supplementary Fig. 4). However, switching on dominance in the last 5,000 generations of the history phase resulted in estimates that were significantly upwardly biased, especially when dominance effects were strong (Fig. 3e). However, at the relatively low map length in the history phase (0.5 morgan) used in these simulations, the additive genetic variance for fitness was substantially lower than the additive genic variance (Supplementary Fig. 5). This suggested the build up of unusually high negative linkage disequilibria between recessive deleterious alleles, generating pseudo-overdominance (Ohta and Kimura 1970; Abu-Awad and Waller 2023), a phenomenon typically observed in regions of extremely low recombination (Salson et al. 2025). At higher levels of map length in the history phase (5 morgans), our method provided unbiased estimates of $V_A$ (Fig. 3f).

Finally, we re-analyzed our standard set of full simulations in two different ways (Fig. 5e–f). First, to accommodate cases in which phased genomes, and therefore $\mathbf{L}_0'$, are unavailable, we assumed that $\mathbf{L}_0''$ was 0 and set $\mathbf{L}_0' = \mathbf{L}_0$. In other words, we assumed that $\tilde{\mathbf{L}}_0 = \mathbf{L}_0$. Our analyses suggest that this assumption leads to a slight downward bias in our estimates of $V_A$ (Fig. 5e). Second, to account for instances where $N_E$ in the experiment phase is unknown, we replaced our estimate of $N_E$ in the experiment phase by the number of individuals. Our analyses suggest that this does not affect our estimates of $V_A$ in any way (Fig. 5f), with the faster than expected drift being absorbed by an increased residual variance (Supplementary Fig. 3): estimates of $N_E$ can be obtained by dividing the assumed value of $N_E$ by the estimated residual variance.

### Simulating pool-seq

When we incorporated the expected covariance structure due to pool-seq sampling in our models, the performance of our method was practically unaffected by obtaining allele frequencies in the experiment phase using simulated pool-seq in the simplified simulations (Fig. 6a), although a modest negative bias was observed

## Simplified simulations

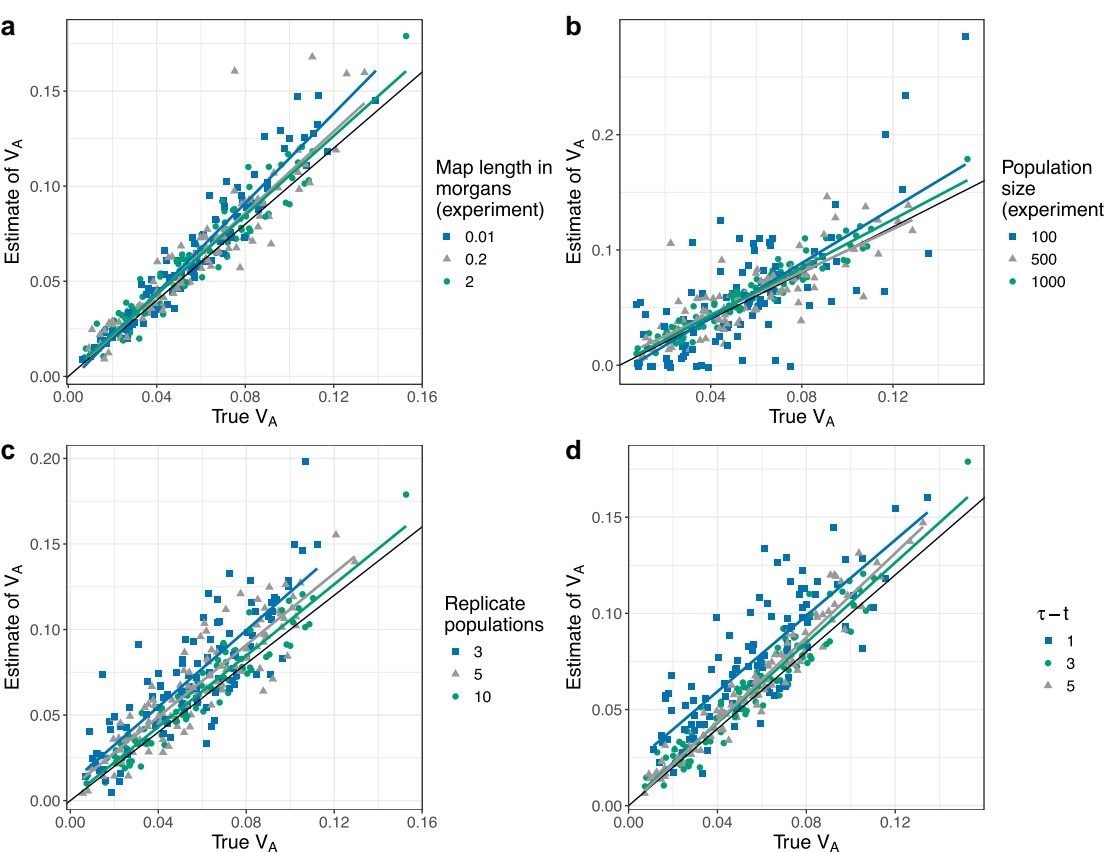

**Fig. 2.** Scatter plots of estimates of $V_A$ vs true values of $V_A$ for simplified simulations at different levels of a) map length in the experiment, b) population size of each replicate population in the evolve and resequence experiment, c) number of replicate populations, and d) number of generations over which allele frequency changes were recorded in the experiment. In each case, other than the parameter to be varied, the other parameters were fixed at their default values (the mean of the gamma distribution from which effect sizes for log absolute fitness were sampled for non-neutral mutations ($E[|\eta|]) = 0.02$, map length in the history phase = 0.5 morgan, map length in the experiment phase = 2 morgans, number of replicate populations = 10, population size = 1,000, number of generations = 3, and no dominance ($\kappa = 0$)). The solid black line indicates the 1:1 line. The colored lines represent regression lines for estimates of $V_A$ vs true values of $V_A$.

at low (100×) coverage in the full simulations (Supplementary Fig. 6). Additionally, both in the simplified and the full simulations, the degree of overdispersion in the number of reads mapping to an individual did not adversely affect our estimates of $V_A$. Along expected lines (see Supplementary information S6), omitting the covariance structure of pool-seq sampling from our models led to estimates of $V_A$ that were significantly downwardly biased, with virtually no $V_A$ detected at 100× coverage (Fig. 6b).

## Comparison with B&C

In both simplified and full simulations, all implementations of B&C's method resulted in a strong upward bias at low recombination rates in the history phase (Fig. 7 and Fig. 8). This was likely a consequence of Assumption G being violated; the correlation between the contribution of a selected site $j$ to $V_a$ and the persistent association of this site with neutral alleles ($\sum_{\mathcal{N}_i} (R_{i_m j_m} R_{i_n j_n})$) was considerably higher than 0 when the map length in the history phase was low (panel h of Fig. 7 and Fig. 8). In contrast, the estimates of $V_A$ provided by our method were unbiased and had far greater precision compared to any of the six implementations of B&C's method in the simplified simulations (Fig. 7g), although there was a small upward bias at low recombination rates in the full simulations (Fig. 8g). Due to the difficulty of distinguishing

selected from unselected sites, B&C suggested that the average LD between all sites could be used instead of the LD between selected and unselected sites (Assumption Ib) and indeed this assumption seems to be well justified (panels d & f in Fig. 7 and Fig. 8 are very similar). However, an inability to distinguish selected from unselected sites would likely mean that B&C's method is applied to allele frequency change data from *all* segregating sites, contravening Assumption B. This results in considerable overestimation of $V_A$ in simplified simulations (Fig. 7a–b) but not in full simulations (Fig. 8a–b). Under all scenarios the bias in the method of B&C can be reduced (but not eliminated) by relaxing Assumption N (compare panels b, d, and f in Fig. 7–Fig. 8 to panels a, c and e). Under assumption N, it is assumed that the (co)variance in allele frequency change weighted by the square root of twice the genetic diversities can be approximated by the (co)variances in allele frequency change divided through by twice the average diversities. However, this approximation is not required.

## Discussion

In this paper we estimate the additive genetic variance for relative fitness ($V_A$) directly from the change in the genetic composition of

# Simplified simulations

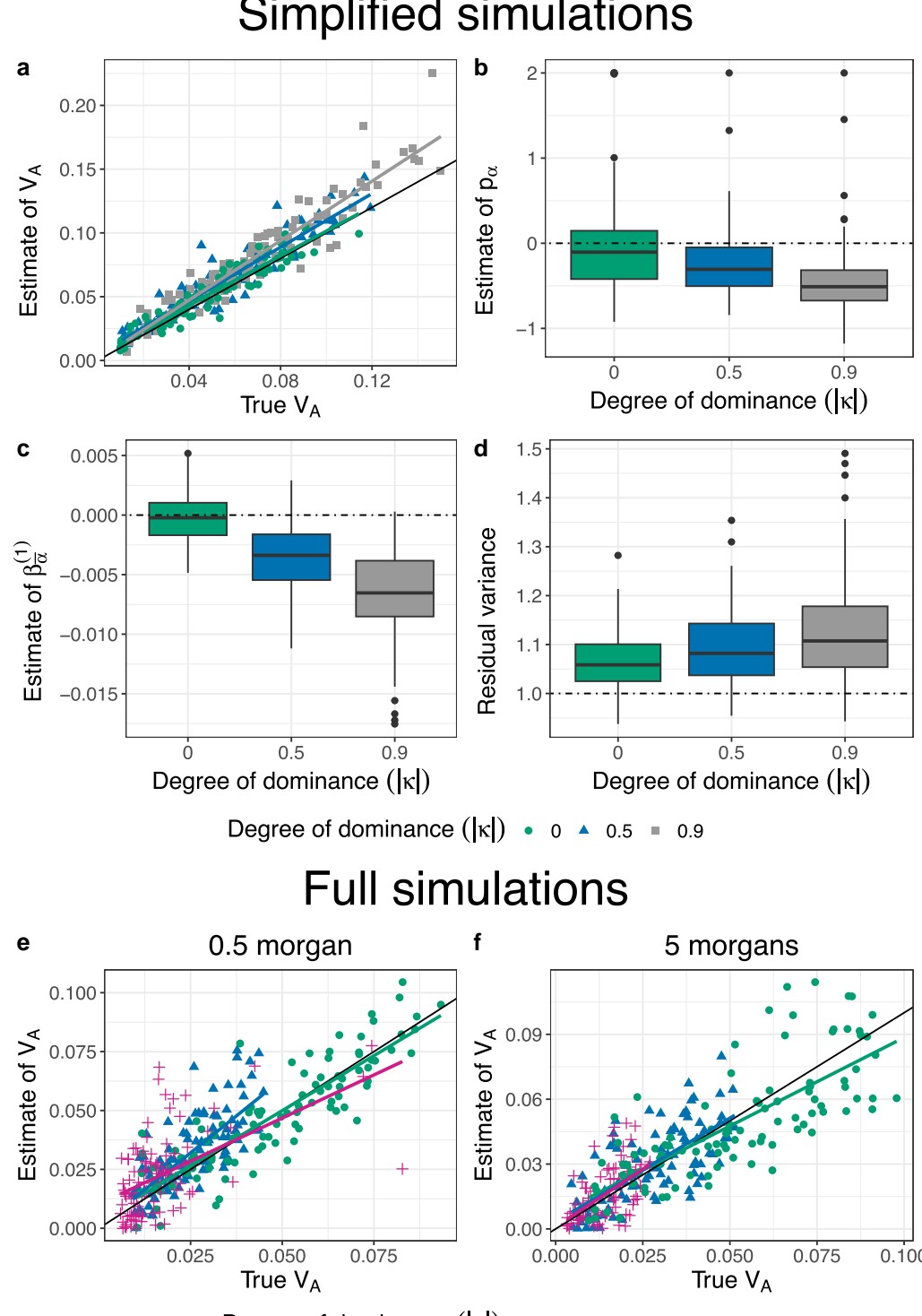

## Full simulations

**Fig. 3.** Results of simulations with map lengths in the experiment phase = 2 morgans, number of replicate populations = 10, population size = 1,000, number of generations = 3, and different degrees of dominance: $\kappa = 0$ (green circles), $\kappa = 0.5$ (blue triangles), $\kappa = 0.75$ (magenta plus signs) and $\kappa = 0.9$ (grey squares). (a–d) are the results from simplified simulations with the map length in the history phase = 0.5 morgan and the mean of the gamma distribution from which effect sizes for log absolute fitness are sampled for non-neutral mutations ($E[|\eta|]$) = 0.02. (a) A scatter plot of estimates of $V_A$ vs true values of $V_A$. The solid black line indicates the 1:1 line. The colored lines represent regression lines for estimates of $V_A$ vs true values of $V_A$. (b–d) Estimates of model parameters $p_{\bar{a}}$, $\beta_{\bar{a}}^{(1)}$, and the residual variance, respectively. (e–f) Scatter plots of estimates of $V_A$ vs true values from full simulations (with $E[|\eta|]$ = 0.03) when the map length in the history phase was either 0.5 morgan (e) or 5 morgans (f).

a population caused by selection. Assuming only the absence of meiotic drive and ignoring mutation, we show that $V_A$ can be conveniently expressed as a function of the genome-wide genetic diversity matrix **L**, and the vector of genome-wide expected allele frequency change due to selection, $E[\Delta\mathbf{p}]$. In our inference approach, we describe how a linear mixed model can be employed

# Full simulations

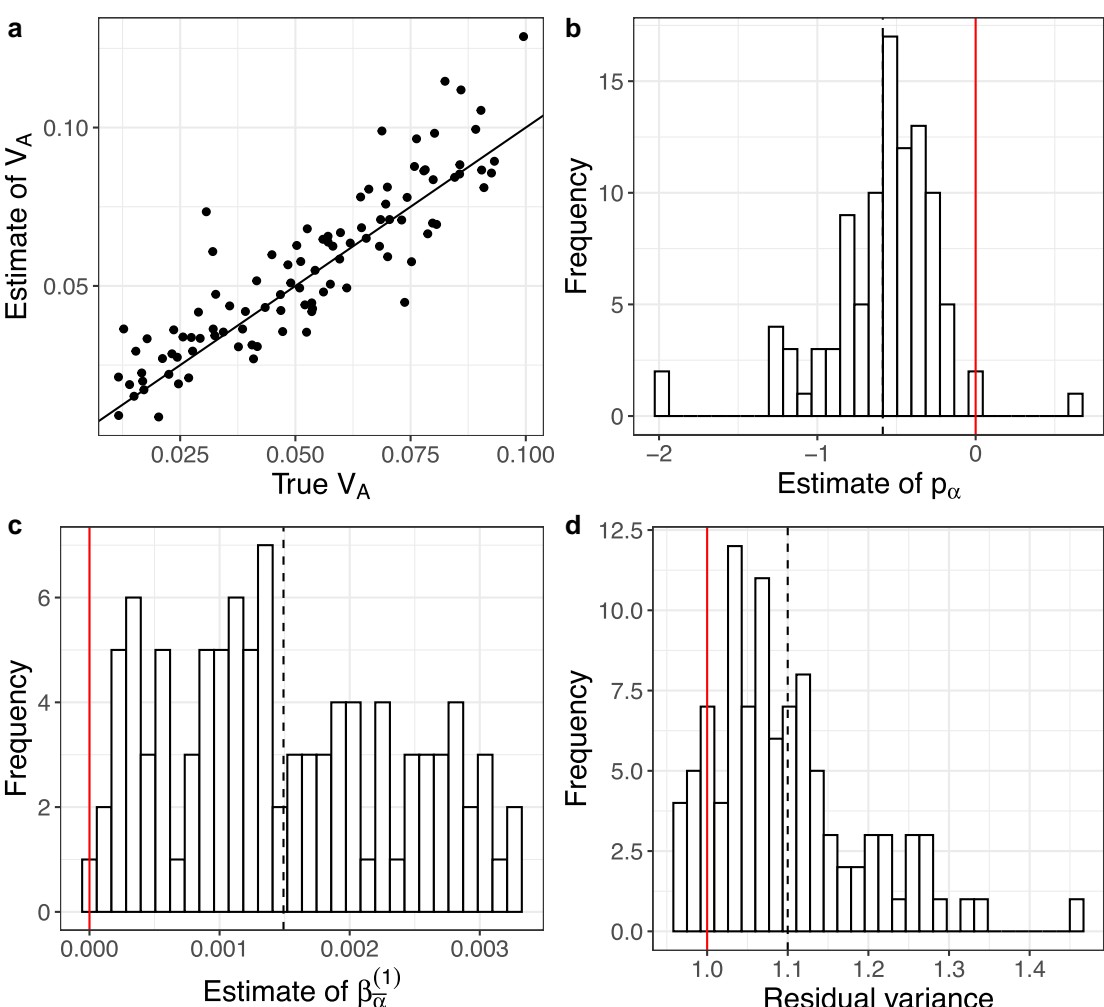

**Fig. 4.** Results of full simulations with a burn-in phase of 25,000 generations (map length in the history phase = 0.5 morgan, map length in the experiment phase = 2 morgans, number of replicate populations = 10, population size = 1,000, number of generations = 3, the mean of the gamma distribution from which effect sizes for log absolute fitness were sampled for non-neutral mutations ($E[|\eta|]$) = 0.02, and no dominance (i.e. $\kappa = 0$)). a) A scatter plot of estimates of $V_A$ vs true values of $V_A$. The solid black line indicates the 1:1 line. The inference of $V_A$ was obtained by modeling the mean and the (co)variance of the average effects for relative fitness as $\boldsymbol{\mu}_{\bar{a}} = \beta_{\bar{a}}^{(1)}(\mathbf{p}_0 - \mathbf{q}_0)$ and $\mathbf{V}_{\bar{a}} = \sigma_{\bar{a}}^2 \mathbf{L}_0^{p_{\bar{a}}}$, respectively. b)–d) Histograms of the estimates of $p_{\bar{a}}$, $\beta_{\bar{a}}^{(1)}$, and the residual variance, respectively. The vertical solid red lines indicate null expectations (0 for $p_{\bar{a}}$ and $\beta_{\bar{a}}^{(1)}$, and 1 for the residual variance). The black dashed lines indicate the means of the respective distributions.

to estimate $E[\Delta\mathbf{p}]$ via independent evolutionary replicates derived from the same base population with a known $\mathbf{L}$—a common feature of evolve-and-resequence studies. Unlike alternative methods (Buffalo and Coop 2019), this allows us to obtain estimates of $V_A$ that are largely robust to the underlying genetic properties of the population and the genetic architecture of fitness. Moreover, the underlying modeling framework not only allows $V_A$ to be estimated, but allows inferences about the relationship between effect sizes and allele frequency.

Although $E[\Delta p]$'s at individual loci cannot be usefully estimated, for our purposes it is sufficient to estimate their distribution as parameterized through the mean vector $\boldsymbol{\mu}_{\bar{a}}$ and the (co)variance matrix $\mathbf{V}_{\bar{a}}$ for the distribution of the average effects for fitness, the $\alpha$'s. Our inference approach uses a low-dimensional (3-parameter), but biologically sensible, model for the means and (co)variances. It is superficially surprising that such a simple model for an $n_L$ (the number of loci) dimensional

distribution can produce accurate results. However, in a typical dataset one expects the number of individuals, $N$, to be far smaller than the number of loci, $n_L$. Therefore, the non-null subspace of $\mathbf{L}$ has only $N$ dimensions, and we can work with allele frequency changes projected into this reduced space. This provides a route to understanding how the distribution of $\alpha$'s can be estimated from selected changes in allele frequency that must be negligible compared to the impact of drift: the projection defines "*chunks*" of genome, and it is the frequency changes in these chunks, rather than individual alleles, that are tracked. Since the aggregate fitness effects of alleles across chunks will be more substantial, they can be more easily detected. Moreover, since these aggregate effects may involve a large number of loci, they will, thanks to the central limit theorem, tend to normality and, conditional on the projection, converge in distribution. Since the projection is defined by $\mathbf{L}$, a model of the $\alpha$'s that conditions on aspects of $\mathbf{L}$ ($p - q$ and $pq$ in our case) is expected to be sufficient.

# Full simulations

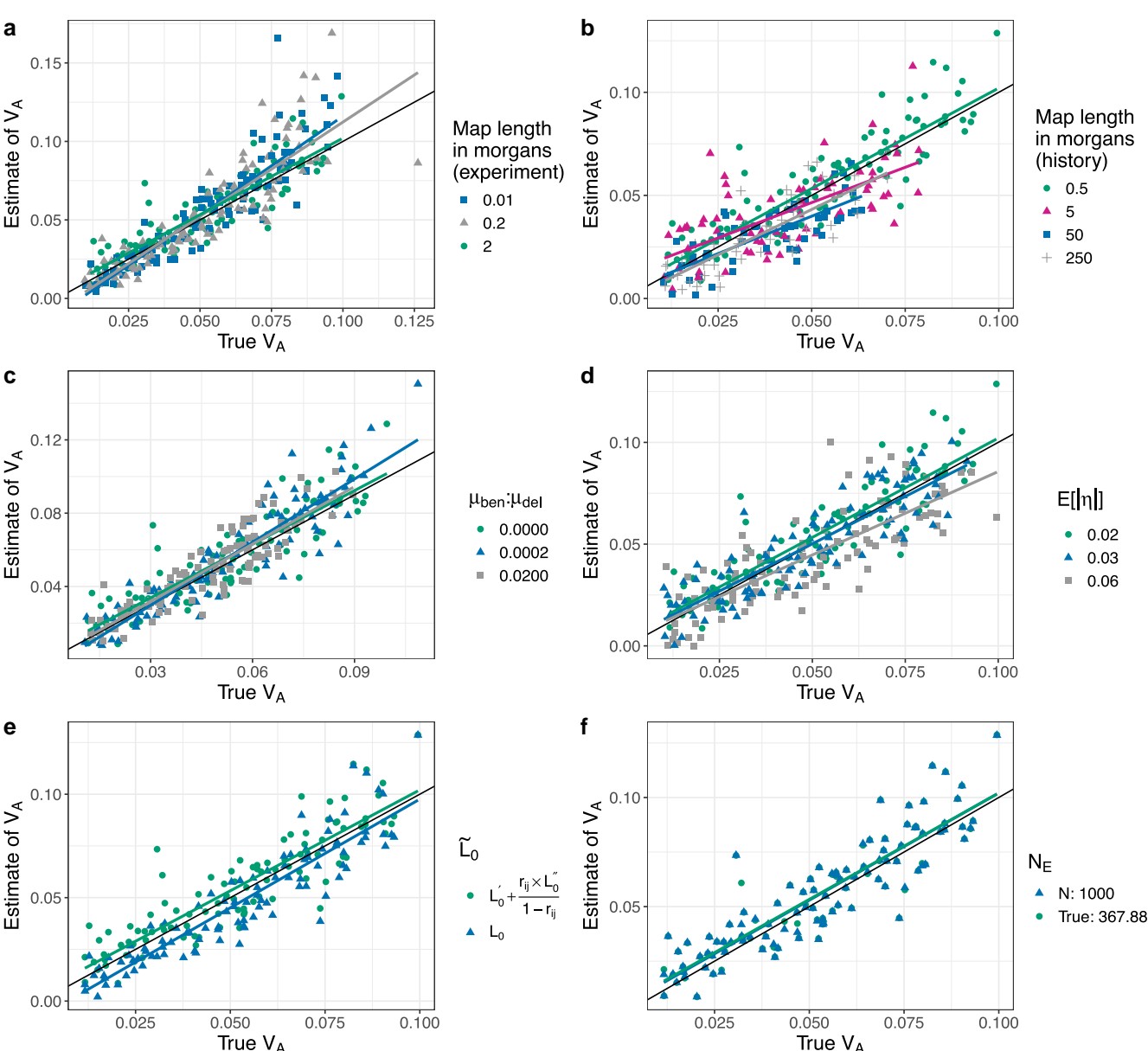

**Fig. 5.** Scatter plots of estimates of $V_A$ vs true values of $V_A$ for full simulations with a burn-in phase of 25,000 generations at different values of a) map length in the experiment phase, b) map length during the history phase, c) the ratio of rates of beneficial and deleterious mutations in the history phase, and d) the mean of the gamma distribution from which effect sizes for log absolute fitness were sampled for non-neutral mutations ($E[|\eta|]$). In each case, other than the parameter to be varied, the other parameters were fixed at their default values (map length in the history phase = 0.5 morgan, map length in the experiment phase = 2 morgans, number of replicate populations = 10, population size = 1,000, number of generations = 3, $E[|\eta|] = 0.02$, and no dominance (i.e. $\kappa = 0$)). The solid black line indicates the 1:1 line. The colored lines represent regression lines for estimates of $V_A$ vs true values of $V_A$. The effect of analyzing the standard set of simulations (Fig. 4) e) without partitioning $\tilde{\mathbf{L}}_0$ into its gametic ($\mathbf{L}_0'$) and nongametic phase ($\mathbf{L}_0''$) components, and (f) using $N_E = N = 1,000$. Note that in (f), a large number of green circles are eclipsed by the blue triangles.

The results of our simulations demonstrate that our approach provides usefully precise and consistent estimates of $V_A$ over a wide range of parameter combinations and experimental designs. In the simplified simulations, in which the relationship between allele frequencies and $\alpha$'s was expected to be absent, the model requires only a single parameter: the variance in average effects, $\sigma_{\bar{a}}^2$. If such a model is assumed, this is a relatively straightforward inference problem and the method of Buffalo and Coop (2019) can give accurate estimates if selected and

neutral sites can be distinguished, and certain patterns of recombination and linkage-disequilibrium hold. Although our approach does not assume such a model, it can accurately infer $V_A$ and also the lack of relationship between allele frequencies and $\alpha$'s, with the two parameters that determine the relationship between allele frequency and $\alpha$ all centered on their null expectations (Fig. 1). However, in reality, independence between allele frequency and $\alpha$ seems implausible at mutation-selection-drift balance, and the elevated contribution of high-diversity loci to

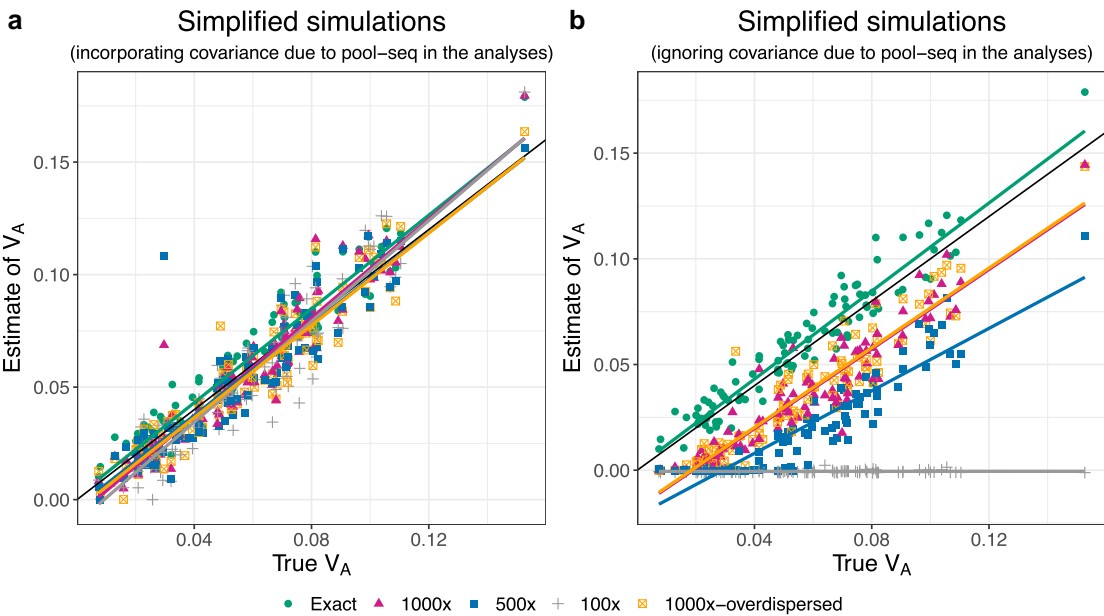

**Fig. 6.** Scatter plots of estimates of $V_A$ vs true values of $V_A$ for simplified simulations (map length in the history phase = 0.5 morgan, map length in the experiment phase = 2 morgans, number of replicate populations = 10, population size = 1,000, number of generations = 3, the mean of the gamma distribution from which effect sizes for log absolute fitness were sampled for non-neutral mutations ($E[|\eta|] = 0.02$), and no dominance (i.e. $\kappa = 0$)) using either exact allele frequencies in the experiment phase (green circles), or allele frequencies in the experiment phase obtained via simulated pool-seq implemented without any overdispersion in the number of reads mapping to an individual, at three different levels of coverage (expected number of reads mapping a segregating site: 1,000× (magenta triangles), 500× (blue squares), and 100× (grey plus symbols)) as well as estimates obtained from simulated pool-seq implemented with overdispersion ($V_x = \log(2)$) in the number of reads mapping to an individual, at 1,000× coverage (orange empty boxes with crosses). Reads were modeled to be 800 base-pairs long. The solid black line indicates the 1:1 line. The colored lines represent regression lines for estimates of $V_A$ vs true values of $V_A$. Estimates of $V_A$ were obtained by either incorporating (a) or omitting (b) the expected covariance structure due to pool-seq sampling in the models.

$V_A$ under the simple, but unrealistic, scenario may result in greater power to detect $V_A$.

At mutation-selection-drift balance a negative relationship is expected between $\alpha$'s and allele frequencies (Charlesworth and Charlesworth 2010). Although it could be argued that the strength of this relationship will be reduced in experimental evolution studies, which generally expose populations to novel environmental stressors (such as a laboratory environment, pathogens (Basu et al. 2024), extreme population densities (Joshi and Mueller 1996) and temperatures (Singh K et al. 2015; Hsu et al. 2024), desiccation (Gibbs et al. 1997), malnutrition (Kawecki et al. 2021), or toxic substances (Xiao et al. 2019; Godinho et al. 2024)) it seems unlikely that no relationship would persist. Even under the more realistic scenario of our full simulations, in which the base population had evolved under selection for 25,000 generations, our approach provided reliable estimates of $V_A$ that were remarkably robust to the details of the distribution of fitness effects, such as the relative frequency of beneficial mutations, or properties of the population such as the recombination rate. Furthermore, it did so by correctly inferring a negative relationship between fitness effects of alleles and genetic diversity, and a positive relationship between fitness effects and $p - q$, as expected under mutation-selection balance.

An important assumption of our method is that any consistent change in the $\alpha$'s is vanishingly small in the time-frame over which allele frequency changes are recorded. However, when fitness-causing alleles act nonadditively, $\alpha$'s depend on allele frequency and so are expected to change as allele frequencies change. In spite of this, in the simplified simulations where allele frequency change was expected to be large, the performance of

the method was minimally affected when allowing dominance at all loci. Moreover, given the form $\alpha_i \approx \eta_i[1 - \kappa_i(p_i - q_i)]$ (where the product $\eta_i\kappa_i$ is always positive in our simulations), the method correctly inferred that the two parameters describing the relationship between the $\alpha$'s and allele frequencies would become increasingly negative with the degree of dominance (Fig. 3). Surprisingly, when we allowed dominance effects in the history phase and so allele frequencies and effects had come to an equilibrium we found that our method overestimated $V_A$ to varying degrees depending on how recessive the deleterious allele was. However, this was not because of a failure of the infinitesimal approximation in the experimental phase, but because substantial negative LD had built up during the history phase and our model of $V_{\bar{a}}$ failed to capture this: the additive genetic variance was roughly half the additive genic variance. It seems that our simulations were resulting in substantial pseudo-overdominance (Abu-Awad and Waller 2023)—a phenomenon characteristic of regions of very low recombination (Salson et al. 2025). When we performed identical simulations at higher recombination rates in the history phase, this problem was not observed.

Another key assumption of our method is that any consistent, selection-induced changes in **L** are negligible, which then allows us to model the expected evolution of **L** due to the action of drift and recombination. Selection-induced changes in **L** may be safely ignored if the genetic architecture of fitness is sufficiently polygenic such that selection coefficients associated with each locus are vanishingly small. It is important to note that this assumption was fully relaxed in all of our simulations, in which selection was free to influence the evolution of **L** along with drift and recombination. Despite this, the performance of our method was virtually

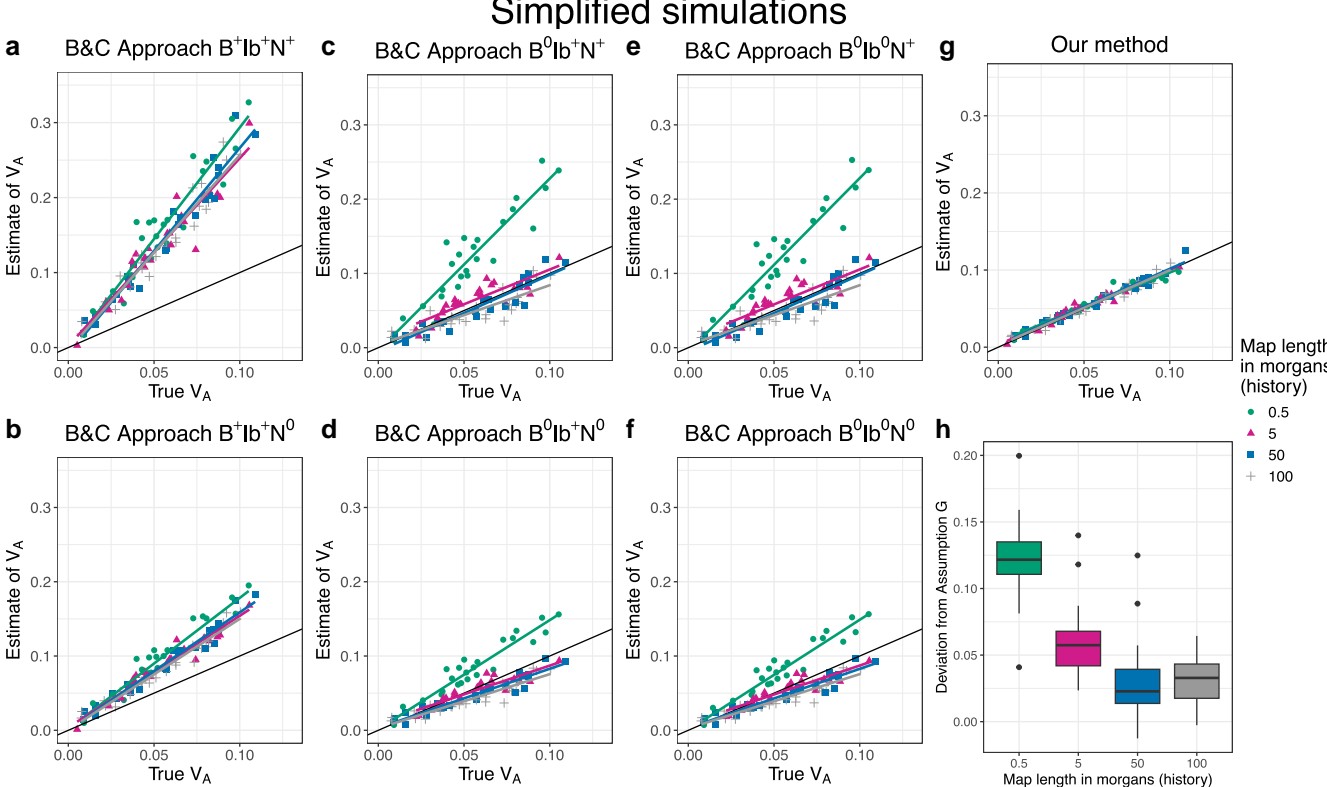

**Fig. 7.** Estimates of $V_A$ obtained using our method (g) and various applications of the B&C Approach (a–f) from simplified simulations with varying map lengths in the history phase: 0.5 morgan (green circles), 5 morgans (magenta triangles), 50 morgans (blue squares), and 100 morgans (grey plus symbols). The solid black line indicates the 1:1 line. Each application of B&C is defined in terms of which three assumptions (B, Ib, N) are required or not (designated with a "+" or "0" superscript, respectively). Assumption B is that allele frequencies are tracked at unselected loci only, Assumption Ib is that the average LD between all sites is equal to the average LD between selected and unselected sites, and Assumption N is that the covariance between projected allele frequency changes is equal to the covariance between allele frequencies changes scaled by the average projection (see Table 3 for details). Panel h is the correlation between the contribution of selected sites to $V_d$ and their persistent associations with neutral alleles (i.e. deviations from Assumption G in B&C).

unaffected, even in the simplified simulations where selection-induced changes in **L** were expected to be relatively large.

While our method generally performs well, some biases were observed. In terms of experimental design, small upward biases are evident when power is low - either because there are few replicates and/or allele frequency change is only calculated over a single generation. In terms of genetic architecture, our approach marginally underestimates $V_A$ when the fitness effects of new non-neutral mutations are large. This downward bias seems to arise because there is a large contribution of rare highly deleterious variants to $V_A$ and these get lost during the experimental phase. Nevertheless, even when 25% of $V_A$ was lost during the course of the experiment the impact on estimates was rather minor. Similarly, modest downward biases were observed when the recombination rate in the history phase was increased, but the source of this bias has been harder to diagnose. Increasing the recombination rate resulted in an increase in the number of segregating selected sites and their genetic diversity, and a steeper relationship between $\alpha$ and genetic diversity, consistent with a reduction in the effects of background selection (Charlesworth et al. 1993) on weakly selected sites (Stephan et al. 1999). While it is not clear why this causes downward bias in the estimates, it is also unclear whether real populations would ever have such extreme genetic architecture where the bulk of the additive genetic variance for fitness is caused by highly deleterious variants

segregating at very low frequencies. In order to make the forward simulations manageable we were working with considerably smaller population and genome sizes than are typical of real populations. It is not clear, however, whether the standard rescaling of mutations rates, recombination rates and selection coefficients to accommodate this downsizing results in genetic architectures that would be typical of larger populations and larger genomes (Dabi and Schrider 2025).

While our method performs well when applied to simulated data, application to real-world data would involve overcoming a number of challenges. First, we require genome-wide allele frequency change data from multiple independent evolutionary replicates—although this should be readily available for most evolve-and-resequence experiments using approaches such as pool-seq. An important point of consideration, therefore, is the minimum pool-seq coverage that the method requires. For our simplified simulations, there was little deterioration in model performance as coverage dropped or overdispersion increased, suggesting modest coverage (100×) should be sufficient if the appropriate covariance structure is used. While this might be surprising, the relevant parameter is probably not coverage *per se*, but the number of reads overlapping at least one segregating site per unit map length, as this determines how accurately the frequency of a "chunk" of genome can be measured. Using a similar argument, Tilk et al. (2019) demonstrated that supplementing pool-seq sampling with haplotype inference tools can result in nearly 500×

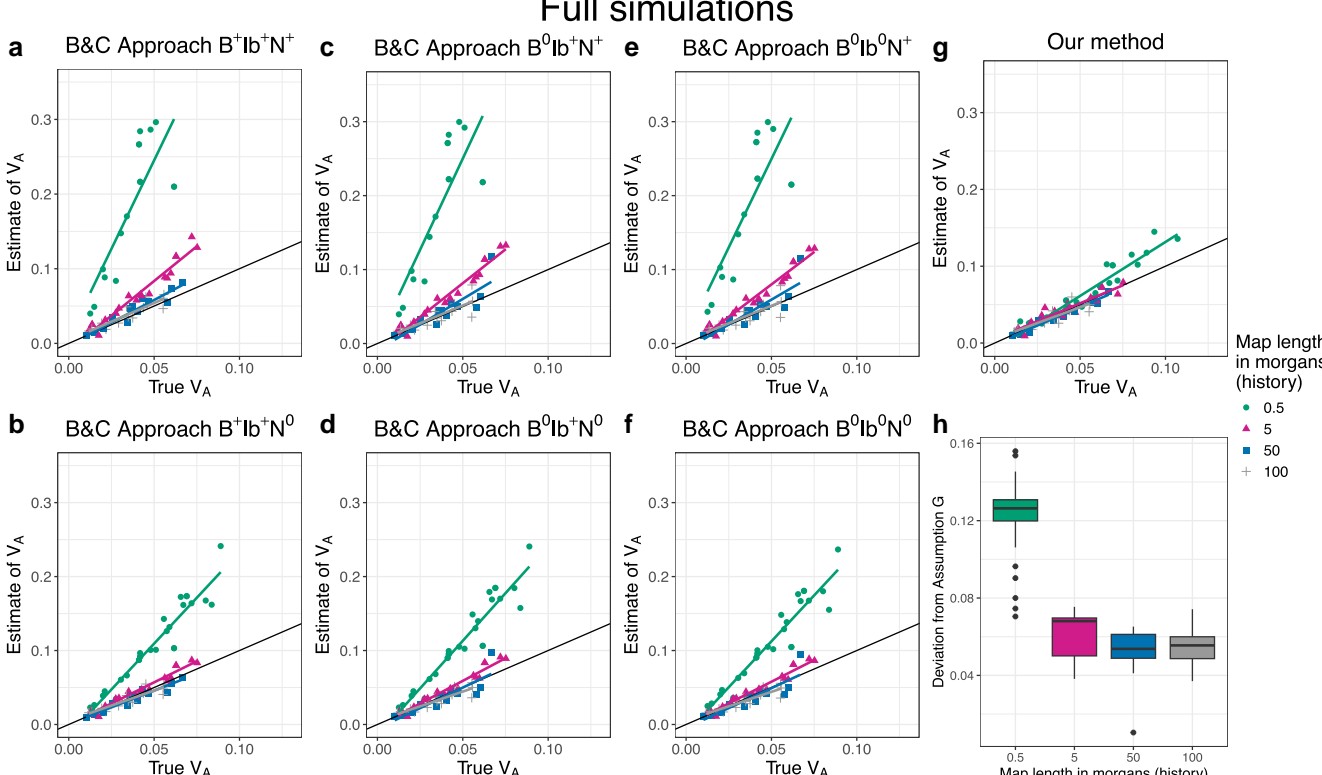

**Fig. 8.** Estimates of $V_A$ obtained using our method (g) and various applications of the B&C Approach (a–f) from full simulations with varying map lengths in the history phase: 0.5 morgan (green circles), 5 morgans (magenta triangles), 50 morgans (blue squares), and 100 morgans (grey plus symbols). The solid black line indicates the 1:1 line. Each application of B&C is defined in terms of which three assumptions (B, Ib, N) are required or not (designated with a "+" or "0" superscript, respectively). Assumption B is that allele frequencies are tracked at unselected loci only, Assumption Ib is that the average LD between all sites is equal to the average LD between selected and unselected sites, and Assumption N is that the covariance between projected allele frequency changes is equal to the covariance between allele frequencies changes scaled by the average projection (see Table 3 for details). Panel h is the correlation between the contribution of selected sites to $V_a$ and their persistent associations with neutral alleles (i.e. deviations from Assumption G in B&C). For clarity of presentation, in (a), (c), and (e), we have restricted the range of the y axis between 0 and 0.35. As a consequence, points having an estimate of $V_A$ above 0.35 have been excluded from the plot for 0.5 morgan (green circles).

"effective" coverage at 10× empirical coverage. Given the ancestral haplotype structure is a core requirement of our method, it may be possible that even greater precision could be achieved by also using such tools. However, this could come with the risk that estimation errors are correlated across replicates and be mistaken for patterns of selection. For our full simulations, a moderate downward bias was observed at 100× coverage. While this may suggest that our method demands relatively high coverage (≥500×), it is worth noting that we only implemented an approximate (i.e. diagonal) covariance structure for pool-seq sampling while analyzing our simulations due to computational limitations. Incorporating the full covariance structure should lead to improved estimates.

Second, we require the genetic diversity matrix **L** in the base population from which the replicates are derived. This is not always the case for evolve-and-resequence studies, in which base populations are often split into replicate baseline populations long before the experiment, or when newer selection regimes are derived mid-experiment (Burke et al. 2010; Singh K et al. 2015; Gupta et al. 2016; Robinson et al. 2023). Furthermore, our method requires that sufficient individuals from the base population are individually sequenced to estimate **L**, or—even better—its gametic phase (**L**′) and nongametic phase (**L**″) components, although the required phasing should become more readily available with long-read sequencing. Third, to predict how **L** evolves, we require an estimate of the recombination probability between

all pairs of segregating sites, such as a recombination map for the population. In reality, recombination maps are likely to have been derived from other populations, in which recombination patterns may differ (Johnston 2024). Although recombination maps can be approximated, for example by using Haldane's mapping function, it is not clear how sensitive the method is to errors in the recombination map. Fourth, our method requires the mean number of generations over which allele frequency changes are calculated, which may be hard to infer with overlapping generations. Since allele-frequency change will be roughly proportional to the number of generations that have elapsed ($n_g$), and $V_A$ is quadratic in allele-frequency change, estimates of $V_A$ might be out by a factor $(n_g/\widehat{n}_g)^2$, where $\widehat{n}_g$ is the assumed number of generations. Fifth, our method uses effective population size to predict how **L** evolves ($N_e$) and to derive expressions for the drift (co)variance ($N_E$). Although reliable estimates of both effective population sizes may be hard to obtain, this is unlikely to be a major issue because: (i) **L**′ decays with a rate roughly proportional to $1 - 1/N_e$ (likely making it insensitive to errors in estimating $N_e$) and (ii) our simulations suggest that using the wrong $N_E$ does not adversely affect our estimates of $V_A$ because the mis-specification is absorbed by the residual variance of the model. Sixth, our simulations minimize the unpredictable response to selection by modeling a constant environment. However, with selection coefficients that vary in time or across replicates, the unpredictable response to selection will be greater. Indeed in outdoor mesocosms of *D.*

*melanogaster*, Bitter et al. (2024) report that allele frequency changes can switch signs over time-points separated by a matter of weeks in spite of exhibiting highly concordant evolution between replicates. It is not clear to what degree this will affect inferences. Finally, selection can often act in different ways in different contexts such as space (Whitlock 2015; Delph 2018), time, and between the two sexes (Schenkel et al. 2018). Our approach captures the effects of selection averaged over all these different contexts. Specifically, if loci have different fitness effects in males and females, we effectively estimate $(V_{A,f} + V_{A,m} + 2COV_{A,mf})/4$, where $V_{A,f}$ and $V_{A,m}$ are the additive genetic variances for relative fitness in females and males respectively, and $COV_{A,mf}$ is the intersexual additive genetic covariance for relative fitness. For these reasons, when applied to replicate populations, our estimates of $V_A$ are perhaps best thought of as the additive genetic *covariance* in fitness between replicates. Although rarely made explicit, estimates from wild systems should be interpreted in the same way: the additive genetic covariance between the environments in which relatives live (Vehviläinen et al. 2008).

Analyzing data from wild populations is likely to entail further challenges. For example, in natural populations, allele frequency change from immigration may be consequential, and without some modification is likely to be mistaken for allele frequency change caused by natural selection (Simon and Coop 2024). Furthermore, replicate populations are unlikely to be available for natural systems—with some exceptions such as Trinidadian guppies (Reznick and Bryga 1996)—although with some modifications our method could be adapted to use allele frequency change data from multiple time points. However, a lack of individual-level sequences in the base population would mean that the estimates of **L** will likely be considerably noisier in natural populations.

While we accept that the data requirements for our method are steep, and the list of caveats appears long, we do think that information about $V_A$ can be successfully leveraged from current evolve and resequence studies. Going forward, we hope this work will inform future evolve and resequence study design, and that estimates of $V_A$ from a wide range of organisms and environments become available. Partitioning $V_A$ into genomic features is an obvious next step, and in the future we hope to go beyond simply knowing the magnitude of $V_A$ and start to understand its underlying causes.

## Data availability

This study does not use any data. The code used for simulations and analyses is available in the following GitHub repository: https://github.com/manas-ga/Va_simulations

Supplemental material available at GENETICS online.

## Acknowledgments

The authors would like to thank Bill Hill, Brian Charlesworth, Peter Keightley, Konrad Lohse, Bruce Walsh, Ben Longdon and Vince Buffalo for their insightful comments, and The Argyle for hosting us at the early stages of this project. Computer simulations and analyses described here were performed on the AC3 computing cluster based at Ashworth Laboratories, and on Eddie, the University of Edinburgh's main high performance computing facility.

## Funding

This work was funded by a Natural Environment Research Council (NERC) grant (reference: NE/W001330/1). For the purpose of open access, the authors have applied a Creative Commons Attribution (CC BY) license to any Author Accepted Manuscript version arising from this submission.

*Conflicts of interest.* The authors have no conflicts of interest to declare.

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

*Editor: N. Barton*