## [Peer Review File · Genetics]

Estimating the additive genetic variance for relative fitness from changes in allele frequency

Manas Geeta Arun, Aidan W. Angus-Henry, Darren Obbard, and Jarrod Hadfield

NOTE: The reviews and decision letters are unedited and appear as submitted by the reviewers.

In extremely rare instances and as determined by a Senior Editor or the EIC, portions of a review may be redacted. If a review is signed, the reviewer has agreed to no longer remain anonymous.

The review history appears in chronological order.

Review Timeline:

Submission Date:	2025-05-13
Editorial Decision:	2025-06-08
Resubmission Received:	2025-08-28
Editorial Decision:	2025-09-26
Resubmission Received:	2025-10-13
Accepted:	2025-10-16

June 8, 2025

GENETICS-2025-308171

Estimating the additive genetic variance for relative fitness from changes in allele frequency

Dear Dr. Arun:

Two experts in the field have reviewed your manuscript, and I have read it as well. This is an important paper, and could in principle be of wide interest. While your manuscript is not currently acceptable for publication in GENETICS, we would welcome a substantially revised manuscript. The reviewers have comments and concerns to be addressed in a revised manuscript. You can read their reviews at the end of this email, as well as some comments of mine.

Selection causes systematic changes in allele frequency, which can now be seen in genome-wide datasets. Buffalo and Coop estimated the fraction of allele frequency change attributable to selection in three empirical studies, but did not explicitly estimate the additive fitness variance. In this paper, Arun et al build on that work, to set out an elegant and general framework for estimating V_a , and validate it by simulation.

The first reviewer (a trainee) was generally positive; their comments may be useful for improving clarity. (Their point about plant vs animal studies I think arises from a difference in terminology). The third reviewer appreciates the elegance of the method, but would like to see a more accessible and intuitive explanation, with an empirical example, and associated software.

The second reviewer is impressed, but has some serious concerns. Their main worry is that the method is not as general as claimed: they have specific concerns about implicit assumptions, and also point out that simulations are run under essentially the assumptions of B&Coop, and so establish accuracy rather than generality. They also point out problems with definitions when change is over multiple generations, and ask for more precise citations (for example, to justify using fixed vs random effects).

Overall, this paper makes a substantial theoretical advance, and can stand alone, without application to data (that would be a substantial exercise in itself, and would over-burden the paper). However, the points raised by the second reviewer must be addressed convincingly, to clarify the likely scope of the method. In addition, arguments must be made more accessible, particularly to empirical workers who will be interested in applying the method.

We look forward to receiving your revised manuscript. Please let the editorial office know approximately how long you expect to need for revisions.

Upon resubmission, please include:

1. A clean version of your manuscript;
2. A marked version of your manuscript in which you highlight significant revisions carried out in response to the major points raised by the editor/reviewers (track changes is acceptable if preferred);
3. A detailed response to the editor's/reviewers' feedback and to the concerns listed above. Please reference line numbers in this response to aid the editor and reviewers.

Your paper will likely be sent back out for review.

Additionally, please ensure that your resubmission is formatted for GENETICS
<https://academic.oup.com/genetics/pages/general-instructions>

Follow this link to submit the revised manuscript: Link Not Available

Sincerely,

Nick Barton
Senior Editor
GENETICS

Approved by:
Howard Lipshitz
Editor in Chief
GENETICS

Reviewer #1 :

Peer Review: Estimating the additive genetic variance for relative fitness from changes in allele frequency by Manas Geeta Arun et al. 2025.

"Estimating the additive genetic variance for relative fitness from changes in allele frequency" by Manas Geeta Arun et al. develops a novel statistical framework for estimating the additive genetic variance in relative fitness (VA) using changes in allele frequencies. Traditionally, VA is estimated through pedigree-based approaches, experimental crosses and/or fitness proxies, which are difficult to obtain for several species. The authors instead use temporal allele frequency data (such as from evolve-and-resequence experiments) and a quantitative genetic "top-down" framework to infer VA directly from genome-wide changes in allele frequency.

As the authors claim in the article, while the question is not entirely new, their study presents an advance by enabling the estimation of additive genetic variance for fitness using only genomic data from time-series experiments. This opens possibilities for large-scale comparative studies of evolvability across taxa and environments, particularly where conventional fitness measurements are challenging.

The article is well-written and represents an impressive amount of work. However, it is dense and difficult to follow, especially after considering all the assumptions and considerations the authors establish for running the simulations.

Concerns

In the introduction, it is unclear whether the authors are referring to animal models, plant models, or other organisms. When they claim that few studies in quantitative genetics calculate VA in the field, this is not accurate. There is a substantial body of work presenting VA in plants, for example, under field conditions-not only in laboratory settings. There is an entire field of ecological genetics/genomics. It would be beneficial if the authors acknowledged and incorporated these references and provided better context. They should also specify which models they are referring to when describing wild populations, as the manuscript appears to focus on animal populations. The authors also claim that fitness proxies as the gold standard to calculate VA, which is true, but in annual plants (for example), survival and direct fitness measures are routinely taken and they are considered as direct fitness estimations.

Another concern is that if the authors consider how pseudo SNPs and copy number variants (CNVs) can affect the entire estimation of VA?, it seems they do not address how filtering these variants would affect VA estimates. Most studies have not taken CNVs into account, yet it has been proven that CNVs can inflate several genetic population estimations, and I presume this would also affect VA. It would be good if they at least discuss this. The authors should also provide context for why they use an additive model for log absolute fitness, $\log(W)$, rather than raw absolute fitness or relative fitness.

The methods section is extremely dense and difficult to follow, remembering all the parameters, and understanding after presenting numerous considerations, adjustments, and options is mind-blowing. A diagram/figure or workflow would greatly benefit this paper, given the many assumptions and considerations involved.

I have some minor comments about the technical specifications of the method;

The entire framework hinges on the accurate estimation of the genome-wide LD matrix in the base population. In real-world datasets, especially for non-model organisms, estimating this matrix precisely is challenging due to limited sample sizes, sequencing errors, and phasing uncertainty. How do the authors account for this?

Although the model allows recombination rates between loci to vary, it assumes these rates are known or can be approximated accurately. This is not always the case, especially in taxa with poorly resolved genetic maps.

Immigration, assortative mating, and overlapping generations can masquerade as selection and inflate VA estimates. Is this assumption true in nature? Moreover, sampling error in allele frequencies is treated as drift, but non-binomial errors (unequal coverage, mapping bias) will again inflate residuals and reduce statistical power. How does the method deal with this?

Simulation results show biases under strong selection (larger effect sizes) or with increasing recombination (e.g., downward bias with large n scale or high map lengths). The authors attribute this to the loss of additive genetic variance, but this highlights that the method breaks down in biologically realistic scenarios where large-effect alleles or directional selection are common. This restricts the practical utility of the method to systems with weak selection and high recombination. Thus, discussing these limitations more thoroughly, particularly regarding strong selection scenarios, would benefit to understand and clarify better the method.

While the authors position their method as a generalization or relaxation of Buffalo & Coop's model, it still retains critical assumptions (e.g., about LD decay, random mating, no selection during estimation of recombination decay). Their critique of Buffalo & Coop is valid but simultaneously underplays the restrictive assumptions of their own model (e.g., fixed α across time,

accurate L0, and homoscedastic drift). The comparison, though well-detailed, may not sufficiently justify the complexity of their model given the modest empirical gains shown in simulations.

Reviewer #2 :

Comments are in a pdf entitled comments_to_authors.pdf

Reviewer #3 :

Review of "Estimating the additive genetic variance for relative fitness from changes in allele frequency"

This manuscript presents a novel theoretical framework for estimating the additive genetic variance for relative fitness directly from genome-wide changes in allele frequency observed across experimental replicates. The authors build upon Fisher's Fundamental Theorem of Natural Selection to develop mathematical relationships between VA, linkage disequilibrium matrices, and expected changes in allele frequency due to selection. The theoretical contribution is substantial and mathematically rigorous. The authors derive an interesting and general expression showing that $VA = E[\Delta p]^\top L^{-1} E[\Delta p]$, where $E[\Delta p]$ is the expected changes in allele frequency and L is the genome-wide diversity/linkage disequilibrium matrix. This approach elegantly sidesteps many of the practical difficulties of traditional pedigree-based methods for estimating VA, particularly the challenge of defining appropriate fitness proxies. The method relaxes several restrictive assumptions of previous approaches, notably the B&C method, including the requirement to distinguish selected from neutral sites and assumptions about linkage disequilibrium patterns.

The simulation studies are comprehensive and demonstrate that the method provides unbiased estimates of VA across a wide range of biologically realistic parameter combinations. The comparison with B&C clearly shows superior performance, particularly at low recombination rates where alternative approaches exhibit substantial bias.

The mathematical complexity is indeed formidable and may limit accessibility to the genetics community. The derivation spans multiple appendices with intricate notation that could benefit from simplification or more intuitive explanation. Although the authors provide some biological interpretations, the gap between theory and practical implementation is still substantial.

A significant limitation is the lack of any empirical example. For a method claiming practical utility, demonstration on real evolve-and-resequence data would significantly strengthen the manuscript's impact. The data requirements are: phased genomes from base populations, precise recombination maps, and carefully controlled experimental replicates. These prerequisites may severely constrain the method's practical applicability.

The authors acknowledge that several assumptions may be violated in real populations, including independence of unpredictable selection responses across replicates and potential complications from varying selection environments.

The manuscript would benefit from a clearer presentation of the core algorithm and its computational requirements. The development of accompanying software would enhance its potential for application. Additionally, while the discussion of limitations is thorough, it could be helpful for readers to assess the practical feasibility.

Associate Editor Comments:

Eq 15 only includes dominance - how restrictive is this?

329 Why should residual variance equal one ???

477 Fitness variance is maintained solely by mutation, which likely underestimates its actual extent.

Fig 1 Remind the reader what p_α is; generally, make captions more self-contained.

pp 19-20 The justification for these scalings did not convince me - at least, the explanation was baffling. The constraint is primarily on the number of simulated loci. However, since the inference is on a small experimental population (1000 individuals), selected for a modest time, only the effects of fairly large blocks of genome should matter. This suggests that the key parameter is the fitness variance per map length. The simulation regime could be better justified by showing that this is the case, so that a few thousand loci on a realistic genome can represent a much larger number, each with smaller effect. This seems more straightforward than simulating over a broad range of (mostly unrealistic) recombination rates.

754-780 Following on from this, can one show more directly whether it is the fitness variance per map length that is being estimated? In a simulation, one can label the initial genomes, and then identify 'chunks' directly. How are these reflected in the

low-dimensional structure derived from the LD matrix?

Review for: Estimating the Additive Genetic Variance for Relative Fitness from Changes in Allele Frequency

In this paper, the authors develop a method to estimate the additive variance in relative fitness, V_A , without measuring fitness proxies. Estimating additive variance in fitness is typically challenging because it requires detailed fitness and pedigree data, which are hard to obtain. Their method instead harnesses the role that additive fitness variance plays in determining the rate of adaptation and—instead of fitness data for individuals—it requires data for changes in allele frequency for multiple replicates in a population for which the initial LD structure is known. The method is similar to that developed in Buffalo and Coop, 2019 [1]. However, relying on Robertson’s identity which gives a very general relationship between changes in allele frequencies and average effects of segregating alleles, they are able to relax many of the assumptions in Buffalo and Coop, 2019 [1]. Overall, I think the method and results presented in the paper are novel and interesting. In particular, I think the basic idea of connecting additive genetic variance to changes in allele frequency via average effects (in a way that is model independent) is elegant and important. Nevertheless, I do have a number of major concerns that I discuss below. Possibly the most significant of these concerns is that I believe the authors method is considerably less general than advertised, when used over multiple generations (details below). In spite of this, I think the potential for the method to be used under quite general conditions with data providing changes in allele frequency over multiple replicates for a single (initial) generation is still exciting. However, it is notable that although the method is meant to be incredible general in comparison to Buffalo and Coop (2019) [1], the simulations employed to test it do not relax the majority of the numerous simplifying assumptions employed by Buffalo and Coop. I think this is not necessary unacceptable since the authors approach does provide a considerable improvement in accuracy over Buffalo and Coop when some of those assumptions are relaxed, and impressively accurate estimates of additive genetic variance in the conditions in which it is tested (Figure 6), i.e., with random mating, no dominance, no epistasis. However, it still needs to be stated prominently as a caveat for the claim that the machinery is validated with individual based simulations (e.g., line 141 in the introduction) and also addressed in the Discussion section.

Some comments/concerns:

1. The authors express the additive genetic variance at time zero as $V_A(0) = \bar{\alpha}^\top \mathbf{L}_0 \bar{\alpha}$ where $\bar{\alpha}$ is the vector of mean average effects across *time* and replicates (line 192 and also in Eq. 12). But this is not correct, it should be $V_A(0) = \bar{\alpha}_0^\top \mathbf{L}_0 \bar{\alpha}_0$ where $\bar{\alpha}_0$ is the vector of mean average effects across replicates at time 0. The authors are approximating the initial average effects by a mean *across time* of average effects and this should not be obfuscated. This is relevant to line 433 where the authors say ”We also make Assumption H if we choose to interpret $V_A(0)$ as an additive genetic variance rather than an additive genetic covariance.” With $\bar{\alpha}$ an average over time, it is not at all clear how $\bar{\alpha}^\top \mathbf{L}_0 \bar{\alpha}$ can be interpreted as a covariance (more on this in a later comment). It is also relevant to line 435 where the authors say that ”Assumptions E & H result in the unpredictable response to selection being zero. While we do not make this assumption, we do assume that the unpredictable responses to selection are independent across replicates.” If more than one time point is being used and if that time point is not the initial time point, it is not obvious that the authors are not making this assumption (having implicitly assumed that $\bar{\alpha}_0^\top \mathbf{L}_0 \bar{\alpha}_0 = \bar{\alpha}^\top \mathbf{L}_0 \bar{\alpha}$ from the start)
2. I have a great many concerns about about the section ”Extending our approach to practical situations” starting line 181, which are in some ways related to the above comment. In line 232 the authors indicate that $\mathcal{L}_{t,m} \bar{\alpha}$ captures the predictable change due to selection. However, this ignores the fact that average effects, allele frequencies and linkage can all change due to selection in a way that is predictable. In line 235 the authors say ”Note that $\frac{\Delta \mathbf{p}}{D}$ and $\frac{\Delta \mathbf{p}}{U}$ have expectation zero and that the

$\frac{\Delta \mathbf{p}}{D}$ terms are independent across replicates, generations, and generations within replicates (Buffalo and Coop, 2019).” I do not see how Buffalo and Coop can be appealed to for an assertion that $\frac{\Delta \mathbf{p}}{U}$ has zero expectation, since Buffalo and Coop (and Santiago and Caballero, 1998, [4]) make many assumptions that the authors of this paper are trying to relax. I think the authors need to properly discuss the conditions under which $\frac{\Delta \mathbf{p}}{U}$ has zero expectation as numerous strong assumptions seem to go into that. Below I provide specifics, both about some of the assumptions that I think are required to make that assertion, and about my problems with this section:

- The authors ”decompose \mathbf{L} at a particular time into a part that can be predicted by \mathbf{L}_0 and the action of drift and recombination, and a part that cannot be predicted.” (line 206). They write the deviation of $\mathbf{L}_{t,m}$ —from the part that can be predicted by \mathbf{L}_0 and the action of *drift and recombination*—as $\Delta \mathbf{L}_{t,m} = \Delta \mathbf{L}'_{t,m} + \mathbf{L}''_{t,m}$, where ” $\mathbf{L}''_{t,m}$ is the matrix of nongametic-phase disequilibria that arises in generation t in replicate m ” (line 224) and ” $\Delta \mathbf{L}'_{t,m}$ is a stochastic term with zero expectation that represents the accumulated change in \mathbf{L}' between generations 0 and t in replicate m that cannot be predicted” (line 222). But there is a component of change in $\mathbf{L}_{t,m}$ that has been completely ignored: the change in $\mathbf{L}_{t,m}$ that can be predicted from \mathbf{L}_0 (and $\bar{\alpha}_0$) by the action of *selection*. Natural selection might well generate changes in allele frequency (the diagonal of \mathbf{L}) and correlations between alleles (the off-diagonal of \mathbf{L}) that are correlated between replicates and *have nonzero expectation*. Further, these changes might well be correlated with changes in average effects due to the action of natural selection. To neglect such changes it would appear that many of the assumptions of Buffalo and Coop would need to be made. In addition, it would seem that assuming $\mathbf{L}''_{t,m}$ has zero expectation requires assuming that non-gametic phase equilibrium is not generated by natural selection.
 - When $\Delta \alpha_{t,m}$ is the difference between the *initial* average effect $\bar{\alpha}_0$ and a deviation from that value (as it must be if $\bar{\alpha}_0$ is going to be used to estimate the initial additive variance), then dropping the Buffalo and Coop assumption of additive gene action would result in $\Delta \alpha_{t,m}$ having nonzero expectation. As an example, if we assume there is dominance then $\alpha_t = s/2(1 + (1 - 2h)(1 - 2p_t))$ (where s is the selection coefficient against the homozygote, h is the dominance coefficient and p_t the allele frequency at time t) and, importantly, expected changes in allele frequency at selected alleles are likely to have nonzero expectation. Consequently, average effects will have a nonzero expected change that is correlated across replicates.
 - More generally, when gene action is not additive, selection can remain constant in the sense that alleles causal effects on fitness can remain unchanged, but average effects on alleles might still change as allele frequencies and patterns of linkage disequilibrium change (owing to dominance or epistasis). And these changes are likely to be correlated to changes in \mathbf{L} .
3. From the section on ”Comparison with (Buffalo and Coop, 2019)” (starting line 338), it would seem that the authors believe that their method gets around the issues detailed in the previous remark. This makes it seem possible I have misunderstood their method in a way that accounts for all these concerns. In that case I still believe that the manuscript should be revised to increase clarity and make such misunderstandings unlikely. In this Section, variance as a function of time is defined (correctly) in Equation 21 in a way that contradicts the earlier definition of $V_A(0)$. What is referred to as additive genetic and genic covariances (Equations 23 and 24) are also defined. To be interpreted as covariances, they would need to be the genetic and genic covariances between the breeding value of a random individual in generation t with the breeding value that that same individual would have had if they had instead they existed in generation τ : thereby accounting for changes in average effects due to changes in the environment/selection but not changes that would occur as a result of allele frequency changes or changes in the correlation structure between alleles. The authors claim that the early (I believe incorrect) $V_A(0)$ is equal to this rather artificial (genetic) covariance when average effects are not constant in time (line 399)—however it is not clear how this could be so, i.e. it is not clear why $\alpha_t \mathbf{L}_t \alpha_\tau^\top$ should be approximated by $\bar{\alpha}^\top \mathbf{L}_0 \bar{\alpha}$, where $\bar{\alpha}$ is an average of α over time, without some pretty strong assumptions. Even more confusingly, earlier in the paper (line 291) and in the discussion (line 874), it is stated that the covariance in fitness that the method is meant to be estimating when average effects are not constant, is a covariance between replicates (rather than between times). It would be helpful if this between-replicate covariance were defined, i.e., if it were stated between what two random variables the quantity is a covariance, since this requires some care.

And also if the inconsistency in exactly which covariance is being estimated when average effects are not constant were resolved. Indeed, mathematical expressions should be provided to demonstrate that this (properly defined) covariance actually corresponds with the quantity being estimated as is claimed (many times, e.g., line 434). I would also like to see some comment on the significance of this quantity—on the face of it, it rather artificial; is there any reason we would want to estimate it beyond its potentially providing a reasonable approximation for the additive variance?

4. Another concern that I had with the manuscript was that at some point (in the paragraph starting line 259, p9) the authors shift from talking about fixed effects to random effects without offering any explanation of what this means and how the quantities that they infer connect to those of classical quantitative genetics. $\boldsymbol{\mu}_{\bar{\alpha}}$ and $\mathbf{V}_{\bar{\alpha}}$ are described as the mean and covariance structure of the mean average effects, but it is not stated what this mean is being taken over. How are we to interpret $\mathbf{V}_{\bar{\alpha}}$? Is it some kind of Bayesian uncertainty? (in which case, it surely should not appear in an estimate of additive genetic variance). Upon switching to random effects, the authors say to "See Appendix S3 and Gianola et al. (2009) for a discussion on what this implies". Appendix 3, which I discuss in a later comment, touches on this point only tangentially; it just discusses how, given a switch to random effects, one needs to ensure the distribution of $\bar{\alpha}$ transforms appropriately when the reference and alternate allele are switched. So, to try and understand the implications, I read the first few sections of Gianola et al. [2] which examines "relationships between the (Bayesian) variance of marker effects in some regression models and additive genetic variance", and this did not bring clarity. To quote from that article: "In short, the connection between the uncertainty variance σ_a^2 and the additive variance V_A (which involves the effect of the locus) is elusive when both genotypes and effects are random variables." If the authors possess a thorough understanding of the implications of switching from fixed effects to random effects and also how the reader should interpret $\boldsymbol{\mu}_{\bar{\alpha}}$ and $\mathbf{V}_{\bar{\alpha}}$ (and the parameters they infer to describe them: $\beta_{\bar{\alpha}}^{(0)}$, $\beta_{\bar{\alpha}}^{(1)}$ and $p_{\bar{\alpha}}$), then this should be clearly explained. If, however, there is simply a lack of clarity on these issues not only in this manuscript, but elsewhere in the literature, then this should be made transparent. In the discussion (lines 775 to 777), the authors say that in "the simplified simulations, in which the relationship between allele frequencies and α 's was expected to be absent, the model requires only a single parameter: the variance in average effects, $\sigma_{\bar{\alpha}}^2$ ", which seems to asserting that $\sigma_{\bar{\alpha}}^2$ is the variance (across alleles segregating in the population) of average effects, and this is *only* true when $p_{\bar{\alpha}}$ is zero (as is indeed the case with their simplified simulations). More generally, it seems the randomness in $\bar{\alpha}$ assumed by the authors arises not from Bayesian uncertainty but from variation in average effects across loci, and that making the randomness locus specific is not a means to specify locus-specific uncertainty. Rather it is a tool to allow the assumed distribution of average effects to vary with allele frequency. When $p_{\bar{\alpha}}$ is zero then $\sigma_{\bar{\alpha}}^2$ is the variance of average effects, when $p_{\bar{\alpha}}$ is minus one, $\sigma_{\bar{\alpha}}^2$ is the expected per allele contribution to additive variance, and for other values of $p_{\bar{\alpha}}$ the relationship is more complex. If this is so, it would be helpful if this, along with interpretations of $\beta_{\bar{\alpha}}^{(0)}$, $\beta_{\bar{\alpha}}^{(1)}$ and $p_{\bar{\alpha}}$ were made explicit.
5. It took me quite a while to get to grips with the purpose of Appendix 3 (starting line 1225) and I am still not completely confident that I have. I think it would be helpful to orient the reader, if the authors state near the beginning of the Appendix that the purpose of the Appendix is to obtain assumptions on distribution of random effects which ensure that the distribution transforms appropriately when the reference and alternate allele are switched—and to provide candidates for the mean and variance that comply with these assumptions. This, at least, is the purpose of the Appendix as I understand it. The authors conclude that a choice of $V_{\bar{\alpha}} = \sigma_{\bar{\alpha}}^2 \mathbf{L}_0^p$ satisfies the requirement that covariances involving a locus switch sign when its allele coding is flipped (but its variance remains positive). The authors also say (line 1256) "For, $\boldsymbol{\mu}_{\bar{\alpha}}$ this implies that suitable models should be (weighted) sums of differences between invariant properties of the alleles such that the difference reverses sign when the reference and alternate allele are switched." This is confusing given that in the main text the authors model the mean effects as:

$$\boldsymbol{\mu}_{\bar{\alpha}} = \beta_{\bar{\alpha}}^{(0)} + \beta_{\bar{\alpha}}^{(1)}(\mathbf{p}_0 - \mathbf{q}_0)$$

(Equation 15 on page 11) which, for $\beta_0 \neq 0$ does not immediately appear to satisfy this requirement. It would be helpful if the authors stated explicitly in the Appendix how their conclusions about permissible models for $\boldsymbol{\mu}_{\bar{\alpha}}$ are captured by the model employed in the main text. Unrelated, but also in Appendix S3, Equation S19 differs from Equation 12 in the main text in ways that are confusing: $V_A(0)$ changes to V_A and the expectation gets a subscript of \mathbf{L}_0 the purpose of which is unclear.

6. Near the beginning of the section on the Inference Outline, the authors say (line 299) "We chose a projection that collapses allele frequency changes into the non-null subspace of \mathbf{L}_0 , since $V_A(0)$ only depends on this subspace (de Los Campos et al., 2015)." I took an (admittedly fairly cursory) look at de Los Campos et al. (2015) [3] and so far as I can tell the goal of this paper was to develop a rigorous definition of "genomic heritability" within a classical quantitative genetics framework, distinguishing it from statistical model-based interpretations used in whole-genome regressions. There is a particular focus on highlighting that estimates from marker-based models may suffer from bias or inconsistency when causal variants are not fully captured, and the paper cautions against naive inference from such models. However, I could not locate the result that $V_A(0)$ depends only on the non-null subspace of \mathbf{L}_0 cited by the authors from that paper. If $V_A(0)$ does indeed only depend on the non-null subspace of \mathbf{L}_0 , then I believe work is required to move from what is presented in de Los Campos et al. (2015) to that conclusion. It does seem fairly plausible; nevertheless, if the authors are just assuming it to hold, then there should be transparency about this.
7. In numerous places in the manuscript, subscripts are given subscripts in a way that seems inappropriate. For example, in Equation (S4) I strongly believe that $L''_{i+1,j_{t+1+1}}$ is *meant* to denote the (i, j) th element of \mathbf{L}''_{t+1} . If this is correct, then $t + 1$ should be a subscript (or superscript or a something) attached directly to L'' , not to the i and j (unless the specific loci being indexed are actually changing with time, which seems problematic). This issue exists in many places for many variables and should be corrected throughout.

References

- [1] V. Buffalo and G. Coop. "The Linked Selection Signature of Rapid Adaptation in Temporal Genomic Data". In: *Nature Ecology & Evolution* 3 (2019), pp. 329–335. DOI: 10.1038/s41559-019-0792-7.
- [2] Daniel Gianola et al. "Additive genetic variability and the Bayesian alphabet". In: *Genetics* 183.1 (2009), pp. 347–363.
- [3] Gustavo de Los Campos, Daniel Sorensen, and Daniel Gianola. "Genomic heritability: what is it?" In: *PLoS Genetics* 11.5 (2015), e1005048.
- [4] Enrique Santiago and Armando Caballero. "Effective size and polymorphism of linked neutral loci in populations under directional selection". In: *Genetics* 149.4 (1998), pp. 2105–2117.

We thank the Reviewers and Associate Editor for their constructive comments on
this manuscript and their generally positive reviews. Our detailed responses to the
comments can be found below each comment and here we detail the more substantial
changes to the manuscript we have made:

- • As suggested by Reviewers 1 and 3, we have included a box that describes
the step-by-step workflow for implementing our method. Additionally, in re-
sponse to some of the concerns raised by Reviewer 2, we have now significantly
rearranged and rewritten the sections ‘Extending our approach to practical sit-
uations’ and ‘Inference outline’ so that the infinitesimal model assumptions of
the approach are more transparent and the implication/interpretation of treat-
ing $\bar{\alpha}$ as random is clearer. We hope this improves the clarity and accessibility
of our paper.
- • Reviewer 2 also pointed out that if allelic effects are non-additive, evolutionary
change due to selection can bring about systematic changes in average effects.
To address this important concern, we have now included results from simula-
tions incorporating dominance effects. The method appears relatively robust
to dominance.
- • Reviewers 1 and 3 raised important questions about the practicalities of apply-
ing the method to real-world systems. Reviewer 1 pointed out that sampling
noise around true allele frequencies may not be binomial. We think this is an
important point and we have now developed additional methodology to accom-
modate it and used it to reanalyse some of our simulations in which we obtain
estimated allele frequencies via simulated pool-seq. The method appears rela-
tively robust at modest coverage and seems to be robust to overdispersion in
the number of reads mapping to an individual which is estimated.
- • Finally, we have corrected some small mistakes in the application of our ap-
proach and the simulation code. Reviewer 2 pointed out that that our original
model for the mean average effects for fitness was not invariant to reference al-
lele assignment. When calculating the expected changes in the genome-wide LD
matrix \mathbf{L} we accidentally used N_e^2 instead of N_e . In the simplified simulations,
the recombination rates were inadvertently different between the msprime and
SLiM parts of the history phase. Additionally, the scales of the gamma ‘DFE’s
described in the original manuscript were half the actual scales used in the
simulations. We would like to apologise for these errors. We have now cor-
rected these errors and reanalysed our simulations. None of our results change
significantly.

- • In relation to the last point, we also felt that using the scale of the gamma
distribution was not the most natural way of representing the expected effect
sizes of mutations on fitness in the figures. Therefore, we have now replaced
the scale with the mean of the non-neutral gamma DFE (i.e. $\text{scale} \times \text{shape}$).
Note that the shape of the gamma DFE was fixed to 0.3 in all our simulations.

In order to highlight the changes we have made in this revision, wherever required
in this document, we use line numbers from the clean (i.e. non-marked) version of
the revised manuscript, unless explicitly specified otherwise. Additionally, we could
not get figures to be cross-referenced correctly in the marked version of the revised
manuscript, in which supplementary figures are also referred to as ‘Figure 1’, ‘Figure
2’, etc. instead of ‘Supplementary Figure 1’, ‘Supplementary Figure 2’. We apologise
for this and request the reviewers to refer to the figures in the clean version of the
manuscript.

**Associate Editor**

Selection causes systematic changes in allele frequency, which can now be seen in
genome-wide datasets. Buffalo and Coop estimated the fraction of allele frequency
change attributable to selection in three empirical studies, but did not explicitly
estimate the additive fitness variance. In this paper, Arun et al build on that work,
to set out an elegant and general framework for estimating V_a , and validate it by
simulation.

The first reviewer (a trainee) was generally positive; their comments may be
useful for improving clarity. (Their point about plant vs animal studies I think
arises from a difference in terminology). The third reviewer appreciates the elegance
of the method, but would like to see a more accessible and intuitive explanation,
with an empirical example, and associated software.

The second reviewer is impressed, but has some serious concerns. Their main
worry is that the method is not as general as claimed: they have specific concerns
about implicit assumptions, and also point out that simulations are run under essen-
tially the assumptions of B&Coop, and so establish accuracy rather than generality.
They also point out problems with definitions when change is over multiple gen-
erations, and ask for more precise citations (for example, to justify using fixed vs
random effects).

Overall, this paper makes a substantial theoretical advance, and can stand alone,
without application to data (that would be a substantial exercise in itself, and would
over-burden the paper). However, the points raised by the second reviewer must
be addressed convincingly, to clarify the likely scope of the method. In addition,
arguments must be made more accessible, particularly to empirical workers who will
be interested in applying the method.

We thank the Associate Editor for their summary of the reviewers' comments. We
have now made substantial changes to our manuscript to address the issues raised by
the reviewers and the Associate Editor and these are detailed above. More detailed
responses to specific comments by the Associate Editor and the Reviewers can be
found below.

Eq 15 only includes dominance - how restrictive is this?

Embarrassingly, we had not even noticed the (original) model would also be the
model for the average effect if the additive and dominance deviations were constant.
The motivation behind the model was simply to allow the average affects to depend
on allele-frequency in a way that is not sensitive to the choice of the reference allele.

Reviewer 2 pointed out that the model defined by the original model does, in fact,
depend on the choice of the reference allele if the intercept term, $\beta_{\alpha}^{(0)}$, is not zero. We
have now removed the intercept in the manuscript (Lines 332-333) and reanalysed
our simulations using the new model. However, we note that the term $\beta_{\alpha}^{(1)}$ is not
simply informative about dominance - even when we simulated log-linear additivity
(where dominance is negligible) $\beta_{\alpha}^{(1)}$ is estimated to be negative since deleterious
alleles tend to be rarer than beneficial alleles. For the new simulations where deleterious
alleles are recessive, we do see however that $\beta_{\alpha}^{(1)}$ becomes more negative as the
Associate Editor intuits (Figure 3C).

329 Why should residual variance equal one ???

The important point here is that we are working with suitably projected allele
frequency change data. We calculate the covariance structure for allele frequency
change due to drift and calculate its eigenvectors (U) and eigenvalues (D). We then
project the allele frequencies onto this new basis defined by the U , which makes the
residuals independent (since eigenvectors are orthogonal), and then scale the residu-
als by D^{-1} which causes them to have a variance of 1 if drift alone contributes to the
residuals (i.e. if unpredictable responses to selection are absent). However, in our
simulations the residual variance was slightly greater than one (Figures 1d and 4d)
and we believe this is because our N_E calculations only incorporated variability in
offspring number due to environmental factors (and even then was an approximation)
and ignored any non-additive genetic variance. Indeed, in our new simulations we
note that the estimate of the residual variance becomes increasingly greater than one
as the degree of dominance increases, presumably because the non-additive genetic
variance is reducing N_E by more than was assume (Figure 3d). However, we have
also shown that our inferences regarding V_A are largely insensitive to errors in N_E as
the estimate of the residual variation becomes smaller or larger to accommodation
any misspecification. In fact, we can even replace N with N_E and use the estimate
of the residual variance (which would then likely be far greater than one) to obtain
estimates of N_E (Supplementary Figure 3).

477 Fitness variance is maintained solely by mutation, which likely underesti-
mates its actual extent.

This is true and we now acknowledge it in the ‘Multilocus simulations’ section
with other caveats highlighted by Reviewer 2. (Lines 510-516)

Fig 1 Remind the reader what p_α is; generally, make captions more self-contained.

**Corrected**

pp 19-20 The justification for these scalings did not convince me - at least, the
explanation was baffling. The constraint is primarily on the number of simulated
loci, However, since the inference is on a small experimental population (1000 indi-
viduals), selected for a modest time, only the effects of fairly large blocks of genome
should matter. This suggests that the key parameter is the fitness variance per map
length. The simulation regime could be better justified by showing that this is the
case, so that a few thousand loci on a realistic genome can represent a much larger
number, each with smaller effect. This seems more straightforward than simulating
over a broad range of (mostly unrealistic) recombination rates.

The key point here is that our simulations have two distinct phases with two
slightly different goals. In the ‘experiment’ phase, we simulate a typical evolve-and-
resequence experiment and generate data using which inferences on V_A are then to
be made. We agree with the Associate Editor that one of the relevant parameters
in this phase of the simulations is the the additive genetic variance per unit map
length, the other being the rate at which LD between closely-linked sites gets de-
pleted by recombination. On the other hand, evolution in the ‘history’ phase does
not by itself contribute any data that is used for inference and is relevant only as far
as its consequences on the genetic composition of the base population from which
the experiment phase is subsequently derived. Here, it is very hard to come up with
a single set of scalings that produce a base-population that is realistic in all aspects
such as haplotype composition, dependencies of allele frequency on average effect and
linkage-disequilibria between selected sites. Recent work suggests that there proba-
bly isn’t a single scaling that would generate realistic properties for all aspects of the
base-population (Dabi and Schrider, 2025). However, we agree that our explanation
for scalings used in our simulations is rather confusing. We have now re-written this
section following the Associate Editor’s suggestion. (Lines 627-653)

754-780 Following on from this, can one show more directly whether it is the
fitness variance per map length that is being estimated? In a simulation, one can
label the initial genomes, and then identify ‘chunks’ directly. How are these reflected
in the low-dimensional structure derived from the LD matrix?

We do not think our method estimates the fitness variance per map length. It

estimates the total additive genetic variance contributed by the genome, irrespective
of the map length of the genome.

An important point to note is that the 'chunks' are not defined by recombination
during the experiment phase, but, instead, depend on the linkage structure of the
base population. We agree that a visual description of what the 'chunks' defined
by our projection matrix actually represent would have been useful for the readers.
However, plotting the loadings of the first few eigenvectors of L_0 versus the positions
of sites segregating in the base population did not result in any visually striking pat-
terns - especially at higher map lengths. Therefore, we have chosen not to include
these figures in the manuscript.

Reviewer 1

"Estimating the additive genetic variance for relative fitness from changes in
allele frequency" by Manas Geeta Arun et al. develops a novel statistical framework
for estimating the additive genetic variance in relative fitness (VA) using changes in
allele frequencies. Traditionally, VA is estimated through pedigree-based approaches,
experimental crosses and/or fitness proxies, which are difficult to obtain for several
species. The authors instead use temporal allele frequency data (such as from evolve-
and-resequence experiments) and a quantitative genetic "top-down" framework to
infer VA directly from genome-wide changes in allele frequency.

As the authors claim in the article, while the question is not entirely new, their
study presents an advance by enabling the estimation of additive genetic variance for
fitness using only genomic data from time-series experiments. This opens possibili-
ties for large-scale comparative studies of evolvability across taxa and environments,
particularly where conventional fitness measurements are challenging.

The article is well-written and represents an impressive amount of work. How-
ever, it is dense and difficult to follow, especially after considering all the assumptions
and considerations the authors establish for running the simulations.

In the introduction, it is unclear whether the authors are referring to animal mod-
els, plant models, or other organisms. When they claim that few studies in quanti-
tative genetics calculate V_A in the field, this is not accurate. There is a substantial
body of work presenting V_A in plants, for example, under field conditions - not only
in laboratory settings. There is an entire field of ecological genetics/genomics. It
would be beneficial if the authors acknowledged and incorporated these references
and provided better context. They should also specify which models they are re-
ferring to when describing wild populations, as the manuscript appears to focus on

animal populations.

We feel the reviewer has slightly misinterpreted our point. We never suggested
that there are few quantitative genetic studies estimating V_A . In the Introduction,
we were trying to make the following points. First, the data required for measuring
V_A in the wild are difficult to obtain. Second, often there are confounding factors
in wild systems that are difficult to resolve. And, traditional methods depend on
using one (among many possible) measures of fitness. We had referred to three meta
analyses that have reviewed studies estimating V_A . We do agree with the reviewer's
concern that the studies that these meta-analyses (with the exception of Burt (1995))
are either exclusively focused on animals or exhibit a bias towards animal species.
We have now corrected this by incorporating a reference to estimating V_A from field
populations of annual plants. (Lines 109-113)

The authors also claim that fitness proxies as the gold standard to calculate V_A ,
which is true, but in annual plants (for example), survival and direct fitness measures
are routinely taken and they are considered as direct fitness estimations.

We apologise for the confusion caused by the word 'proxies'. Our only point
here is that no definition of fitness – even seemingly complete measurements such as
survival and lifetime reproductive success – is perfect, and that there is little con-
sensus among evolutionary biologists for what the best definition of fitness ought to
be. Therefore, the best that empiricists can do is use the most appropriate measure
of fitness in their model system, which we had referred to as a 'proxy' for fitness.
However, we have now realised that the phrase 'proxies of fitness' was a poor choice,
as it may imply characters correlated with (but not direct measurements of) fitness
(components) such as body size. We have avoided using the phrase 'proxy/proxies
of fitness'. (For example, Lines 18-19 and Lines 117-120)

Another concern is that if the authors consider how pseudo SNPs and copy num-
ber variants (CNVs) can affect the entire estimation of V_A ?, it seems they do not
address how filtering these variants would affect V_A estimates. Most studies have
not taken CNVs into account, yet it has been proven that CNVs can inflate several
genetic population estimations, and I presume this would also affect V_A . It would be
good if they at least discuss this.

Since our method explicitly uses information on the LD between sites, we do not
think we need any additional machinery to deal with pseudo-SNPs, which are, es-

sentially, a group of SNPs in perfect linkage. At the same time, we agree that there
will inevitably be variants (like CNVs, but also others) that may be missed and, as
a consequence, not included in the analysis. However, our method estimates V_A by
tracking the frequencies of *chunks* of genome defined by linkage, as opposed to track-
ing allele frequency change at individual sites. Therefore, even if some variants are
not included in the analyses, their contribution to V_A can, in principle, be captured
by measuring the changes in the frequency of the respective chunks of genome these
variants map to.

The authors should also provide context for why they use an additive model for
log absolute fitness, $\log(W)$, rather than raw absolute fitness or relative fitness.

This is an important point. We had referred to the consequences of this choice
on the average effects for relative fitness in the ‘Model for fitness’ subsection in the
‘Multilocus simulations’ section, with formal derivations included in Appendix S6 in
the original manuscript (S8 in the revised manuscript). However, we have realised
that a clearer explanation is warranted. We have now added the following sentence:
‘This choice was motivated by the fact that the variance in $\log(W)$ is approximately
equal to the variance in relative fitness (see Appendix 1 in Lynch and Walsh (1998))’
(Lines 522-526). In addition, it also ensures that fitness remains positive even when
genotypic values can be simulated without constraint.

The methods section is extremely dense and difficult to follow, remembering
all the parameters, and understanding after presenting numerous considerations, ad-
justments, and options is mind-blowing. A diagram/figure or workflow would greatly
benefit this paper, given the many assumptions and considerations involved.

We appreciate the reviewer’s point. This issue was also flagged by reviewer 3.
We have now included a new box (‘Box 1’) (Line 1830) that provides a step-by-
step workflow of our method. Furthermore, we have now substantially rewritten
and rearranged the theory sections (including the corresponding supplements) to ac-
commodate concerns raised by Reviewer 2, and reorganised our description of the
simulation parameters to make it more streamlined. However, we acknowledge that
the work remains difficult, and that even we continue to struggle with some aspects
of it.

I have some minor comments about the technical specifications of the method;
The entire framework hinges on the accurate estimation of the genome-wide LD

matrix in the base population. In real-world datasets, especially for non-model or-
ganisms, estimating this matrix precisely is challenging due to limited sample sizes,
sequencing errors, and phasing uncertainty. How do the authors account for this?
Although the model allows recombination rates between loci to vary, it assumes these
rates are known or can be approximated accurately. This is not always the case, es-
pecially in taxa with poorly resolved genetic maps. Immigration, assortative mating,
and overlapping generations can masquerade as selection and inflate V_A estimates.
Is this assumption true in nature?

We agree with the reviewer that the data requirements for applying our method to
real-world systems – especially wild populations – are quite strong. In fact, we have
already described in considerable detail many of the issues raised by the reviewer
here (eg. non-availability of L , non-availability of recombination maps, migration,
errors in estimating the number of generations, etc.) in the Discussion section. We
have also discussed other challenges such as non-availability of estimates of N_E , con-
sequences of selection varying in time and space, etc. (Lines 1008-1052). While we
acknowledge that these may seem like formidable challenges, the best we can do at
the moment is to be as transparent about the data requirements and challenges as
possible such that empirical workers can identify settings where the method can be
applied with confidence.

Moreover, sampling error in allele frequencies is treated as drift, but non-binomial
errors (unequal coverage, mapping bias) will again inflate residuals and reduce sta-
tistical power. How does the method deal with this?

This is a crucial point. In a new supplement (Supplementary information S6), we
have now derived an expression for the covariances in estimation error in projected
allele frequency change that arises in pool-seq and has an additional overdispersion
parameter that accounts for any heterogeneity in the probability that an individual
is sampled. We refer to this in the ‘Inference outline’ subsection in the main text
(Lines 384-393). Using (an approximation to) this method, we have now re-analysed
some of our existing simulations by obtaining allele frequencies in the experiment
phase by mimicking pool-seq sampling (Figure 6 and Supplementary Figure 6). The
details of the approach we took for this can be found in the subsection ‘Simulating
pool-seq’ in the section ‘Multilocus simulations’ (Lines 725-752). Briefly, we choose a
read-length such that the average number of segregating sites a read covers is similar
to what would be observed in a typical *D. melanogaster* pool-seq experiment (Bitter
*et al.*, 2024) using 150bp reads (~ 2.5 segregating sites per read). We then var-

ied average coverage, and whether there was overdispersion in the number of reads
mapping to an individual to simulate the process of mapping reads to the genome.
This generates natural variation in coverage across the genome and heterogeneity in
the probability that an individual is sampled. In the simplified simulations there is
little deterioration in the method if we drop coverage down to 100x (despite replicate
populations consisting of 1,000 individuals) (Figure 6). While this might be surpris-
ing, the relevant parameter is probably not coverage *per se*, but the number of reads
overlapping at least one segregating site per unit map length, as this determines how
accurately the frequency of a ‘chunk’ of genome can be measured. Using a similar
argument, Tilk *et al.* (2019) demonstrated that supplementing pool-seq sampling
with haplotype inference tools can result in nearly 500x ‘effective’ coverage at 10x
empirical coverage. For the full simulations, while unbiased estimates were obtained
at high coverage ($\geq 500x$), we observed a moderate downward bias at 100x coverage
(Supplementary Figure 6).

Simulation results show biases under strong selection (larger effect sizes) or with
increasing recombination (e.g., downward bias with large η_{scale} or high map lengths).
The authors attribute this to the loss of additive genic variance, but this highlights
that the method breaks down in biologically realistic scenarios where large-effect al-
leles or directional selection are common. This restricts the practical utility of the
method to systems with weak selection and high recombination. Thus, discussing
these limitations more thoroughly, particularly regarding strong selection scenarios,
would benefit to understand and clarify better the method.

We think the reviewer has slightly misunderstood our point about the negative
bias when the mean η was large. Our η 's were drawn from a gamma distribution.
We found that the downward bias was a consequence of loss in V_A driven by just 1%
loci with exceptionally large (greater than 0.3) effect sizes. Our point is that this is
an issue with our simulations not being realistic enough as opposed to an issue with
the method. While we have not yet diagnosed the cause of the negative bias at large
map lengths (in the history phase of the simulations), we do acknowledge this in the
Discussion section. (Lines 964-984)

While the authors position their method as a generalization or relaxation of Buf-
falo & Coop's model, it still retains critical assumptions (e.g., about LD decay, ran-
dom mating, no selection during estimation of recombination decay). Their critique
of Buffalo & Coop is valid but simultaneously underplays the restrictive assumptions
of their own model (e.g., fixed α across time, accurate L0, and homoscedastic drift).

The comparison, though well-detailed, may not sufficiently justify the complexity of
their model given the modest empirical gains shown in simulations.

The reviewer makes some fair points. To address the reviewer's point about
constant α 's (which was flagged by Reviewer 2 as well), we have now included ad-
ditional simulations incorporating dominance (Figure 3, Supplementary Figure 4),
such that the α 's are frequency-dependent. Although, our theory models LD decay
as a consequence of the action of drift and recombination only and ignores selec-
tion, our simulations impose no such restriction. The \mathbf{L} matrix is free to evolve as
a consequence of selection in the simulations. In spite of this, our method provides
good estimates of V_A . As for the remaining limitations of our method, we have
acknowledged them to the best of our abilities in the Discussion section.

Reviewer 2

In this paper, the authors develop a method to estimate the additive variance in
relative fitness, V_A , without measuring fitness proxies. Estimating additive variance
in fitness is typically challenging because it requires detailed fitness and pedigree
data, which are hard to obtain. Their method instead harnesses the role that ad-
ditive fitness variance plays in determining the rate of adaptation and – instead of
fitness data for individuals – it requires data for changes in allele frequency for mul-
tiple replicates in a population for which the initial LD structure is known. The
method is similar to that developed in Buffalo and Coop (2019). However, relying
on Robertson's identity which gives a very general relationship between changes in
allele frequencies and average effects of segregating alleles, they are able to relax
many of the assumptions in Buffalo and Coop (2019). Overall, I think the method
and results presented in the paper are novel and interesting. In particular, I think
the basic idea of connecting additive genetic variance to changes in allele frequency
via average effects (in a way that is model independent) is elegant and important.
Nevertheless, I do have a number of major concerns that I discuss below. Possi-
bly the most significant of these concerns is that I believe the authors method is
considerably less general than advertised, when used over multiple generations (de-
tails below). In spite of this, I think the potential for the method to be used under
quite general conditions with data providing changes in allele frequency over multiple
replicates for a single (initial) generation is still exciting. However, it is notable that
although the method is meant to be incredible general in comparison to Buffalo and
Coop (2019), the simulations employed to test it do not relax the majority of the
numerous simplifying assumptions employed by Buffalo and Coop. I think this is
not necessary unacceptable since the authors approach does provide a considerable

improvement in accuracy over Buffalo and Coop when some of those assumptions
are relaxed, and impressively accurate estimates of additive genetic variance in the
conditions in which it is tested (Figure 6), i.e., with random mating, no dominance,
no epistasis. However, it still needs to be stated prominently as a caveat for the
claim that the machinery is validated with individual based simulations (e.g., line
141 in the introduction) and also addressed in the Discussion section.

We have now included additional simulations incorporating dominance effects
(Figure 3, Supplementary Figure 4) and added the other caveats while describing
our simulations (Lines 160-161, Lines 510-516). We discuss the specific points in
more detail below.

The authors express the additive genetic variance at time zero as $V_A(0) = \bar{\alpha}^\top \mathbf{L}_0 \bar{\alpha}$,
where $\bar{\alpha}$ is the vector of mean average effects across time and replicates (line 192
and also in Eq. 12). But this is not correct, it should be $V_A(0) = \bar{\alpha}_0^\top \mathbf{L}_0 \bar{\alpha}_0$, where
$\bar{\alpha}_0$ is the vector of mean average effects across replicates at time 0. The authors are
approximating the initial average effects by a mean across time of average effects and
this should not be obfuscated. This is relevant to line 433 where the authors say “We
also make Assumption H if we choose to interpret $V_A(0)$ as an additive genetic vari-
ance rather than an additive genetic covariance.” With $\bar{\alpha}$ an average over time, it is
not at all clear how $\bar{\alpha}^\top \mathbf{L}_0 \bar{\alpha}$ can be interpreted as a covariance (more on this in a later
comment). It is also relevant to line 435 where the authors say that “Assumptions E
& H result in the unpredictable response to selection being zero. While we do not
make this assumption, we do assume that the unpredictable responses to selection
are independent across replicates.” If more than one time point is being used and if
that time point is not the initial time point, it is not obvious that the authors are not
making this assumption (having implicitly assumed that $\bar{\alpha}_0^\top \mathbf{L}_0 \bar{\alpha}_0 = \bar{\alpha}^\top \mathbf{L}_0 \bar{\alpha}$ from
the start)

We agree that our notation had become confused in this respect and we now use
the notation $V_{\bar{A}}(0) = \bar{\alpha}^\top \mathbf{L}_0 \bar{\alpha}$ which equals $V_A(0)$ when average effects are constant.
When they are not constant $V_{\bar{A}}(0)$ is equal to the additive genetic covariance in fit-
ness between replicate/time-points in the base population (Lines 213-219). We cover
this in more detail below where the point is raised again.

I have a great many concerns about about the section “Extending our approach to
practical situations” starting line 181, which are in some ways related to the above
comment. In line 232 the authors indicate that $\mathcal{L}_{t,m} \bar{\alpha}$ captures the predictable

change due to selection. However, this ignores the fact that average effects, allele
frequencies and linkage can all change due to selection in a way that is predictable.
In line 235 the authors say “Note that $\Delta_D \mathbf{p}$ and $\Delta_U \mathbf{p}$ have expectation zero and that
the $\Delta_D \mathbf{p}$ terms are independent across replicates, generations, and generations within
replicates (Buffalo and Coop, 2019).” I do not see how Buffalo and Coop can be
appealed to for an assertion that $\Delta_U \mathbf{p}$ has zero expectation, since Buffalo and Coop
(and Santiago and Caballero, 1998, [4]) make many assumptions that the authors of
this paper are trying to relax. I think the authors need to properly discuss the con-
ditions under which $\Delta_U \mathbf{p}$ has zero expectation as numerous strong assumptions seem
to go into that. Below I provide specifics, both about some of the assumptions that
I think are required to make that assertion, and about my problems with this section:

The authors “decompose \mathbf{L} at a particular time into a part that can be predicted
by \mathbf{L}_0 and the action of drift and recombination, and a part that cannot be pre-
dicted.” (line 206). They write the deviation of $\mathbf{L}_{t,m}$ – from the part that can be
predicted by \mathbf{L}_0 and the action of *drift and recombination* – as $\Delta \mathbf{L}_{t,m} = \Delta \mathbf{L}'_{t,m} + \mathbf{L}''_{t,m}$
where $\mathbf{L}''_{t,m}$ is the matrix of nongametic-phase disequilibria that arises in generation t
in replicate m ” (line 224) and $\Delta \mathbf{L}'_{t,m}$ is a stochastic term with zero expectation that
represents the accumulated change in \mathbf{L}' between generations 0 and t in replicate m
that cannot be predicted” (line 222). But there is a component of change in $\mathbf{L}_{t,m}$
that has been completely ignored: the change in $\mathbf{L}_{t,m}$ that can be predicted from \mathbf{L}_0
(and $\bar{\alpha}_0$) by the action of selection. Natural selection might well generate changes in
allele frequency (the diagonal of \mathbf{L}) and correlations between alleles (the off-diagonal
of \mathbf{L}) that are correlated between replicates and have nonzero expectation. Further,
these changes might well be correlated with changes in average effects due to the
action of natural selection. To neglect such changes it would appear that many of
the assumptions of Buffalo and Coop would need to be made. In addition, it would
seem that assuming $\mathbf{L}''_{t,m}$ has zero expectation requires assuming that non-gametic
phase equilibrium is not generated by natural selection.

The reviewer makes important and valid criticism here and we apologise for the
lack of clarity. Our treatment rests on the key assumption of the infinitesimal model
that selection causes only slight perturbations to allele frequencies at individual loci
(Barton *et al.*, 2017). This assumption would likely be true over the time scales of a
few generations and indeed our method performed well on simulations where these
assumptions are not met. We realise that our treatment’s reliance on the infinites-
imal model was not stated clearly enough in the main text. We have now made

this more explicit (Lines 265-287) and extended our simulations to include domi-
nance (Lines 523-535, Lines 654-678, Figure 3 and Supplementary Figure 4) where
selection-induced changes in the average effects are more likely (see below).

When $\Delta\alpha_{t,m}$ is the difference between the initial average effect $\bar{\alpha}_0$ and a deviation
from that value (as it must be if $\bar{\alpha}_0$ is going to be used to estimate the initial additive
variance), then dropping the Buffalo and Coop assumption of additive gene action
would result in $\Delta\alpha_{t,m}$ having nonzero expectation. As an example, if we assume there
is dominance then $\alpha_t = s/2(1 + (1 - 2h)(1 - 2p_t))$ (where s is the selection coefficient
against the homozygote, h is the dominance coefficient and p_t the allele frequency at
time t) and, importantly, expected changes in allele frequency at selected alleles are
likely to have nonzero expectation. Consequently, average effects will have a nonzero
expected change that is correlated across replicates.

More generally, when gene action is not additive, selection can remain constant
in the sense that alleles causal effects on fitness can remain unchanged, but average
effects on alleles might still change as allele frequencies and patterns of linkage dis-
equilibrium change (owing to dominance or epistasis). And these changes are likely
be correlated to changes in \mathbf{L} .

While we completely agree with all points, we still believe that under polygenic
selection allele frequency changes *caused by selection* will be small and so there will
be little predictable change in the average effects. However, we have now added
additional simulations incorporating dominance effects within loci (Lines 523-535,
Lines 654-678). Initially, we added dominance effects to our simplified simulations
(Figure 3a-d) where there was no history phase. In theory, this scenario should lead
to the greatest change in average effect because genetic diversity at causal loci is
high and so there is substantial dominance variance and allele frequency change is
comparatively fast. There is no obvious deterioration in our method's capabilities
compared to pure additivity. Similarly, when we allowed a history phase with ad-
ditivity, but added dominance effects in the experiment phase, our method seemed
to cope well (Supplementary Figure 4). Surprisingly, when we allowed dominance
effects in the history phase (Figure 3e), and so allele frequencies and effects had come
to an equilibrium, we found that our method overestimated V_A to varying degrees
depending on how recessive the deleterious allele was. However, this was not be-
cause of a failure of the infinitesimal approximation under the experimental phase,
but because substantial negative LD had built up during the history phase and our
model of $V_{\bar{\alpha}}$ failed to capture this: the additive genetic variance was roughly half the
additive genic variance (Supplementary Figure 5a). It seems that our simulations

were resulting in substantial pseudo-overdominance (Abu-Awad and Waller, 2023) -
a phenomenon which is characteristic of regions of very low recombination (Salson
*et al.*, 2025). When we performed identical simulations at higher recombination rates
in the history phase (Figure 3f), this problem was not observed.

From the section on ‘Comparison with Buffalo and Coop (2019)’ (starting line
338), it would seem that the authors believe that their method gets around the issues
detailed in the previous remark. This makes it seem possible I have misunderstood
their method in a way that accounts for all these concerns. In that case I still believe
that the manuscript should be revised to increase clarity and make such misunder-
standings unlikely.

Following the section where we outline the assumptions that underpin Buffalo
and Coop (2019) we detail the assumptions that we also have failed to relax (Lines
481-501) - in particular we note our method technically assumes the unpredictable re-
sponse to selection is zero which will not be the case under selection-induced changes
in \mathbf{L} and $\bar{\alpha}$. However, we do believe that the method is relatively robust to this
assumption - for example, in Supplementary Figure 2 we show that in simulations
where a (unrealistically) large fraction of V_A is lost during the experimental phase
the estimates are only slightly downwardly biased, despite their being presumably
large selection-induced changes in \mathbf{L} .

In this Section [Comparison with Buffalo and Coop (2019)], variance as a func-
tion of time is defined (correctly) in Equation 21 in a way that contradicts the
earlier definition of $V_A(0)$. What is referred to as additive genetic and genic covari-
ances (Equations 23 and 24) are also defined. To be interpreted as covariances, they
would need to be the genetic and genic covariances between the breeding value of a
random individual in generation t with the breeding value that that same individual
would have had if they had instead they existed in generation τ : thereby accounting
for changes in average effects due to changes in the environment/selection but not
changes that would occur as a result of allele frequency changes or changed in the
correlation structure between alleles. The authors claim that the early (I believe
incorrect) $V_A(0)$ is equal to this rather artificial (genetic) covariance when average
effects are not constant in time (line 399)—however it is not clear how this could
be so, i.e. it is not clear why $\alpha_t^\top \mathbf{L}_t \alpha_t$ should be approximated by $\bar{\alpha}^\top \mathbf{L}_0 \bar{\alpha}$, where
$\bar{\alpha}$ is an average of α over time, without some pretty strong assumptions. Even
more confusingly, earlier in the paper (line 291) and in the discussion (line 874), it
is stated that the covariance in fitness that the method is meant to be estimating

when average effects are not constant, is a covariance between replicates (rather than
 between times). It would helpful if this between-replicate covariance were defined,
 i.e., if it were stated between what two random variables the quantity is a covari-
 ance, since this requires some care. And also if the inconsistency in exactly which
 covariance is being estimated when average effects are not constant were resolved.
 Indeed, mathematical expressions should be provided to demonstrate that this (prop-
 erly defined) covariance actually corresponds with the quantity being estimated as
 is claimed (many times, e.g., line 434). I would also like to see some comment on
 the significance of this quantity—on the face of it, it rather artificial; is there any
 reason we would want to estimate it beyond its potentially providing a reasonable
 approximation for the additive variance?

We agree that our definition of $V_A(0)$ (Line 192 of the original submission) is
 inconsistent with the definition of $V_A(t)$ in Equation 21 and only coincides when
 average effects are constant across replicates and time-points. We have now used
 the notation $V_{\bar{A}}(0)$ to designate this quantity and show below that when the average
 effects are not constant $V_{\bar{A}}(0)$ can be interpreted as the additive genetic covariance
 between replicates/time-points for individuals sampled from the base population
 (Lines 213-219): as the reviewer points out, this is the covariance in breeding value
 an individual sampled from the base population would have if we picked two different
 replicates and two different time points. This is slightly different from our previous
 claim that it is the additive genetic covariance between replicates. We do not think
 this interpretation requires any assumptions (see below), but the process by which
 we infer this quantity from allele frequency changes across replicates does require
 assumptions. In the main text (Line 259 of the original submission) we had stated
 that we make the assumption that changes in \mathbf{L} and $\boldsymbol{\alpha}$ are independent across repli-
 cates and that (Line 249 of the original submission) we assume that the changes in \mathbf{L}
 caused by selection can be ignored (and make reference to the infinitesimal model).
 These assumptions were also detailed in Appendices S1 and S2 although we agree
 they could have been more clearly stated in the main manuscript. We have now tried
 to make these concepts clearer in the main text. (Lines 265-287)

In our original definition we had $\boldsymbol{\alpha}_{t,m} = \bar{\boldsymbol{\alpha}} + \Delta\boldsymbol{\alpha}_{t,m}$ and $\mathbf{L}_{t,m} = \boldsymbol{\mathcal{L}}_{t,m} + \Delta\mathbf{L}_{t,m}$.
 However, it is perhaps best to express the average effects as $\boldsymbol{\alpha}_{t,m} = \bar{\boldsymbol{\alpha}} + \Delta\boldsymbol{\alpha}_t + \Delta\boldsymbol{\alpha}_m +$
 $\Delta\boldsymbol{\alpha}_{t,m}$ where $\Delta\boldsymbol{\alpha}_t$ is the deviation of the average effects (averaged over replicates)
 at time t and $\Delta\boldsymbol{\alpha}_m$ is the deviation of the average effects (averaged over time) in
 replicate m , leaving $\Delta\boldsymbol{\alpha}_{t,m}$ as the ‘residual’ deviation in $\boldsymbol{\alpha}$. By definition, the terms
 $\Delta\boldsymbol{\alpha}_t$, $\Delta\boldsymbol{\alpha}_m$ and $\Delta\boldsymbol{\alpha}_{t,m}$ have expectation zero, where the expectations are taken
 over time, replicates and all time-replicate combinations respectively. $C_{\alpha_{t_m}, \alpha_{\tau_n}} =$

$\alpha_{t,m}^\top \mathbf{L}_0 \alpha_{\tau,n}$ is the additive genetic covariance between replicate m at time t and
 replicate n at time τ given the base population structure.

$$C_{\alpha_{t,m}, \alpha_{\tau,n}} = (\bar{\alpha}^\top + \Delta \alpha_t^\top + \Delta \alpha_m^\top + \Delta \alpha_{t,m}^\top) \mathbf{L}_0 (\bar{\alpha} + \Delta \alpha_\tau + \Delta \alpha_n + \Delta \alpha_{\tau,n}) \quad (1)$$

If we take the expectation of this over pairs of replicates we get:

$$C_{\alpha_t, \alpha_\tau} = (\bar{\alpha}^\top + \Delta \alpha_t^\top) \mathbf{L}_0 (\bar{\alpha} + \Delta \alpha_\tau) \quad (2)$$

since the Δ terms, and the product of pairs of Δ terms when $n \neq m$, have
 zero expectation. Here, $C_{\alpha_t, \alpha_\tau}$ is the between-replicate covariance at time t and
 τ , and has expectation (with respect to time) $\bar{\alpha}^\top \mathbf{L}_0 \bar{\alpha}$ when $t \neq \tau$ and $\bar{\alpha}^\top \mathbf{L}_0 \bar{\alpha} +$
 $Tr(\mathbf{L}_0 VAR(\Delta \alpha))$ when $t = \tau$ where $VAR(\Delta \alpha)$ are the (co)variances of $\Delta \alpha$ over
 time. We don't think this requires any assumptions.

Another concern that I had with the manuscript was that at some point (in the
 paragraph starting line 259, p9) the authors shift from talking about fixed effects
 to random effects without offering any explanation of what this means and how the
 quantities that they infer connect to those of classical quantitative genetics. $\mu_{\bar{\alpha}}$ and
 $\mathbf{V}_{\bar{\alpha}}$ are described as the mean and covariance structure of the mean average effects,
 but it is not stated what this mean is being taken over. How are we to interpret $\mathbf{V}_{\bar{\alpha}}$?
 Is it some kind of Bayesian uncertainty? (in which case, it surely should not ap-
 pear in an estimate of additive genetic variance). Upon switching to random effects,
 the authors say to "See Appendix S3 and Gianola et al. (2009) for a discussion on
 what this implies". Appendix 3, which I discuss in a later comment, touches on this
 point only tangentially; it just discusses how, given a switch to random effects, one
 needs to ensure the distribution of $\bar{\alpha}$ transforms appropriately when the reference
 and alternate allele are switched. So, to try and understand the implications, I read
 the first few sections of Gianola et al. [2] which examines "relationships between
 the (Bayesian) variance of marker effects in some regression models and additive ge-
 netic variance", and this did not bring clarity. To quote from that article: "In short,
 the connection between the uncertainty variance $\sigma_{\bar{\alpha}}^2$ and the additive variance V_A
 (which involves the effect of the locus) is elusive when both genotypes and effects
 are random variables." If the authors possess a thorough understanding of the im-
 plications of switching from fixed effects to random effects and also how the reader
 should interpret $\mu_{\bar{\alpha}}$ and $\mathbf{V}_{\bar{\alpha}}$ (and the parameters they infer to describe them: $\beta_{\bar{\alpha}}^{(0)}$,

$\beta_{\bar{\alpha}}^{(1)}$ and $p_{\bar{\alpha}}$), then this should be clearly explained. If, however, there is simply a
 lack of clarity on these issues not only in this manuscript, but elsewhere in the lit-
 erature, then this should be made transparent. In the discussion (lines 775 to 777),
 the authors say that in “the simplified simulations, in which the relationship between
 allele frequencies and α ’s was expected to be absent, the model requires only a single
 parameter: the variance in average effects, $\sigma_{\bar{\alpha}}^2$, which seems to asserting that $\sigma_{\bar{\alpha}}^2$ is
 the variance (across alleles segregating in the population) of average effects, and this
 is only true when $p_{\bar{\alpha}}$ is zero (as is indeed the case with their simplified simulations).
 More generally, it seems the randomness in $\bar{\alpha}$ assumed by the authors arises not from
 Bayesian uncertainty but from variation in average effects across loci, and that mak-
 ing the randomness locus specific is not a means to specify locus-specific uncertainty.
 Rather it is a tool to allow the assumed distribution of average effects to vary with
 allele frequency. When $p_{\bar{\alpha}}$ is zero then $\sigma_{\bar{\alpha}}^2$ is the variance of average effects, when $p_{\bar{\alpha}}$
 is minus one, $\sigma_{\bar{\alpha}}^2$ is the expected per allele contribution to additive variance, and for
 other values of $p_{\bar{\alpha}}$ the relationship is more complex. If this is so, it would be helpful
 if this, along with interpretations of $\beta_{\bar{\alpha}}^{(0)}$, $\beta_{\bar{\alpha}}^{(1)}$ and $p_{\bar{\alpha}}$) were made explicit.

 We always interpret random effects in a Bayesian context (to quote Meng (2008),
 ‘using penalized likelihood enjoys the Bayesian fruits without paying the B-club fee’).
 In short, we interpret $\mu_{\bar{\alpha}}$ and $\mathbf{V}_{\bar{\alpha}}$ as hyperparameters of the prior distribution for $\bar{\alpha}$
 with $E[V_{\bar{A}}(0)]$ then being the the marginal posterior mean of $V_{\bar{A}}(0)$. Note that we
 are only marginalising $\bar{\alpha}$ and the marginal posterior is still conditional on $\mu_{\bar{\alpha}}$, $\mathbf{V}_{\bar{\alpha}}$
 and \mathbf{L}_0 :

$$637 \quad Pr(V_{\bar{A}}(0)|\mu_{\bar{\alpha}}, \mathbf{V}_{\bar{\alpha}}, \mathbf{L}_0) \propto \int_{\bar{\alpha}} |\mathbf{J}| \cdot Pr(\bar{\alpha}|\mu_{\bar{\alpha}}, \mathbf{V}_{\bar{\alpha}})d\bar{\alpha} \quad (3)$$

 where \mathbf{J} is the Jacobian of the transform from $\bar{\alpha}$ to $V_{\bar{A}}$ and depends on \mathbf{L}_0 . The
 posterior mean is:

$$641 \quad E[V_{\bar{A}}(0)|\mu_{\bar{\alpha}}, \mathbf{V}_{\bar{\alpha}}, \mathbf{L}_0] \propto \int_{\bar{\alpha}} \bar{\alpha}^{\top} \mathbf{L}_0 \bar{\alpha} \cdot |\mathbf{J}| \cdot Pr(\bar{\alpha}|\mu_{\bar{\alpha}}, \mathbf{V}_{\bar{\alpha}}, \mathbf{L}_0)d\bar{\alpha} \quad (4)$$

 which is equivalent to the equation between Equations 17 and 18 of Gianola *et al.*
 (2009) (or Equations A1-A3 in their appendix). Their earlier comment about the
 connection between the uncertainty variance and the additive variance being elusive
 only seems to apply to the single locus-case. In the multilocus case they seem to
 be happy to interpret it as a posterior mean and only seem to be concerned with
 the fact that the ‘marked’ genetic variance might only be a small fraction of the

total. Since we are envisaging all segregating sites are included, this criticism is not
 pertinent. If we had used an MCMC approach, rather than a REML approach, we
 could also have treated $\boldsymbol{\mu}_{\bar{\alpha}}$ and $\mathbf{V}_{\bar{\alpha}}$ as random variables with their own hyper-priors
 and obtained the full marginal posterior mean:

$$E[V_{\bar{A}}(0)|\mathbf{L}_0] \propto \int_{\mathbf{V}_{\bar{\alpha}}} \int_{\boldsymbol{\mu}_{\bar{\alpha}}} \int_{\bar{\alpha}} \bar{\alpha}^\top \mathbf{L}_0 \bar{\alpha} \cdot |\mathbf{J}| Pr(\bar{\alpha}|\boldsymbol{\mu}_{\bar{\alpha}}, \mathbf{V}_{\bar{\alpha}}, \mathbf{L}_0) Pr(\boldsymbol{\mu}_{\bar{\alpha}}, \mathbf{V}_{\bar{\alpha}}|\mathbf{L}_0) d\bar{\alpha} d\boldsymbol{\mu}_{\bar{\alpha}} d\mathbf{V}_{\bar{\alpha}} \quad (5)$$

We have now extended the appendix to include these points (Supplementary
 Information S3: Lines 1479-1488). In addition, our previous submission was a little
 confusing in that we actually treated $\bar{\alpha}$ as random prior to the inference section. We
 have now ensured that prior to the inference section, expectations and (co)variances
 are only taken with respect to the evolutionary process, and in the inference section
 they are taken with respect to both the evolutionary process and the distribution of
 $\bar{\alpha}$.

The reviewer is correct that $\sigma_{\bar{\alpha}}^2$ can only be interpreted as the variance in average
 effects (unconditional on \mathbf{L}_0) over loci when $p_{\bar{\alpha}} = 0$ (and when $\beta_{\bar{\alpha}}^{(1)} = 0$) and we have
 made this clear and now state that when $p_{\bar{\alpha}} = 0$ average effects are independent
 of genetic diversity, and when $p_{\bar{\alpha}} = -1$ average effects scale inversely with genetic
 diversity. $p_{\bar{\alpha}} = -1$ is a common assumption in many related approaches (e.g. Yang
 *et al.*, 2011) and under this assumption $\sigma_{\bar{\alpha}}^2$ is the average contribution of a locus to
 the additive genic variance. (Lines 338-342)

It took me quite a while to get to grips with the purpose of Appendix 3 (starting
 line 1225) and I am still not completely confident that I have. I think it would be
 helpful to orient the reader, if the authors state near the beginning of the Appendix
 that the purpose of the Appendix is to obtain assumptions on distribution of ran-
 dom effects which ensure that the distribution transforms appropriately when the
 reference and alternate allele are switched—and to provide candidates for the mean
 and variance that comply with these assumptions. This, at least, is the purpose of
 the Appendix as I understand it. The authors conclude that a choice of $\mathbf{V}_{\bar{\alpha}} = \sigma_{\bar{\alpha}}^2 \mathbf{L}_0^{p_{\bar{\alpha}}}$
 satisfies the requirement that covariances involving a locus switch sign when its allele
 coding is flipped (but its variance remains positive). The authors also say (line 1256)
 “For, $\boldsymbol{\mu}_{\bar{\alpha}}$ this implies that suitable models should be (weighted) sums of differences
 between invariant properties of the alleles such that the difference reverses sign when
 the reference and alternate allele are switched.” This is confusing given that in the
 main text the authors model the mean effects as: $\boldsymbol{\mu}_{\bar{\alpha}} = \beta_{\bar{\alpha}}^{(0)} + \beta_{\bar{\alpha}}^{(1)}(\mathbf{p}_0 - \mathbf{q}_0)$ (Equa-

tion 15 on page 11) which, for $\beta_0 \neq 0$ does not immediately appear to satisfy this requirement. It would be helpful if the authors stated explicitly in the Appendix how their conclusions about permissible models for $\boldsymbol{\mu}_{\bar{\alpha}}$ are captured by the model employed in the main text. Unrelated, but also in Appendix S3, Equation S19 differs from Equation 12 in the main text in ways that are confusing: $V_A(0)$ changes to V_A and the expectation gets a subscript of \mathbf{L}_0 the purpose of which is unclear.

We have now extended the Appendix (now referred to as ‘Supplementary Information’) so that initially we really do discuss the consequences of treating $\bar{\alpha}$ as random. We thank the reviewer for identifying the issue that using $\boldsymbol{\mu}_{\bar{\alpha}} = \beta_{\bar{\alpha}}^{(0)} + \beta_{\bar{\alpha}}^{(1)}(\mathbf{p}_0 - \mathbf{q}_0)$ does not result in a model that is invariant to reference allele assignment for $\beta_{\bar{\alpha}}^{(0)} \neq 0$. This was indeed a mistake. We now drop the intercept and simply use $\boldsymbol{\mu}_{\bar{\alpha}} = \beta_{\bar{\alpha}}^{(1)}(\mathbf{p}_0 - \mathbf{q}_0)$ (Lines 332-332). It is important to note, however, that reanalysing our original simulations using this new model does not affect our estimates in any meaningful way. We also apologise for confusing \mathbf{L}_0 subscript to COV - this was a hangover from a previous choice of notation and has now been removed.

We also agree with the reviewer that our appendices (now referred to as ‘Supplementary Information’) require some introductory text that helps orient the reader. We have now corrected this issue.

Near the beginning of the section on the Inference Outline, the authors say (line 299) ‘We chose a projection that collapses allele frequency changes into the non-null subspace of \mathbf{L}_0 , since $V_A(0)$ only depends on this subspace (de Los Campos *et al.*, 2015).’ I took an (admittedly fairly cursory) look at de Los Campos *et al.* (2015) and so far as I can tell the goal of this paper was to develop a rigorous definition of “genomic heritability” within a classical quantitative genetics framework, distinguishing it from statistical model-based interpretations used in whole-genome regressions. There is a particular focus on highlighting that estimates from marker-based models may suffer from bias or inconsistency when causal variants are not fully captured, and the paper cautions against naive inference from such models. However, I could not locate the result that $V_A(0)$ depends only on the non-null subspace of \mathbf{L}_0 cited by the authors from that paper. If $V_A(0)$ does indeed only depend on the non-null subspace of \mathbf{L}_0 , then I believe work is required to move from what is presented in de Los Campos *et al.* (2015) to that conclusion. It does seem fairly plausible; nevertheless, if the authors are just assuming it to hold, then there should be transparency about this.

Our assertion is not directly stated by de Los Campos *et al.* (2015), but instead,

it requires a few short logical steps from their results. We apologise for this oversight.

In Supplementary Methods I, de Los Campos *et al.* (2015) show that additive
genetic variances are ‘rotationally invariant’; i.e. they are invariant to linear trans-
formations of the genotypes. Therefore, $V_A(0)$ should remain unchanged if we express
the genotype matrix and the α ’s using a basis defined by the eigenvectors of the orig-
inal \mathbf{L}_0 . The covariance matrix for the transformed genotypes is then diagonal (since
eigenvectors are orthogonal) with as many non-zero elements on the diagonal (i.e.
the eigenvalues of \mathbf{L}_0) as the rank (R) of the original \mathbf{L}_0 matrix. It is then straightfor-
ward to see that the additive genetic variance depends only on the first R dimensions
corresponding to non-zero eigenvalues of \mathbf{L}_0 , i.e. the non-null subspace of \mathbf{L}_0 . We
have now added this explanation to the manuscript, while also pointing the reader
specifically to Supplementary Methods I, de Los Campos *et al.* (2015). (Supplemen-
tary Information S4: Lines 1537-1553)

In numerous places in the manuscript, subscripts are given subscripts in a way
that seems inappropriate. For example, in Equation (S4) I strongly believe that
$\mathbf{L}''_{i_{t+1},j_{t+1+1}}$ is *meant* to denote the (i, j)th element of \mathbf{L}'' at time $t + 1$. If this is
correct, then $t + 1$ should be a subscript (or superscript or a something) attached
directly to \mathbf{L}'' , not to the i and j (unless the specific loci being indexed are actually
are actually changing with time, which seems problematic). This issue exists in many
places for many variables and should be corrected throughout.

Changed to $\mathbf{L}''_{t+1,ij}$, and corrected throughout.

Reviewer 3

This manuscript presents a novel theoretical framework for estimating the addi-
tive genetic variance for relative fitness directly from genome-wide changes in allele
frequency observed across experimental replicates. The authors build upon Fisher’s
Fundamental Theorem of Natural Selection to develop mathematical relationships
between V_A , linkage disequilibrium matrices, and expected changes in allele frequency
due to selection. The theoretical contribution is substantial and mathematically
rigorous. The authors derive an interesting and general expression showing that
$V_A = E[\Delta p]^\top L^{-1} E[\Delta p]$, where $E[\Delta p]$ is the expected changes in allele frequency
and L is the genome-wide diversity/linkage disequilibrium matrix. This approach
elegantly sidesteps many of the practical difficulties of traditional pedigree-based
methods for estimating V_A , particularly the challenge of defining appropriate fitness

proxies. The method relaxes several restrictive assumptions of previous approaches,
notably the B&C method, including the requirement to distinguish selected from
neutral sites and assumptions about linkage disequilibrium patterns.

The simulation studies are comprehensive and demonstrate that the method pro-
vides unbiased estimates of V_A across a wide range of biologically realistic parameter
combinations. The comparison with B&C clearly shows superior performance, par-
ticularly at low recombination rates where alternative approaches exhibit substantial
bias.

The mathematical complexity is indeed formidable and may limit accessibility
to the genetics community. The derivation spans multiple appendices with intricate
notation that could benefit from simplification or more intuitive explanation. Al-
though the authors provide some biological interpretations, the gap between theory
and practical implementation is still substantial.

Reviewer 1 also had similar concerns. We have now included a box ('Box 1') that
describes the workflow for applying our method to real data. (Line 1830)

A significant limitation is the lack of any empirical example. For a method
claiming practical utility, demonstration on real evolve-and-resequence data would
significantly strengthen the manuscript's impact. The data requirements are: phased
genomes from base populations, precise recombination maps, and carefully controlled
experimental replicates. These prerequisites may severely constrain the method's
practical applicability.

One of our main goals is to apply our method to data from a real evolve and
resequence experiment we performed recently. Briefly, we evolved 20 replicate pop-
ulations of *Drosophila melanogaster* (all derived from the same 1,000 wild-caught
parents) in either a typical laboratory-like environment or in a 'seminatural' envi-
ronment in outdoor cages. We aim to estimate V_A in the two environments, as well
as the additive genetic covariance for fitness between the two environments. We are
currently processing the data from this experiment. We feel that we have collected a
fairly rich dataset that warrants a separate manuscript in its own right. Furthermore,
including those data here would almost certainly overload the current manuscript,
particular given that new extensions to the model/code are required. These include
a) allowing a multivariate structure to estimate the between-environment genetic
correlation b) developing new methods for performing singular-value decomposition
on very large matrices and c) new methods for estimating the number of generations
that have elapsed in an evolve-and-resequence study based on the frequency of re-

combinant reads d) incorporating sex-linkage into the theory and methodology.

The authors acknowledge that several assumptions may be violated in real popu-
lations, including independence of unpredictable selection responses across replicates
and potential complications from varying selection environments.

We agree with the reviewer that applying our method to real systems is likely to
entail numerous challenges. We have attempted to acknowledge these limitations in
a transparent way in the Discussion section. (Lines 985-1062)

The manuscript would benefit from a clearer presentation of the core algorithm
and its computational requirements. The development of accompanying software
would enhance its potential for application. Additionally, while the discussion of
limitations is thorough, it could be helpful for readers to assess the practical feasi-
bility.

We have now included a box ('Box 1') that describes the workflow of the core
algorithm and have made a publicly available R library (`V_w`) that implements our
method - available here.

References

Abu-Awad, D., and D. Waller, 2023 Conditions for maintaining and eroding pseudo-
overdominance and its contribution to inbreeding depression. *Peer Community*
*Journal* **3**.

Barton, N. H., A. M. Etheridge, and A. Véber, 2017 The infinitesimal model: Defi-
nition, derivation, and implications. *Theoretical population biology* **118**: 50–73.

Bitter, M., S. Berardi, H. Oken, A. Huynh, E. Lappo *et al.*, 2024 Continuously
fluctuating selection reveals fine granularity of adaptation. *Nature* **634**: 389–396.

Buffalo, V., and G. Coop, 2019 The linked selection signature of rapid adaptation in
temporal genomic data. *Genetics* **213**: 1007–1045.

Dabi, A., and D. R. Schrider, 2025 Population size rescaling significantly biases out-
comes of forward-in-time population genetic simulations. *Genetics* **229**: iyae180.

de Los Campos, G., D. Sorensen, and D. Gianola, 2015 Genomic heritability: what
is it? *PLoS Genetics* **11**: e1005048.

- Gianola, D., G. de Los Campos, W. G. Hill, E. Manfredi, and R. Fernando, 2009
Additive genetic variability and the Bayesian alphabet. *Genetics* **183**: 347–363.
- Lynch, M., and B. Walsh, 1998 *Genetics and analysis of quantitative traits*. Sinauer
Associates, Inc., Sunderland.
- Meng, X.-L., 2008 Discussion: one-step sparse estimates in nonconcave penalized
likelihood models: Who cares if it is a white cat or a black cat? *The Annals of*
*Statistics* **36**: 1542–1552.
- Salson, M., M. Duranton, S. Huynh, C. Mariac, C. Tranchant-Dubreuil *et al.*, 2025
Interplay between large low-recombining regions and pseudo-overdominance in a
plant genome. *Nature Communications* **16**: 6458.
- Tilk, S., A. Bergland, A. Goodman, P. Schmidt, D. Petrov *et al.*, 2019 Accurate al-
lele frequencies from ultra-low coverage pool-seq samples in evolve-and-resequence
experiments. *G3: Genes, Genomes, Genetics* **9**: 4159–4168.
- Yang, J., S. H. Lee, M. E. Goddard, and P. M. Visscher, 2011 Gcta: a tool for
genome-wide complex trait analysis. *The American Journal of Human Genetics*
**88**: 76–82.

September 26, 2025

GENETICS-2025-308527

Estimating the additive genetic variance for relative fitness from changes in allele frequency

Dear Dr. Arun:

I am pleased to inform you that, with minor revisions, your MS is potentially suitable for publication in GENETICS. The previous reviewer is satisfied that the most important points have been addressed, but still has some comments and concerns that need to be addressed in a revised manuscript. You can read their review in the attached PDF.

The revision is a substantial improvement, and deals with most of the key points raised before. However, it would be very helpful to gather together your various assumptions, and to really clarify the role of the covariance, versus the additive variance, as detailed by the reviewer. This will be a significant paper, and it is important to make it as clear as possible.

We look forward to receiving your revised manuscript. Please let the editorial office know approximately how long you expect to need for revisions.

Upon resubmission, please include:

1. A clean version of your manuscript;
2. A marked version of your manuscript in which you highlight significant revisions carried out in response to the major points raised by the editor/reviewers (track changes is acceptable if preferred);
3. A detailed response to the editor's/reviewers' comments and to the concerns listed above. Please reference line numbers in this response to aid the editors.

Additionally, please ensure that your resubmission is formatted for GENETICS.

<https://academic.oup.com/genetics/pages/general-instructions>

Follow this link to submit the revised manuscript: Link Not Available

Sincerely,

Nick Barton
Senior Editor
GENETICS

Approved by:
Howard Lipshitz
Editor in Chief
GENETICS

Reviewer #2 :

My review is in the attached PDF comments_to_authors_2.pdf

Review for: Estimating the Additive Genetic Variance for Relative Fitness from Changes in Allele Frequency

This is my second review of this paper, and the authors have made substantial revisions in response to the reviews. The paper is now clearer in places, the mathematical notation is more consistent, and the addition of simulations with dominance is welcome. I appreciate the corrections to the earlier inconsistencies and the new workflow figure, which improves accessibility. That said, several of my major concerns remain only partly addressed. I outline these below.

1. Interpretation of $V_A(0)$ as variance vs covariance.

In my first review I raised two points here.

- I asked the authors to show explicitly how their definition of $V_A(0)$ can be interpreted as a covariance, rather than a variance, when average effects change. They have now done this.
- I also said that equating this covariance to a variance requires strong assumptions. Here, the authors appear to have misunderstood me, taking my remark to mean that assumptions were needed for it to be a covariance at all. That was not my point.

The remaining issue is that they still have not explained why this covariance, as opposed to variance, is a biologically meaningful target of inference in its own right, rather than simply a proxy for V_A . If the real interest is only in how closely it tracks V_A under certain assumptions, the manuscript should say so explicitly, rather than leaving the impression that this covariance is an equally natural object of interest.

2. Assumptions and use of the infinitesimal model.

The authors now acknowledge that their method relies on the infinitesimal model. This is an important clarification. However, I am concerned that they invoke the infinitesimal model selectively. In particular, they justify neglecting *unpredictable* selection-induced changes in LD and average effects under the infinitesimal model, while still treating the *predictable* component as informative. Yet, I believe that under the strict infinitesimal model, both predictable and unpredictable per-locus responses to selection could be neglected. If the authors intend instead some new kind of looser regime, in which unpredictable components are negligible but predictable responses remain, then this should be stated explicitly.

More generally, while the authors lay out Buffalo & Coop's assumptions with welcome clarity, their own assumptions remain dispersed. As I understand it, their framework still requires substantial assumptions. Rather than being scattered across the text and appendices, these should be collected together in one place in the manuscript—as they did for Buffalo & Coop—including all assumptions required by the theoretical framework, even those for which the simulations show a limited robustness. This would make the scope and limitations of the method much clearer to readers.

3. Readability.

The workflow “Box 1” is a helpful addition, but the methods section is still quite dense. The notation also remains heavy, with expectations sometimes taken over time and sometimes over replicates, but not always made explicit in the main text. I would encourage the authors put some thought into making the exposition easier to digest overall. In addition, the discussion could do a better job of separating assumptions and limitations from future directions. At present some important caveats are not truly discussed or in some cases even mentioned in the Discussion—such as the fact that sometimes the estimator actually recovers a covariance across replicates/timepoints (which only equals the variance under fairly strong assumptions), and that the theory depends on the infinitesimal model.

Reviewer Attachment: September 26, 2025

This revision is a significant improvement, and the authors have responded constructively to most comments. However, some central concerns about what exactly is being estimated (variance vs covariance), and about the scope and assumptions of the method, remain only partly resolved. For the paper to reach the clarity it deserves, the authors should:

- Explicitly distinguish between covariance and variance (preferably in their notation too), and state plainly that equating the two requires strong assumptions.
- Either justify the biological interest of the covariance directly, or say explicitly that it is valuable only insofar as it tracks V_A .
- Collect their own assumptions into a clear, explicit list (as they do for Buffalo & Coop), and clarify whether they mean the strict infinitesimal model or a looser regime.

Associate editor

The revision is a substantial improvement, and deals with most of the key points raised before. However, it would be very helpful to gather together your various assumptions, and to really clarify the role of the covariance, versus the additive variance, as detailed by the reviewer. This will be a significant paper, and it is important to make it as clear as possible.

We thank the reviewers and the associate editor for their constructive comments and suggestions throughout. Addressing this feedback has resulted in substantial improvements in the quality of our manuscript.

In the latest round of revision we have made the following changes:

- We have now added a subsection header (page 7 line 8) to make the list of our assumptions more prominent. We have also changed the presentation of our assumption from a dense paragraph to an itemised list (page 7 lines 8-43). Additionally, right at the beginning of the Materials and Methods section, we now refer the readers to this subsection. (page 2 lines 118-120)
- Based on the point raised by Reviewer 2, we have now deleted references to the infinitesimal model when discussing unpredictable responses to selection caused by systematic selection-induced changes in the genetic diversity/LD matrix and/or average effects for fitness. Instead, we simply state that if the genetic architecture of fitness is sufficiently polygenic, the effect of such changes on the total expected allele frequency change due to selection may be small relative to the predictable component of allele frequency change due to selection. We continue to explicitly state these assumptions and caveats in the subsection 'Extending our approach to practical situations' and address them in the Discussion section. We highlight the fact that our method provides good estimates of V_A despite none of these assumptions being baked into our simulations. (page 4 lines 22-42; page 12 lines 8-48)
- We have now added a short note explaining that the additive genetic covariance between replicates/time-points is not itself a target of inference. We point out that conventional pedigree-based approaches for measuring V_A also, strictly speaking, estimate the additive genetic (co)variance between environments in which relatives reside. We hope this makes the relationship between V_A and the additive genetic covariance between replicates/time-points clearer. (page 3 lines 70-75)

- We have now re-formatted the manuscript to follow the stylistic requirements of *Genetics*.

Reviewer 2

This is my second review of this paper, and the authors have made substantial revisions in response to the reviews. The paper is now clearer in places, the mathematical notation is more consistent, and the addition of simulations with dominance is welcome. I appreciate the corrections to the earlier inconsistencies and the new workflow figure, which improves accessibility. That said, several of my major concerns remain only partly addressed. I outline these below.

1. **Interpretation of $V_A(0)$ as variance vs covariance.** In my first review I raised two points here.
 - I asked the authors to show explicitly how their definition of $V_A(0)$ can be interpreted as a covariance, rather than a variance, when average effects change. They have now done this.
 - I also said that equating this covariance to a variance requires strong assumptions. Here, the authors appear to have misunderstood me, taking my remark to mean that assumptions were needed for it to be a covariance at all. That was not my point.

The remaining issue is that they still have not explained why this covariance, as opposed to variance, is a biologically meaningful target of inference in its own right, rather than simply a proxy for V_A . If the real interest is only in how closely it tracks V_A under certain assumptions, the manuscript should say so explicitly, rather than leaving the impression that this covariance is an equally natural object of interest.

We had alluded to this point in the Discussion section: “Although rarely made explicit, estimates from wild systems should be interpreted in the same way: the additive genetic covariance between the environments in which relatives live...” (Lines 1047-1052 in Revision 1 of the manuscript)

However, we realise that this also needs to be prominently stated in the section ‘Extending our approach to practical situations’, which we have now done. (page 3 lines 70-75)

2. **Assumptions and use of the infinitesimal model.** The authors now acknowledge that their method relies on the infinitesimal model. This is an important clarification. However, I am concerned that they invoke the infinitesimal

model selectively. In particular, they justify neglecting unpredictable selection-induced changes in LD and average effects under the infinitesimal model, while still treating the predictable component as informative. Yet, I believe that under the strict infinitesimal model, both predictable and unpredictable per-locus responses to selection could be neglected. If the authors intend instead some new kind of looser regime, in which unpredictable components are negligible but predictable responses remain, then this should be stated explicitly.

We thank the reviewer for pointing this out. We agree that it was wrong to invoke the infinitesimal model selectively to ignore the unpredictable response to selection while retaining the predictable component. We no longer invoke the infinitesimal model. Instead, we simply state the following:

“However, if the genetic architecture of fitness is sufficiently polygenic, such that selection coefficients associated with individual loci are small, selection-induced changes in \mathbf{L} may be safely ignored. In other words, selection-induced changes in \mathbf{L} may cause the total allele frequency change due to selection to be negligibly different from $\mathcal{L}_m \bar{\alpha}$, the expected allele frequency change due to selection.” (page 4 lines 29-35)

It is important to note, however, that we had already stated the assumptions pertaining to the unpredictable response to selection – as well as the scenarios under which they might fail – in fairly transparent terms in Lines 286-278 of Revision 1. We continue to retain all these caveats. (page 4 lines 22-42)

Crucially, none of these assumptions are baked into our simulations, in which both \mathbf{L} (in all simulations) and the α 's (in simulations implementing dominance) are free to change as a result of selection. Yet, our method provides decent estimates of V_A . While we had already highlighted this in the Discussion in the context of dominance effects, we have now also added a short paragraph on selection-induced changes in \mathbf{L} . (page 12 lines 26-48)

More generally, while the authors lay out Buffalo & Coop's assumptions with welcome clarity, their own assumptions remain dispersed. As I understand it, their framework still requires substantial assumptions. Rather than being scattered across the text and appendices, these should be collected together in one place in the manuscript—as they did for Buffalo & Coop—including all assumptions required by the theoretical framework, even those for which the simulations show a limited robustness. This would make the scope and limitations of the method much clearer to readers.

In addition to describing our assumptions as they naturally arise at various

points, we had also listed them together immediately after describing the assumptions of Buffalo and Coop. We think this is the most natural place for such a list. However, we agree that the list of our assumptions could be highlighted more prominently. We have now added a subsection header to that effect and collated the assumptions into an itemised list to improve clarity (page 7 lines 8-43). Furthermore, we realise that it is also important to direct the readers towards the compiled list of assumptions. We have now added a reference at the beginning of the 'Materials and Methods' section. (page 2 lines 118-120)

3. **Readability.** The workflow "Box 1" is a helpful addition, but the methods section is still quite dense. The notation also remains heavy, with expectations sometimes taken over time and sometimes over replicates, but not always made explicit in the main text. I would encourage the authors put some thought into making the exposition easier to digest overall.

We agree that our expectations and (co)variances are computed over either evolutionary realisations, individuals, replicates/time points, or Bayesian uncertainty. At most places, we had described in words the variable over which expectations/(co)variances were taken. We have now made further modifications to the manuscript to make this clearer wherever required. (eg. page 3 line 27; page 4 lines 52-53)

We acknowledge that our treatment remains difficult. Unfortunately, we cannot think of a way of explaining our approach that is both easy to understand and retains all the essential aspects of our method.

In addition, the discussion could do a better job of separating assumptions and limitations from future directions. At present some important caveats are not truly discussed or in some cases even mentioned in the Discussion—such as the fact that sometimes the estimator actually recovers a covariance across replicates/timepoints (which only equals the variance under fairly strong assumptions), and that the theory depends on the infinitesimal model.

As described above, we had already discussed the fact that many estimators of additive genetic variance actually recover the additive genetic covariance (Lines 1047-1052 in Revision 1 of the manuscript).

Based on the reviewer's point earlier about the applicability of the infinitesimal model to our method, we have now now deleted references to the infinitesimal model while referring to our assumptions about the unpredictable response to selection. Instead, we briefly describe the relevant assumptions in the Discussion and point out that these assumptions are likely to be met if the genetic

architecture of fitness is sufficiently polygenic. We also emphasise the fact that despite relaxing these assumptions in our simulations, our method provides good estimates of V_A . (page 12 lines 8-48)

This revision is a significant improvement, and the authors have responded constructively to most comments. However, some central concerns about what exactly is being estimated (variance vs covariance), and about the scope and assumptions of the method, remain only partly resolved. For the paper to reach the clarity it deserves, the authors should:

- Explicitly distinguish between covariance and variance (preferably in their notation too), and state plainly that equating the two requires strong assumptions.
- Either justify the biological interest of the covariance directly, or say explicitly that it is valuable only insofar as it tracks V_A .

We explicitly state that $V_{\bar{A}}(0) = V_A(0)$ only if the average effects for fitness remain constant, otherwise $V_{\bar{A}}(0)$ must be interpreted as an additive genetic covariance between replicates/time points. We then point out the parallels with conventional methods for estimating V_A . (page 2 lines 70-75)

- Collect their own assumptions into a clear, explicit list (as they do for Buffalo & Coop), and clarify whether they mean the strict infinitesimal model or a looser regime.

We have now added a subsection header and re-organised the list of our assumptions into an itemized list (page 7 line 8-43) and we refer to this subsection at the beginning of the Materials and Methods section (page 2 lines 118-120).

We have now removed references to the infinitesimal model and instead use the looser regime of the genetic architecture for fitness being polygenic. (page 4 line 22-42)

October 16, 2025

RE: GENETICS-2025-308683

Dr. Manas Geeta Arun
University of Edinburgh
Institute of Ecology and Evolution
Ashworth Laboratories
Charlotte Auerbach Road
Edinburgh EH11 1UW
United Kingdom

Dear Dr. Arun:

Congratulations, your manuscript titled "Estimating the additive genetic variance for relative fitness from changes in allele frequency" is accepted for publication in GENETICS! Your response to reviewers' criticisms was thorough, and has substantially clarified the paper. Many thanks for submitting your research to the journal.

To Proceed to Publication:

1. Format your article according to GENETICS style: <https://academic.oup.com/genetics/pages/author-guidelines>
2. Ensure that you comply with data and community resource citation guidelines:
<https://academic.oup.com/genetics/pages/author-guidelines#section-5-9-2>
3. Upload your final files at <https://genetics.msubmit.net>
4. Add oupsupport@scipris.com and genetics.oup@novatechset.com (or the domains @scipris.com and @novatechset.com) to your email program's "safe senders" list. You will be contacted by both at various points during the production process.

Notes:

- Your currently-accepted manuscript (unedited, as submitted, reviewed, and accepted) will be published at GENETICS and deposited into PubMed as an Advance Access article. Notify sourcefiles@thegsajournals.org before signing your license if you do not wish to publish your article via Advance Access.
- We invite you to submit an original color figure related to your paper for consideration as cover art. Please email your submission to the editorial office or upload it with your final files. You can submit a small-sized image for evaluation, and if selected, the final image must be a TIFF file 2513px wide by 3263px high (8.375 by 10.875 inches; resolution of 600ppi). Please avoid graphs and small type.
- After files are sent to Oxford University Press we use SciPris to manage article licensing and payment. If you do not have a SciPris account, you will receive an email from no-reply@scipris.com to sign up to use Oxford University Press' author portal. After logging in, follow the online instructions to sign your license and arrange any payment due.

If you have any questions or encounter any problems while uploading your accepted manuscript files, please email the editorial office at sourcefiles@thegsajournals.org.

Sincerely,

Nick Barton
Senior Editor
GENETICS

Approved by:
Howard Lipshitz
Editor in Chief
GENETICS